# Folding of cohesin's coiled coil is important for Scc2/4-induced association with chromosomes

**Naomi J Petela[1†], Andres Gonzalez Llamazares[2†], Sarah Dixon[1], Bin Hu[3], Byung-Gil Lee[2], Jean Metson[1], Heekyo Seo[4], Antonio Ferrer-Harding[1], Menelaos Voulgaris[1], Thomas Gligoris[1], James Collier[1], Byung-Ha Oh[4], Jan Löwe[2]\*, Kim A Nasmyth[1]\***

[1]Department of Biochemistry, University of Oxford, Oxford, United Kingdom; [2]MRC Laboratory of Molecular Biology, Cambridge, United Kingdom; [3]Institute of Medical Sciences, University of Aberdeen, Aberdeen, United Kingdom; [4]Department of Biological Sciences, KAIST Institute for the Biocentury, Cancer Metastasis Control Center, Korea Advanced Institute of Science and Technology, Daejeon, Republic of Korea

**\*For correspondence:**
jyl@mrc-lmb.cam.ac.uk (JL);
ashley.nasmyth@bioch.ox.ac.uk (KAN)

[†]These authors contributed equally to this work

**Competing interests:** The authors declare that no competing interests exist.

**Abstract** Cohesin's association with and translocation along chromosomal DNAs depend on an ATP hydrolysis cycle driving the association and subsequent release of DNA. This involves DNA being 'clamped' by Scc2 and ATP-dependent engagement of cohesin's Smc1 and Smc3 head domains. Scc2's replacement by Pds5 abrogates cohesin's ATPase and has an important role in halting DNA loop extrusion. The ATPase domains of all SMC proteins are separated from their hinge dimerisation domains by 50-nm-long coiled coils, which have been observed to zip up along their entire length and fold around an elbow, thereby greatly shortening the distance between hinges and ATPase heads. Whether folding exists in vivo or has any physiological importance is not known. We present here a cryo-EM structure of the *apo* form of cohesin that reveals the structure of folded and zipped-up coils in unprecedented detail and shows that Scc2 can associate with Smc1's ATPase head even when it is fully disengaged from that of Smc3. Using cysteine-specific crosslinking, we show that cohesin's coiled coils are frequently folded in vivo, including when cohesin holds sister chromatids together. Moreover, we describe a mutation (*SMC1D588Y*) within Smc1's hinge that alters how Scc2 and Pds5 interact with Smc1's hinge and that enables Scc2 to support loading in the absence of its normal partner Scc4. The mutant phenotype of loading without Scc4 is only explicable if loading depends on an association between Scc2/4 and cohesin's hinge, which in turn requires coiled coil folding.

## Introduction

SMC complexes are highly conserved from prokaryotes to eukaryotes. Best characterised among this family are cohesin and condensin, both of which are DNA translocases (*Ganji et al., 2018*; *Davidson et al., 2019*; *Kim et al., 2019*; *Golfier et al., 2020*). Cohesin and condensin are thought to organise chromosomes in eukaryotes during interphase and mitosis respectively by producing long loops of DNA (*Nasmyth, 1982*), a process called loop extrusion (LE). Cohesin has an additional property, namely the ability to hold sister DNAs together from their genesis during S phase till their eventual disjunction to opposite poles of the cell during anaphase.

Cohesin is composed of two rod-shaped SMC proteins, Smc1 and Smc3, with a dimerisation interface at one end that is connected to an ABC-like ATPase domain via a 50-nm-long coiled coil (*Figure 1A*). Interaction via their dimerisation domains produces a V-shaped Smc1/3 heterodimer

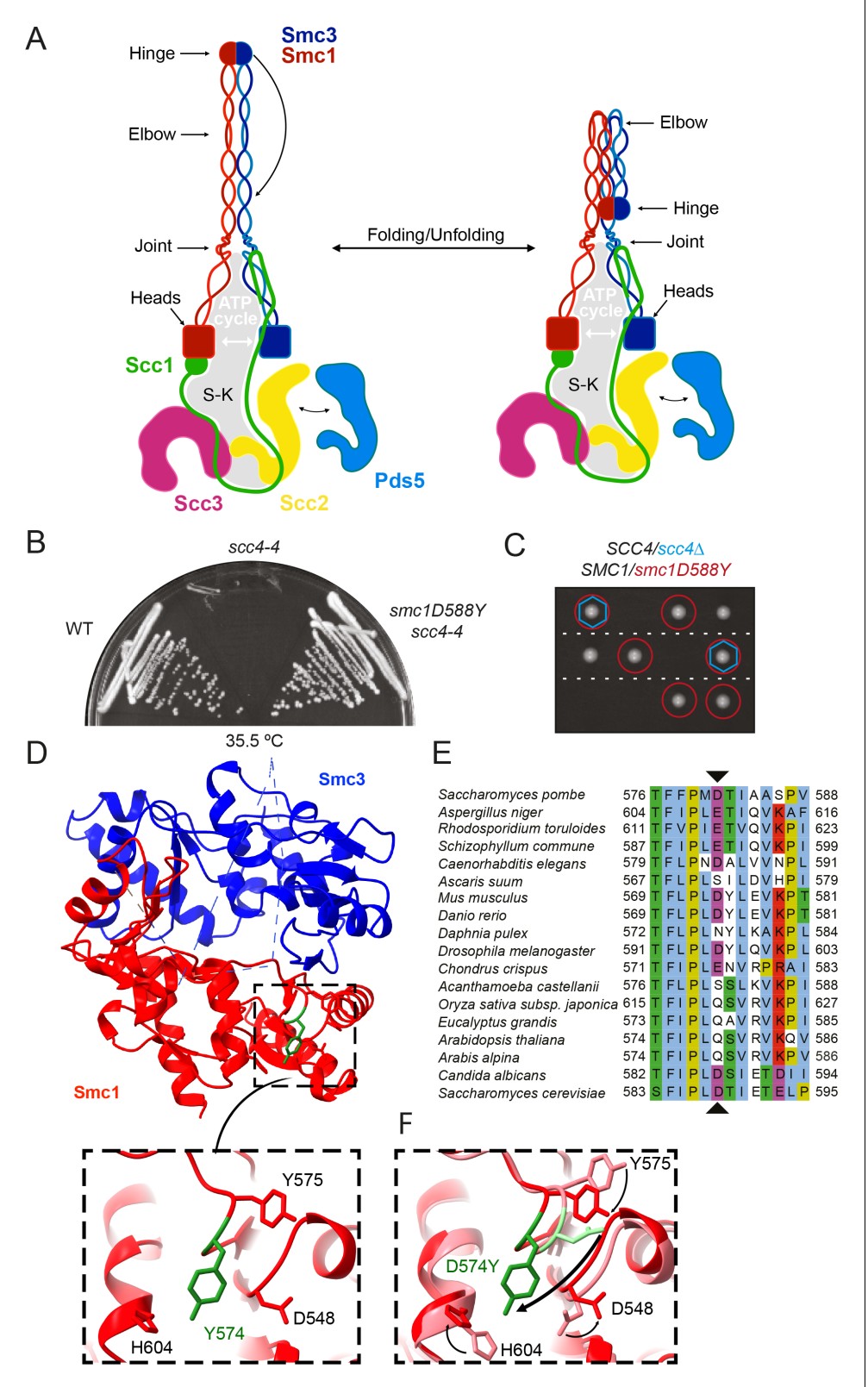

**Figure 1.** A mutation in the hinge domain of Smc1 restores viability in the absence of Scc4. (**A**) Schematic representation of *Saccharomycescerevisiae* cohesin complex and its folding cycle. (**B**) Comparison of growth of wild-type (WT), *scc4-4,* and *scc4-4 smc1D588Y* strains at 35.5℃ (K699, K8326, K19813). (**C**) Tetrad dissection of diploid strains containing *SCC4/scc4Δ SMC1/smc1D588Y* grown at 30℃. Spores expressing *smc1D588Y* are circled in red, and spores that lack Scc4 are indicated with blue hexagons. (**D**) Structure of the mouse Smc3-Smc1D574Y hinge domain (PDB: 7DG5). (**E**) Multiple

*Figure 1 continued on next page*

*Figure 1 continued*

sequence alignment indicating conservation of Smc1D588. (**F**) Structural superposition of the WT hinge and the D574Y mutant hinge. Tyr574 swings out relative to the position of D574 with a concomitant local conformational change of the mutated loop.

The online version of this article includes the following figure supplement(s) for figure 1:

**Figure supplement 1.** A mutation in the hinge domain of Smc1 restores viability in the absence of Scc4.

whose two arms are connected by a central 'hinge' domain. The two ATPase 'head' domains at the apices of this dimer are meanwhile inter-connected by a kleisin subunit, Scc1. Scc1's N- and C-terminal domains bind respectively to the coiled coil emerging from Smc3's head (its neck) and the base of Smc1's ATPase, thereby creating a tripartite SMC-kleisin (S-K) ring (*Figure 1A*). Cohesin's association with DNA as well as its abilities to hold sisters together and extrude DNA loops are facilitated by three large hook-shaped HAWK (HEAT repeat proteins associated with kleisins) proteins; Scc2, Scc3, and Pds5 (*Figure 1A*). Scc3 is thought to be permanently bound to the complex, whereas Scc2 and Pds5, which are mutually exclusive, are more dynamic. Of these, Scc2 has a crucial role in activating cohesin's ATPase at least in vitro, whether in the presence or absence of DNA (*Petela et al., 2018*).

The discovery that anaphase is initiated through the opening of S-K rings due to cleavage of their kleisin moiety by the protease separase (*Uhlmann, 2001*) led to the suggestion that cohesion is mediated by the co-entrapment of sister DNAs within individual S-K rings (*Haering et al., 2002*). This hypothesis, known as the ring model, made the key prediction that site-specific chemical cross-linking of all three of the ring's subunit interfaces would create a covalent topological linkage resistant to protein denaturation between small circular sister DNAs. Such catenated dimers (CDs) are indeed found in cells (*Haering et al., 2008*; *Gligoris et al., 2014*) and only under conditions in which cells form sister chromatid cohesion (*Srinivasan et al., 2018*).

The ring model envisages that once established during DNA replication, maintenance of sister chromatid cohesion during G2 and M phases would not require continued ATP hydrolysis. This notion, namely that cohesion is a passive process, explains why Scc2, though essential for loading and for maintaining cohesin's association with unreplicated DNA in vivo, has no role in maintaining cohesion during G2/M phases (*Ciosk et al., 2000*; *Srinivasan et al., 2019*). Cohesin's ATPase is strictly dependent on Scc2 in vitro and is presumably inactive in vivo upon Scc2's departure. LE in contrast requires continuous ATP hydrolysis dependent on Scc2, at least in vitro (*Davidson et al., 2019*).

Yet another difference is that cohesion depends on passage of DNAs inside S-K rings while LE does not (*Srinivasan et al., 2018*; *Davidson et al., 2019*). Given that cohesion and LE involve at least some different mechanisms, it is perhaps not surprising that there is increasing evidence that the two processes are mutually exclusive in vivo (*Srinivasan et al., 2018*; *Davidson et al., 2019*). Complexes engaged in cohesion do not extrude loops and vice versa.

Though maintenance of cohesion may have little in common with LE, the process by which cohesion is created in the first place may utilise mechanisms common to LE. This is supported by the fact that Scc2 is required for entrapping DNA within S-K rings as well as for the DNA-dependent ATPase activity necessary for LE. DNA entrapment assays combined with cryo-EM structures suggest that a key intermediate common to both processes is the passage of DNA between disengaged ATPase heads followed by its 'clamping' by Scc2 on a surface on top of them created by ATP-dependent head engagement (*Collier et al., 2020*; *Shi et al., 2020*; *Higashi et al., 2020*). It is envisaged that DNA translocation during LE involves recurrent rounds of DNA clamping followed by its release upon ATP hydrolysis. If so, each round presumably involves clamping of DNA successively further along the chromatin fibre. Clamping in this manner may be an important feature of cohesin's association with chromatin, at least during G1 when LE is possibly its main activity. Crucially, clamping in vitro does not require Scc3, which is necessary for cohesin's stable association with chromatin in vivo and ensures, at least in vitro, that clamping is followed or accompanied by transient opening of the S-K ring and thereby entrapment of DNAs within (*Collier et al., 2020*). The key point is that clamping may be a feature not only of LE but also of the entrapment of DNAs within S-K rings necessary for cohesion.

Which interface of the S-K ring is opened through the action of Scc3 is uncertain as is the mechanism, either when individual DNAs are entrapped during G1 (or G2) or when sister DNAs are entrapped during the passage of replication forks. Complexes containing co-translational fusions, either between the C-terminus of Smc3 and the NTD of Scc1, or between Scc1's C-terminus and the NTD of Smc1 are functional and capable of entrapping individual or sister DNAs within S-K rings. In contrast, the artificial connection of the Smc1 and Smc3 hinge domains using rapamycin blocks the establishment but not maintenance of sister chromatid cohesion (*Gruber et al., 2006*), leading to the suggestion that DNAs enter the S-K ring via a gate created by transient dissociation of the hinge. Whether this is really the case awaits more rigorous types of experiments.

Cohesin complexes defective in ATP hydrolysis, due to Smc1E1158Q and Smc3E1155Q (EQEQ) mutations, accumulate in the clamped state in vitro (*Collier et al., 2020*; *Shi et al., 2020*; *Higashi et al., 2020*). Along with Scc2, they also accumulate at *Saccharomyces cerevisiae CEN* sequences (*Hu et al., 2011*), which are sites at which cohesin loads onto chromosomes with especially high efficiency, due to an interaction between the kinetochore protein Ctf19 and Scc4 bound to Scc2's largely unstructured N-terminal domain (*Hinshaw et al., 2017*). This suggests that in addition to being a recurrent feature of LE, formation of the clamped state may be an early step in cohesin's de novo association with chromosomal DNA. Scc4 facilitates Scc2-mediated loading throughout chromosome arms as well as at *CEN* sequences, though how it does so is poorly understood.

When cohesin's ATPase heads are disengaged, the coiled coils of Smc1 and Smc3 associate with each other along much of their length (*Chapard et al., 2019*). When this 'zipping up' includes the sections of coiled coils close to the ATPase heads, it forces them to adopt a configuration in which they are juxtaposed in a 'J' state that is distinct from, and incompatible with ATP-driven head engagement known as the 'E' state. Crucially, the zipping up of coiled coils in this manner is incompatible with the clamping of DNA by Scc2 on top of engaged heads and the latter is therefore accompanied by extensive unzipping, at least up to the elbow (*Collier et al., 2020*). Coiled coil zipping up is a feature of cohesin engaged in holding sister chromatids together, with sister DNAs entrapped within J-K compartments, namely between juxtaposed (J) heads and the kleisin associated with them (*Chapard et al., 2019*). Extensive zipping up may have an important role in preventing unregulated ATP hydrolysis or precocious head engagement.

Along with coiled coil zipping up, the generation of cohesive structures during S phase is accompanied by acetylation of Smc3's K112 and K113 residues (*Guacci et al., 2015*; *Beckouët et al., 2016*). The double acetylation stabilises cohesin's association with chromosomes and increases the residence time of Pds5, which unlike Scc2 is necessary for maintaining cohesion as well as preventing de-acetylation of K112 and K113 (*Chan et al., 2012*; *Chan et al., 2013*). Complexes occupied by Pds5 cannot hydrolyse ATP, and in addition to maintaining cohesive structures in post-replicative cells, replacement of Scc2 or its human orthologue Nipbl by Pds5 appears to block the DNA translocation necessary for LE throughout interphase (*Petela et al., 2018*; *Wutz et al., 2017*; *Dauban et al., 2020*).

An important property of complexes occupied by Pds5, but not those by Scc2, is their ability to dissociate from chromosomes (*Chan et al., 2013*). This releasing activity is blocked by acetylation of Smc3 K112 and K113 during S phase by Eco1, substitution of both residues by glutamine, fusion of Scc1's NTD to Smc3's C-terminus (*Chan et al., 2012*), or mutations that affect the interface between Smc1 and Smc3 ATPase heads when engaged in the presence of ATP (*Elbatsh et al., 2016*). These findings have led to the suggestion that dissociation of Scc1's NTD from Smc3's neck during head engagement in the presence of Pds5 has a key role in triggering release (*Beckouët et al., 2016*). This process normally requires binding of Wapl to Pds5 and Scc3, along with head engagement (*Kueng et al., 2006*; *Muir et al., 2020*). Crucially, neither Wapl nor Pds5 are intrinsic to the release process as neither protein is necessary when Scc2 is inactivated in G1 cells, suggesting that the dissociation of Scc1 from Smc3 necessary for release takes place when heads engage in the absence of Scc2 and that Pds5 and Wapl facilitate the process at least partly by occluding Scc2 (*Srinivasan et al., 2019*). How Smc3's K112 and K113 residues contribute to release when unmodified is not understood. If release involved an intermediate similar to the clamped state, albeit with Pds5 replacing Scc2, then these residues could contribute to the binding of DNA to engaged heads.

As well as their tendency to zip up, a striking feature of cohesin's SMC coiled coils is their folding around an elbow (*Figure 1A*) situated two thirds of the way between the heads and hinge

(*Bürmann et al., 2019*). Folding around this discontinuity results in association of the hinge with a section of the coiled coil close to the so-called joint region, a break in the coiled-coils above the ATPase heads. Folding is a widely conserved feature of SMCs when observed using EM in vitro, both when heads are engaged (*Collier et al., 2020*) or disengaged (*Bürmann et al., 2019*), but whether folding occurs in vivo and has an important physiological function is not known. It has been postulated that the elbow could be involved in LE by coupling cycles of folding and unfolding with DNA translocation (*Bürmann et al., 2019*; *Hassler et al., 2018*). Further, it has been noted that a potential simultaneous interaction of a HAWK with the hinge and kleisin would require some sort of folding (*Murayama and Uhlmann, 2015*, *Huis in 't Veld et al., 2014*, *Bürmann et al., 2019*).

Despite the discovery that Scc2 facilitates binding of DNA to engaged ATPase heads in vitro (*Shi et al., 2020*; *Higashi et al., 2020*) and does so in the absence of Scc3 without entry inside the S-K ring (*Collier et al., 2020*), the mechanism by which Scc2 promotes cohesin's association with and translocation along chromosomes in vivo remains poorly understood. Scc2's unstructured NTD is bound by a superhelical array of 13 tetratricopeptide repeats (TPRs) belonging to its partner Scc4 (*Hinshaw et al., 2015*). Cells lacking either Scc4 or Scc2's NTD are not viable and have greatly reduced levels of chromosomal cohesin. Nevertheless, a version of Scc2 lacking the NTD is fully capable of activating cohesin's ATPase (*Petela et al., 2018*) and clamping DNA on top of engaged ATPase heads in vitro (*Shi et al., 2020*; *Collier et al., 2020*). To gain insight into the role of Scc4, we recently undertook a genetic screen, isolating mutations that suppress lethality caused by loss of Scc4 activity (*Petela et al., 2018*). This identified two different *scc2* point mutations, E822K and L937F. *scc2E822K* lies in the interface between Scc2 and Smc3's K112 and K113, and as such, all three residues are in the vicinity of DNA clamped by Scc2 on top of engaged ATPase heads. Because acetylation of Smc3 K112 K113 greatly reduces cohesin loading as well as release, Scc2E822K might bypass Scc4 by increasing the avidity with which DNA is clamped (*Collier et al., 2020*).

Here, we describe two other types of mutations that suppress *scc4* lethality. One type includes mutations in histone H2A that loosen the association between nucleosomes and DNA, which conceivably act like Scc2E822K, by facilitating cohesin's interaction with naked DNA. The other is an aspartic acid on the surface of Smc1's hinge domain that is replaced by an aromatic residue: *smc1D588Y*. UV-induced crosslinking in cells whose Smc1 hinge contains p-benzoyl L-phenylalanine (BPA) at defined positions revealed that it contacts Scc2, Scc3, and Pds5. Inactivation of Scc4 reduced crosslinking with Scc2, but increased that with Pds5, while *smc1D588Y* had the opposite effect. These findings suggest that Smc1's hinge contacts Scc2 and Pds5 directly, Scc4 facilitates association with Scc2 and hinders that with Pds5, and *smc1D588Y* does likewise and compensates for a lack of Scc4. To explain how Scc2 contacts Smc1's hinge while also bound to Smc1's ATPase, we suppose that cohesin's coiled coil is folded around its elbow, thereby bringing the hinge into contact with HAWK regulatory subunits associated with cohesin's ATPase heads, as recently observed in a cryo-EM structure of the ATP-bound clamped state (*Collier et al., 2020*).

Using cryo-EM, we have now determined the structures of the folded cohesin complex associated with either Scc2 or Pds5, both in the absence of ATP. The structures demonstrate that both Scc2 and Pds5, while attached to the ATPase domains of Smc1 and Smc3, respectively, reach up to the hinge, thus providing a clue regarding the effects of Smc1D588Y and an explanation for previously observed K620BPA crosslinks (*Bürmann et al., 2019*). The resolution of the folded coiled coils and hinge (5–6 Å) not only permitted the identification of the contacts involved in folding but also allowed identification of candidates for cysteine substitution for potential bismaleimidoethane (BMOE) crosslinking to assay folding. One such residue pair, Smc1R578C-Smc3V933C, gave rise to efficient BMOE-induced crosslinking in vivo even when Smc3 was acetylated. Therefore, folding takes place not only when Scc2 is bound but also when cohesin is engaged in holding sister chromatids together in post-replicative cells. Our findings demonstrate that folding of cohesin's coiled coils is not an in vitro artefact. Folding occurs in living cells, it is a feature of cohesin engaged in holding sister chromatids together, and it is of physiological importance during Scc2-mediated cohesin loading.

# Results

To understand better how Scc4 helps Scc2 to load cohesin onto chromosomes, we isolated mutations that enable temperature-sensitive *scc4-4* cells to grow at the restrictive temperature (35.5°C). This yielded both intragenic and extragenic mutations. The *scc4-4* allele was created by error-prone PCR (*Ciosk et al., 2000*) and contains several different mutations, including Y40N. Sequencing of intragenic revertants revealed that wild-type (WT) growth was restored either by restoring tyrosine at position 40 or by substituting it with histidine, implying that the mutation responsible for *scc4-4*'s thermosensitive proliferation is Y40N. When integrated at the *LEU2* locus, this mutation alone conferred temperature sensitive (ts) growth (*Figure 1—figure supplement 1A*). Y40 is a highly conserved residue that is buried in Scc4's superhelical array of TPR motifs (*Figure 1—figure supplement 1B*). It is unlikely that it contacts the Scc2 polypeptide directly, but despite this, Y40N disrupts co-immunoprecipitation of Scc4 and Scc2 (*Figure 1—figure supplement 1C*).

## A mutation in the hinge domain of Smc1 restores viability in the absence of Scc4

All of the extragenic *scc4-4* suppressors (*Figure 1B*) contained a mutation tightly linked to *SMC1*. Indeed, all 12 independently isolated mutations contained the same single base change causing substitution of aspartic acid by tyrosine at position 588 in Smc1 (*Figure 1E*). Tetrad dissection of *SCC4/scc4Δ SMC1/smc1D588Y* diploids revealed that *smc1D588Y* enabled cells to proliferate in the complete absence of Scc4 (*Figure 1C*). Smc1D588 is located in the hinge domain, at the C-terminal end of a β strand that interacts in an antiparallel fashion with a strand in Smc3 (*Figure 1D*). Despite its proximity to the Smc1-Smc3 interface, D588 does not appear to contact Smc3 residues. To address whether suppression arises due to the loss of a relatively conserved acidic residue (*Figure 1E*) or due to the substitution of a bulky aromatic, we tested the ability of a variety of other amino acid substitutions to rescue viability in the absence of Scc4. Mutant or WT alleles of *SMC1* were introduced into the *TRP1* locus of a *SMC1/smc1Δ SCC4/scc4Δ* diploid. Subsequent dissection revealed that mutation to phenylalanine or tryptophan was able to restore growth to a similar degree as tyrosine in the absence of Scc4 (*Figure 1—figure supplement 1D*) but not histidine, arginine, alanine, glutamic acid, or asparagine (*Supplementary file 1*). Suppression of *scc4Δ* lethality also occurred in the presence of WT *SMC1* but was much less effective (*Figure 1—figure supplement 1D*). Thus, we conclude that suppression is due to the introduction of a bulky aromatic amino acid at this crucial position and not through loss of the conserved aspartic acid. It is notable that the DNA base change observed in all 12 suppressors is the only one capable of creating such a transition via a single change and that the equivalent position is never a bulky aromatic in *SMC2*, *SMC3*, and *SMC4* as well as *SMC1*. *smc1D588Y* was able to rescue the proliferation defect of the temperature-sensitive *scc2-4* allele at 30°C, but not *scc2Δ* (*Figure 1—figure supplement 1E, F*), implying that it acts by enhancing the activity of Scc2, not by replacing it.

To determine whether the *smc1D588Y* mutation alters the hinge structure, we introduced the equivalent mutation (*D574Y*) into an isolated mouse hinge. X-ray crystallography revealed that a clash between the tyrosine residue and neighbouring loop causes Y574 to instead swing out relative to the position of D574, causing a local conformational change of the mutated loop (*Figure 1D, F*, *Figure 1—figure supplement 1G*, *Supplementary file 2*). Importantly, the change had little or no impact on the overall structure of the hinge. Despite this, both D588Y and D588W reduced the amount of Smc1 hinge that co-precipitated with Smc3 hinges (*Figure 1—figure supplement 1H*). To address whether Smc1D588Y affects dissociation of pre-assembled Smc1/3 hinge complexes, we co-expressed either WT Smc1 or Smc1D588Y hinge domains with Smc3 hinges, purified Smc1/3 complexes, and compared their persistence in the presence of a fivefold excess of SNAP-tagged Smc1 hinge domains. This revealed that the amount of Smc1D588Y associated with Smc3 hinges declined more rapidly in the presence of a WT competitor than WT Smc1, indicating that Smc1D588Y at least increases the off rate (*Figure 1—figure supplement 1I*).

Our finding that Smc1D588Y increases dissociation of Smc1 from Smc3 hinge domains in vitro raises the possibility that suppression depends on, or indeed is caused by, the greater ease with which hinges can dissociate. We therefore tested whether other non-lethal mutations within the Smc1/Smc3 hinge interface are also capable of bypassing the need for Scc4. A highly conserved lysine residue within Smc3 (Smc3K652) that opposes Smc1D588 was mutated to tyrosine, alanine, or

valine, with no effect. Similarly, previously published mutations in both hinges, designed to weaken their interaction, *smc1L635K K639E; smc1I590K; smc1L564K; smc3E570K; smc3L672R* (*Mishra et al., 2010*), were also unable to support growth in the absence of Scc4. The failure of these other mutations to suppress *scc4∆* lethality, together with our finding that *smc1D588Y* was identified in 12 out of 12 spontaneous extragenic suppressors, suggests that decreased affinity of Smc1/Smc3 hinges is not the mechanism by which Smc1D588Y enables Scc2 to load cohesin without Scc4.

## *smc1D588Y* restores cohesin occupancy on chromosome arms in the absence of Scc4

Calibrated ChIP-seq revealed that *scc4-4* causes a substantial reduction in the level of chromosomal cohesin when G1 cells undergo S phase and enter G2/M at the restrictive temperature (37°C) (*Figure 2A*, *Figure 2—figure supplement 1A*). The reduction is more marked within pericentric sequences, where there is a 10-fold reduction, than along arms where is there merely a fourfold reduction. Average chromosome profiles centred around the centromeric CDEIII, plotted as a percentage of the reads obtained for WT, revealed that *smc1D588Y* restores cohesin occupancy to approximately WT levels on chromosome arms (>30 kb from the centromere), but not around centromeres (*Figure 2A*). The failure to restore loading around centromeres is perhaps not surprising as most pericentric cohesin is loaded at *CEN*s in a process that involves binding of Scc4 to the kinetochore protein Ctf19, a requirement that is apparently not bypassed by *smc1D588Y*. Interestingly, *smc1D588Y* caused a substantial reduction of cohesin occupancy around centromeres even in the presence of WT *SCC4* (*Figure 2A*), an effect that will also have contributed to the lack of suppression in this region of the chromosome.

To investigate the effect of *smc1D588Y* at a more physiological temperature, we used calibrated ChIP-seq to compare cohesin's occupancy of the genome in *SCC4*, *SCC4 smc1D588Y,* and *scc4∆ smc1D588Y* cells following their release from a pheromone-induced G1 arrest and subsequent arrest in G2/M phase at 25°C. Average chromosome profiles around centromeres plotted as a percentage of WT (*SCC4*) revealed that *smc1D588Y* increased cohesin occupancy on chromosome arms to 120– 150% of WT levels, both in the presence and absence of *SCC4* (*Figure 2B*). In other words, *smc1D588Y* enhances cohesin's loading on chromosome arms via a mechanism that is completely independent of Scc4. In contrast, *smc1D588Y* reduced association around centromeres to approximately 60% of WT levels, which was further reduced to 20% by *scc4∆*. In *scc4∆ smc1D588Y* cells, cohesin occupancy within pericentric chromatin resembles that along chromosome arms as if a single Smc1D588Y-driven mechanism is responsible for loading at both locations in these cells (*Figure 2— figure supplement 1B*). Importantly, *smc1D588Y* does not increase occupancy on chromosome arms merely because defective loading at *CEN*s increases the amount of cohesin available to load onto chromosome arms because the *scc4m35* mutation, which disrupts Scc4's association with Ctf19 and also reduces loading at *CEN*s, has no such effect (*Petela et al., 2018*; *Hinshaw et al., 2015*).

It is striking that in *SCC4* cells *smc1D588Y* had far less effect on cohesin's association at *CEN* loading sites themselves. For example, it was 110% of WT in cells growing at 25°C and 75% of WT at 37°C. Because association was greatly reduced in *scc4-4* and *scc4∆* cells (*Figure 2A, B*), it presumably arises as a consequence of Scc4's association with Ctf19. If so, these complexes should be associated with Scc2, which was confirmed by calibrated ChIP-seq showing that Scc2's association with *CEN*s far from being reduced was in fact substantially increased by *smc1D588Y* and fully dependent on Scc4 (*Figure 2C*). Cohesin occupied by Scc2 at *CEN*s could either be in the process of loading (*Petela et al., 2018*; *Hu et al., 2011*) or engaged in LE (*Dauban et al., 2020*; *Paldi et al., 2020*). In both cases, the complexes are likely to adopt at least transiently the clamped state, which is stabilised by *smc1E1158Q* and *smc3E1155Q,* at least in vitro (*Collier et al., 2020*). In cells, cohesin complexes containing these mutations accumulate to especially high levels at *CEN*s, albeit with a short residence time (*Hu et al., 2011*), suggesting that they initiate an early step in the loading process, namely the clamped state, but in the absence of ATP hydrolysis fail to undergo a later step required for stable association and translocation into neighbouring pericentric sequences. Interestingly, *smc1D588Y* not only increased Scc2's association with *CEN*s but also caused a similar increase in Smc3E1155Q's association (*Figure 2D*). This implies that the reduced loading around centromeres arises not from defective formation of the clamped state at *CEN*s by Scc2/4 complexes associated with Ctf19 but from a defect in a subsequent step in the loading/translocation reaction that requires ATP hydrolysis. Because accumulation of Smc1D588Y complexes at *CEN*s resembles that of

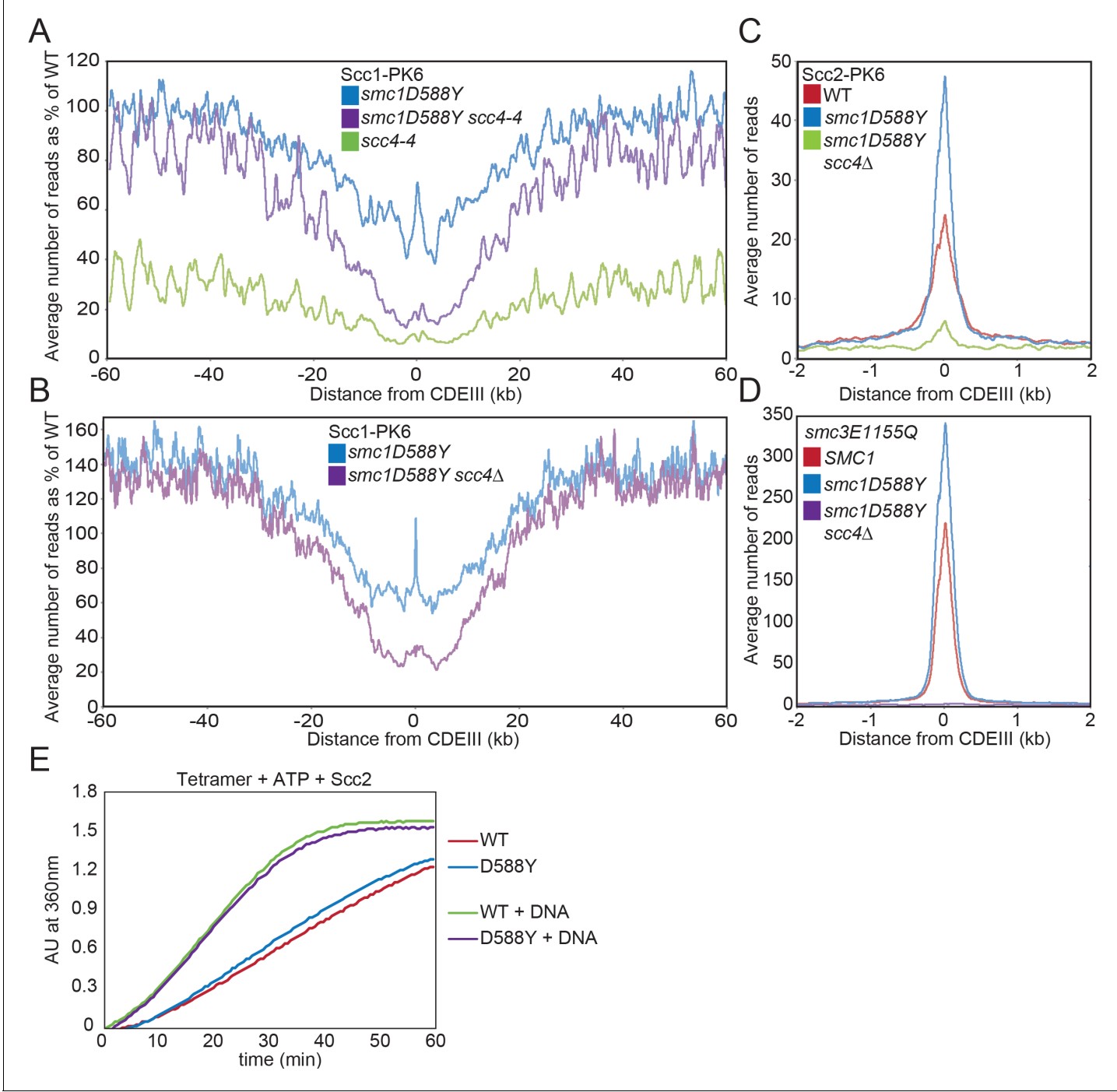

**Figure 2.** *smc1D588Y* restores cohesin occupancy on chromosome arms in the absence of Scc4. (**A**) Average calibrated ChIP-seq profiles of Scc1-PK6 in *smc1D588Y*, *scc4-4*, and *smc1D588Y scc4-4* cells 60 kb either side of *CDEIII* plotted as a percentage of the average number of reads obtained for wild-type (WT) cells. Cells were pheromone arrested in G1 at 25°C before release at 37°C into medium containing nocodazole. Samples were taken 75 min after release (K22005, K22009, K21999, K22001). (**B**) Average calibrated ChIP-seq profiles of Scc1-PK6 in *smc1D588Y*, and *smc1D588Y scc4Δ* cells 60 kb either side of *CDEIII* plotted as a percentage of the average number of reads obtained for WT cells. Cells were pheromone arrested in G1 at 25°C before release at 25°C into medium containing nocodazole. Samples were taken 60 min after release (K22005, K22009, K19624). (**C**) Average calibrated ChIP-seq profiles of Scc2-PK6 2 kb either side of *CDEIII* in cycling WT, *smc1D588Y*, and *smc1D588Y scc4Δ* cells at 25°C (K21388, K24680, K24678). (**D**) Average calibrated ChIP-seq profiles of ectopically expressed Smc3E1155Q-PK6 2 kb either side of *CDEIII* in cycling WT, *smc1D588Y*, and *smc1D588Y scc4Δ* cells at 25°C (K24562, K24689, K24564). (**E**) ATPase activity of WT or mutant tetramers on addition of ATP and Scc2 in the presence and absence of DNA.

*Figure 2 continued on next page*

*Figure 2 continued*

The online version of this article includes the following figure supplement(s) for figure 2:

**Figure supplement 1.** *smc1D588Y* restores cohesin occupancy on chromosome arms in the absence of Scc4.

---

complexes containing Smc3E1155Q, we tested the effect of Smc1D588Y on cohesin's ATPase activity but found little or no effect either in the presence or absence of DNA (*Figure 2E*, *Figure 2—figure supplement 1D*).

## Mutations in *SCC2* and histone genes also suppress *scc4Δ* lethality

To address whether it is possible to identify extragenic *scc4Δ* suppressor mutations besides *smc1D588Y*, we isolated a second set in a *smc1D588E* yeast strain that cannot mutate residue 588 to an aromatic residue through a single base pair mutation (*Petela et al., 2018*). We identified using genetic crosses and genomic sequencing 12 mutations within *SCC2* (described in *Petela et al., 2018*), 49 within *HTA1* (one of two histone H2As), and a single mutation within *HTB1* (one of two H2Bs). All permitted proliferation of *scc4Δ* cells, albeit to a greater or lesser extent (*Figure 3A*).

## Scc4 helps overcome inhibition of loading by nucleosomes

The H2A mutations affected three residues, namely G30, R31, and R34. These mutations (G30D, R31I/T/S/G, and R34I) are all located on a defined patch on the surface of the nucleosome that interacts with DNA and the single H2B mutation (Y44D) is located nearby (*Figure 3B*). Because substitution of two positively charged residues causes suppression, we surmise that the mutations act by weakening the association between histones and DNA. *hta1R31I* was made de novo and shown to suppress the lethality of *scc4-4* cells (*Figure 3A*). Its effect on cohesin loading in *SCC4* and *scc4-4* cells was measured using calibrated ChIP-seq to measure Scc1's association with the genome after cells had undergone DNA replication at 35.5°C following a pheromone-induced G1 arrest at 25°C. A lower restrictive temperature (35.5°C) was used in this instance because *hta1R31I* is itself lethal at 37°C. Consistent with its poor suppression of *scc4Δ* lethality (*Figure 3A*), *hta1R31I* increased loading along chromosome arms more modestly than *smc1D588Y*, raising loading in *scc4-4* cells from 20% to 70% of WT (*HTA1 SCC4*) (*Figure 3C*). As in the case of both *smc1D588Y* and *scc2E822K L937F* (*Petela et al., 2018*), *hta1R31I* failed to suppress the loading defect of *scc4-4* mutants in the vicinity of centromeres (*Figure 3C*). Interestingly, in the presence of WT *SCC4*, *hta1R31I* actually increased loading along chromosome arms over WT by 20%. This implies that the association between histones and DNA within the nucleosome restricts cohesin loading, at least along chromosome arms, not only in *scc4* mutants but also in WT cells. Like *scc2E822K L937F* (*Petela et al., 2018*) but unlike *smc1D588Y*, *hta1R31I* does not per se reduce loading of cohesin around centromeres (*Figure 3C*), suggesting that *hta1R31I* and *smc1D588Y* affect different aspects of the loading process.

It has been suggested that the chromatin structure remodelling complex (RSC) has a key role in loading cohesin onto yeast chromosomes (*Huang et al., 2004*) and that an important function of Scc2/4 along chromosome arms is to facilitate nucleosome remodelling catalysed by RSC (*Lopez-Serra et al., 2014*). This raised the possibility that mutations like *hta1R31I* suppress the loading defects of *scc4* mutants because they bypass the need for RSC and *smc1D588Y* might act likewise. To address this, we used calibrated ChIP-seq to reinvestigate the loading defects of *sth1-3* cells, which contain a temperature sensitive mutation within RSC's ATPase subunit. WT and *sth1-3* cells were arrested in G1 by α-factor at 25°C and then released from the block at 37°C. *sth1-3* delayed budding, DNA replication, and the onset of Smc3 acetylation (*Figure 3—figure supplement 1A–C*), complicating the comparison with WT. We therefore compared the calibrated ChIP-seq profiles *sth1-3* cells 105 min after release, when most but not all cells had both budded and undergone DNA replication, with WT cells at 75 min, a time point at which their cell cycle progression was most similar. Western blotting confirmed that the levels of Scc1 in this pair of samples were also similar (*Figure 3—figure supplement 1C*). Surprisingly, their calibrated ChIP-seq profiles were also very similar not only in the vicinity of centromeres (*Figure 1—figure supplement 1D, F*) but also throughout an interval 60 kb either side of centromeres (*Figure 3—figure supplement 1F*). Crucially, *sth1-3* caused only a modest reduction in the occupancy ratio (OR), which denotes the overall level of association throughout the genome (*Figure 3—figure supplement 1D*). These findings contradict the previous

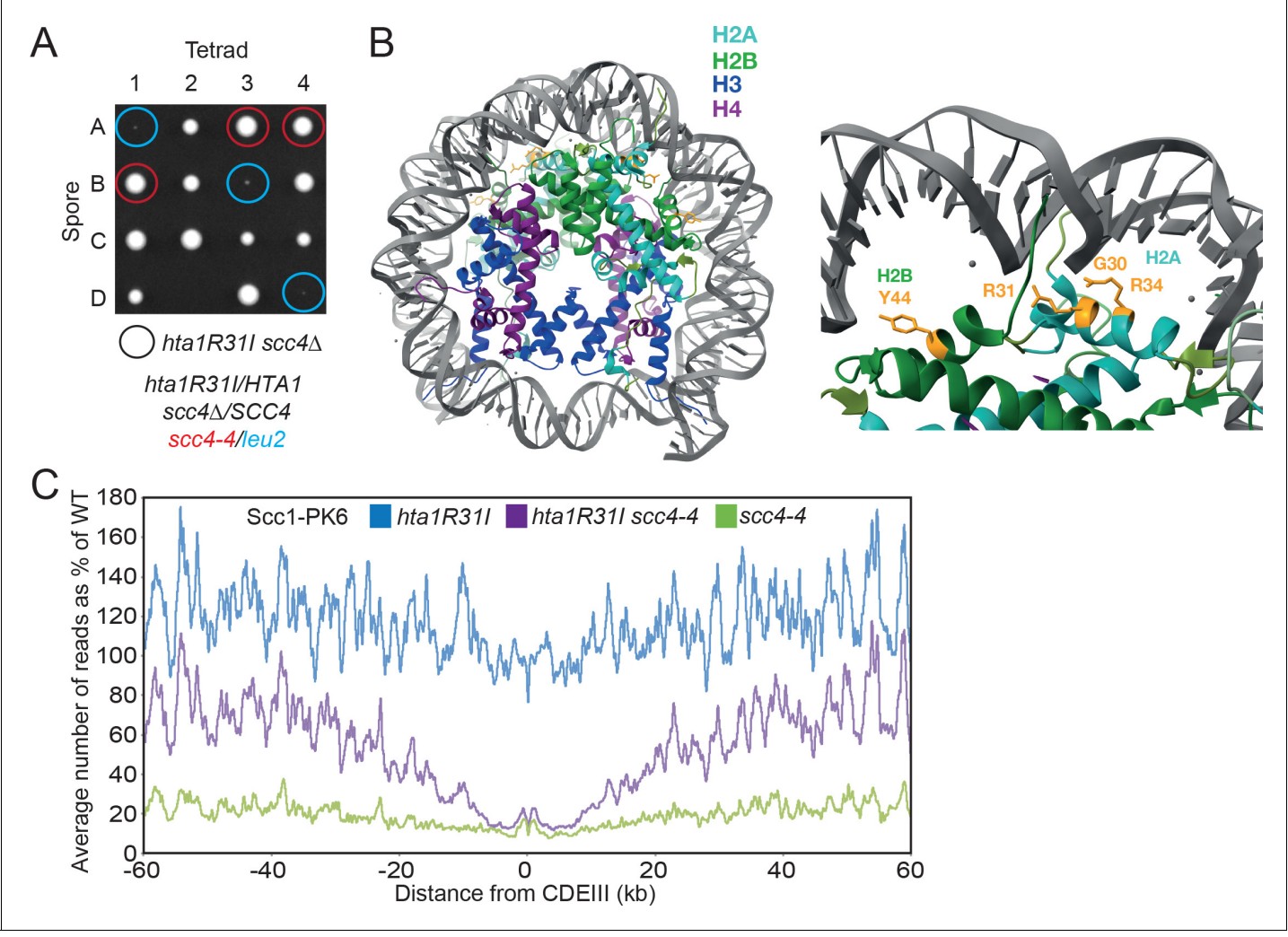

**Figure 3.** Mutations in *SCC2* and histone genes also suppress *scc4Δ* lethality. (**A**) Tetrad dissection of diploid strains containing *SCC4/scc4Δ leu2/scc4-4 HTA1/hta1R31I*. Spores in which *scc4Δ* is rescued by *hta1R31I* are circled in blue. (**B**) Structure of the yeast nucleosome (PDB: 1ID3; *White et al., 2001*). H2A is shown in blue and H2B in green. Suppressor mutations are shown in yellow. (**C**) Average calibrated ChIP-seq profiles of Scc1-PK6 in *hta1R31I*, *scc4-4*, and *hta1R31I scc4-4* cells 60 kb either side of *CDEIII* plotted as a percentage of the average number of reads obtained for wild-type (W)T cells. Cells were pheromone arrested in G1 at 25°C before release at 35.5°C into medium containing nocodazole. Samples were taken 60 min after release (K22005, K24574, K24568, K22001).

The online version of this article includes the following figure supplement(s) for figure 3:

**Figure supplement 1.** Scc4 helps overcome inhibition of loading by nucleosomes.

---

claim that RSC has a crucial role in cohesin loading, based on qPCR measurements at individual loci of the very same *sth1-3* strain (*Lopez-Serra et al., 2014*). Because of this discrepancy, we compared the calibrated ChIP-seq profiles of WT (75 min) and *sth1-3* (105 min) in the vicinity of three loci whose association was previously reported to be 30% of WT. There was little or no effect of the mutation at *CEN3*, a modest reduction at *POA1*, and more surprisingly an increase at *MET10* (*Figure 3—figure supplement 1E*).

Given the pleiotropic consequences of *sth1-3* on cell cycle progression, it is difficult to exclude the possibility that RSC has a modest effect on cohesin loading. However, if Scc4 promoted loading by helping chromatin remodelling by RSC, then *smc1D588Y* should suppress any apparent loading defect caused by RSC. The fact that the Scc1 calibrated ChIP-seq profile of *smc1D588Y sth1-3* double mutants is indistinguishable to that of *sth1-3* single mutants (*Figure 3—figure supplement 1G*) shows that insofar that there is any defect, it is clearly unaffected by *smc1D588Y*. In other words, a

version of cohesin that no longer requires Scc4 does not alter *sth1-3*'s albeit modest defect. It may therefore be a pleiotropic consequence of the mutant's retarded cell cycle progression and not due to an Scc4-dependent RSC activity that creates nucleosome-free regions necessary for cohesin loading.

Recent work has revealed that Scc2 has a key role in clamping DNA onto engaged heads and that Scc2E822K, which also suppresses *scc4Δ*, might function by enhancing DNA binding within the clamped state (*Collier et al., 2020*; *Shi et al., 2020*; *Higashi et al., 2020*). We therefore suggest that the reason why histone mutations suppress the lethality of *scc4* mutants is because they increase the accessibility of DNA and thereby facilitate formation of the clamped state.

## Scc4 regulates an interaction between the hinge domain and HAWKs

How might replacement of a specific surface residue on the Smc1 hinge by a bulky aromatic one help Scc2 function without Scc4? One possibility is that it strengthens a hydrophobic interaction with another cohesin subunit. We have previously described the UV-dependent crosslinking in living yeast cells between Pds5 and a version of Smc1 containing BPA at position K620, which is located in an alpha helix adjacent to the loop containing D588 (*Figure 4A*; *Bürmann et al., 2019*). Pds5 is not required for cohesin loading, and therefore strengthening its interaction with Smc1's hinge cannot be responsible for suppression. We therefore tested whether UV induces crosslinking of Smc1K620BPA to other regulatory subunits. To do this, cells expressing FLAG-tagged versions of Scc2, Scc3, Scc4, or Pds5 in cells whose sole source of Smc1 was Myc-tagged Smc1K620BPA were exposed to UV, and subsequent western blotting was used to detect FLAG-tagged proteins in immunoprecipitates (IPs) of Scc1-containing complexes (*Figure 4B*).

Western blotting for the Myc epitope confirmed that all samples contained a high molecular weight version of Smc1, consisting of proteins crosslinked to K620 (*Figure 4B*). As expected for a subunit that is stably associated with cohesin Smc-kleisin trimers, high levels of Scc3 were detected in IPs from Scc3-FLAG cells, most of which had an electrophoretic mobility expected of uncrosslinked protein, but a small fraction co-migrated with the high molecular weight version of Smc1, suggesting that UV also induces crosslinking of Smc1K620BPA to Scc3. Pds5 is less stably associated, explaining why only modest amounts of uncrosslinked Pds5 are detected in the IPs. Despite this, we observed much more Smc1-Pds5 than Smc1-Scc3 crosslinked protein, confirming that Smc1K620BPA crosslinks to Pds5 with high efficiency (*Bürmann et al., 2019*). Because co-precipitation of unstably associated proteins will be greatly enhanced by crosslinking, it is not possible to assess the actual fraction of crosslinked protein. Scc2's residence time on chromosomal cohesin of approximately 2–4 s (*Hu et al., 2011*) is even less than that of Pds5 and the former is therefore difficult to detect in Smc1 IPs. Nevertheless, the level of Smc1-Scc2 crosslinked protein was comparable to that of Scc3, despite being overall threefold less abundant (*Tóth et al., 1999*). In contrast, we detected no Smc1-Scc4 crosslinked proteins in Scc4FLAG cells. Cryo-EM has revealed that the N-terminal HEAT repeats of Scc2 as well as those of its human ortholog Nipbl are found in close proximity to Smc1's hinge within complexes that have clamped DNA on top of their engaged ATPase domains (*Shi et al., 2020*; *Higashi et al., 2020*; *Collier et al., 2020*) and the crosslinking between Smc1K620BPA and Scc2 may reflect this state. However, they could also reflect an alternative one in which Scc2 is bound to cohesin whose heads are disengaged as described in the next section.

As association of cohesin with Scc2 and Pds5 is mutually exclusive and the latter incapable of activating cohesin's ATPase or association with chromatin (*Petela et al., 2018*), Scc4 and Smc1D588Y could facilitate loading either by enhancing association of the hinge with Scc2 or decreasing it with Pds5. To test this, we measured the effect of *scc4-4* in the presence or absence of *smc1D588Y* on Smc1K620BPA-Pds5 and Smc1K620BPA-Scc2 crosslinking. This revealed that Scc4 inactivation (*scc4-4*) increased Pds5 crosslinking threefold while *smc1D588Y* had the opposite effect. Crucially, *smc1D588Y* was largely epistatic to *scc4-4*. In other words, the elevated crosslinking observed when Scc4 was inactivated dropped in the presence of *smc1D588Y* to the depressed level of *SCC4 smc1D588Y* cells (*Figure 4C, D*). The mutations had the opposite, albeit less dramatic, effects on Smc1K620BPA-Scc2 crosslinking. Scc4 inactivation halved it while *smc1D588Y* restored it 80% of WT levels (*Figure 4E, F*). These results are consistent with the notion that a key function of Scc4 is to facilitate interaction between Scc2 and the Smc1 hinge, either directly or indirectly by impeding the latter's interaction with Scc2's competitor Pds5 or conceivably via both mechanisms.

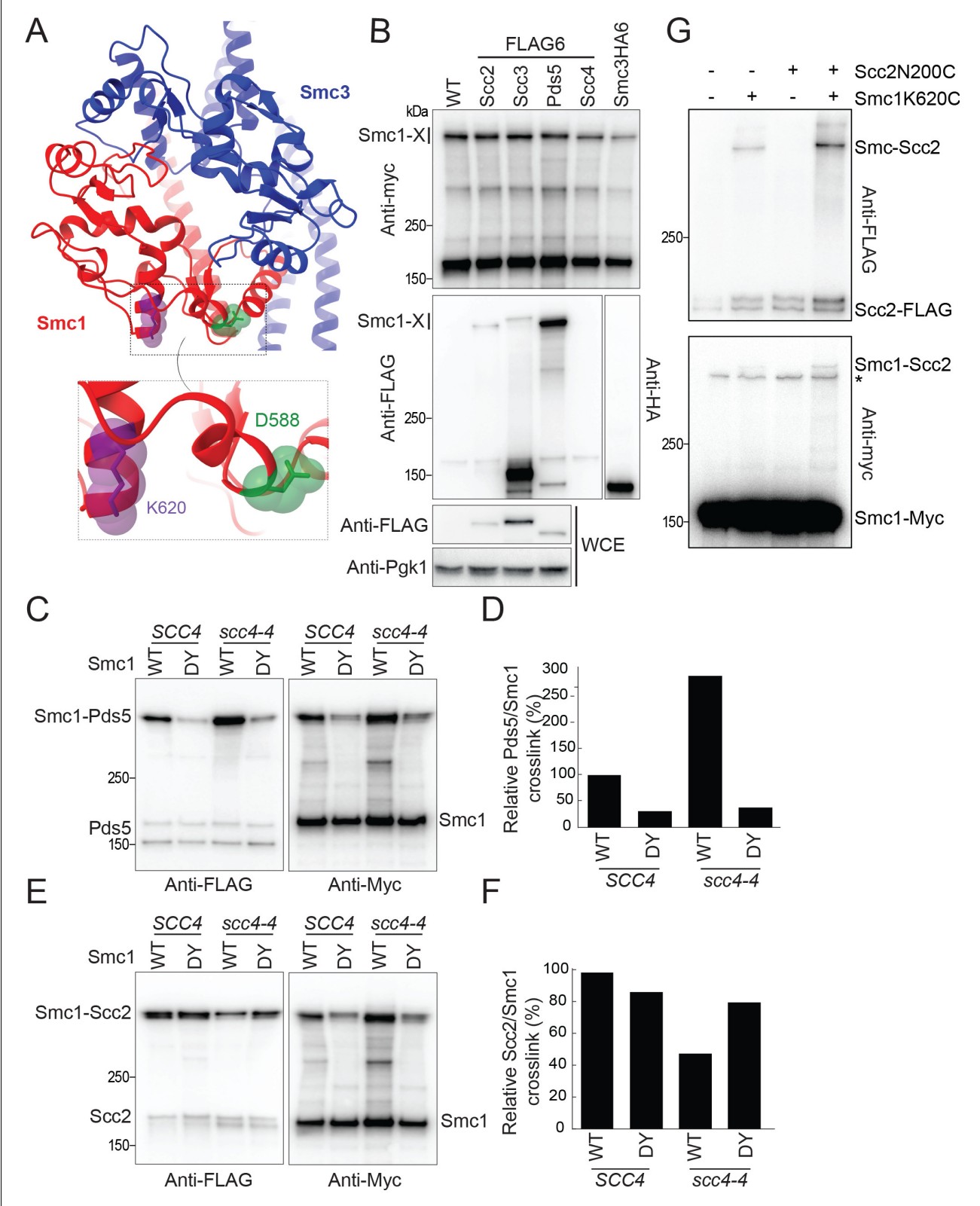

**Figure 4.** Scc4 regulates an interaction between the hinge domain and HAWKs. (**A**) Modelled structure of the yeast cohesin hinge domain based on bacterial SMC hinge from *Thermotoga maritima* (PDB: 1GXL; *Haering et al., 2002*). (**B**) Identification of proteins that crosslink to Smc1 hinge. Strains expressing various cohesin regulators tagged with either FLAG6 or HA6 in combination with Smc1K620BPA-myc were treated with UV prior to immunoprecipitation with PK-tagged Scc1 and the products analysed by western blotting (B1969, B1976, B1983, B2020, B2072, B2079). (**C**) Effect of

*Figure 4 continued on next page*

*Figure 4 continued*

Scc4 and Smc1D588Y on crosslinking between Pds5 and Smc1 hinge. Cells expressing Smc1K620BPA in the presence or absence of *scc4-4* and Smc1D588Y were exponentially grown at 25°C and shifted to 35.5°C for 1 hr. Cells were irradiated with UV, and the cohesin complex was isolated by immunoprecipitation of PK-tagged Scc1. The Myc-tagged Smc1K620BPA was examined by western blot (B2072, B2212, B2214, B2215). (D) Quantification of the crosslinks in (C) as a percentage of the wild-type (WT) Smc1 crosslinking efficiency. (E) Effect of Scc4 and Smc1D588Y on crosslinking between Scc2 and Smc1 hinge. Strains were treated as described in (C) (B1969, B2213, B2216, B2217). (F) Quantification of the crosslinks in (E) as a percentage of the WT Smc1 crosslinking efficiency. The experiments shown in (C–F) were performed twice with the same result. (G) In vivo cysteine crosslinking of Smc1 hinge with Scc2 protein. Yeast cells expressing Smc1K620C and Scc2N200C were incubated with bismaleimidoethane (BMOE) (B3082, B3107, B3114, and B3116). The crosslinked Smc1/Scc2 was isolated by immunoprecipitation of PK-tagged Scc1 and examined by western blot. * Unspecific crosslink band.

The online version of this article includes the following figure supplement(s) for figure 4:

**Figure supplement 1.** Scc4 regulates an interaction between the hinge domain and HAWKs.

If the essential role of Scc4 were merely to hinder an interaction between the Smc1 hinge and Pds5, then Scc4 should be unnecessary for cohesin's association with chromosome arms in cells lacking Pds5. We therefore used calibrated ChIP-seq to measure the effect of depleting Pds5 (using the auxin-dependent AID degron) on cohesin's occupancy of chromosome arms after *scc4-4* cells undergo S phase at 37°C, which revealed that Pds5 depletion had no effect (*Figure 4—figure supplement 1A*). In other words, Pds5 is not necessary for depressing cohesin's association with chromosomes in *scc4-4* mutants. Likewise, if by reducing Pds5's interaction with the Smc1 hinge *smc1D588Y* reduced Pds5's occupancy of chromosomal cohesin, then it should depress the fraction of chromosomal cohesin associated with Pds5. The fact that *smc1D588Y* has no such effect (*Figure 1—figure supplement 1B, C*) implies that though the mutation alters how Pds5 interacts with the Smc1 hinge, this does not in fact alter chromosomal cohesin's occupancy by Pds5.

Our finding that Scc4 does not act solely by hindering Pds5 suggests that Scc4 and Smc1D588Y facilitate Scc2 activity by promoting its interaction with the hinge. To elucidate where the hinge contacts Scc2, we inserted TEV protease cleavage sites at various positions within Scc2 to determine whether Smc1K620BPA crosslinked to the N- or C-terminal fragments created by TEV cleavage (*Figure 1—figure supplement 1D, E*). Analysis of those TEV insertions that were functional in vivo revealed that crosslinking occurred within Scc2's N-terminal sequences, between residues 150 and 215. This interval is between the N-terminal domain that binds Scc4 (*Hinshaw et al., 2015*) and the hook-shaped structure composed of HEAT repeats. This part of Scc2 is not sufficiently ordered to have been visualised in the cryo-EM structure of a complex containing DNA clamped between Scc2 and engaged Smc1/3 ATPases (*Collier et al., 2020*). To confirm the location, we measured BMOE-induced crosslinking in vivo between Smc1K620C and a variety of Scc2 cysteine substitutions between residues 153 and 212. Although Smc1K620C alone gave rise to a Smc1-Scc2 crosslinked species, the crosslinking was more efficient on the introduction of Scc2N200C, suggesting that Smc1 is likely also crosslinking to a natural cysteine in Scc2 (most likely Scc2C224, which sits on a small helix just below N200) (*Figure 4G*). Importantly, the region of Scc2 whose association with the Smc1 hinge is reduced by *scc4-4* and restored by *smc1D588Y* is close to where Scc4 binds to Scc2 (*Hinshaw et al., 2015*). In other words, Scc4 would be close enough to directly influence Scc2's interaction with the hinge.

## Cryo-EM structures of cohesin trimers associated with Scc2 or Pds5 reveal folded coiled coils

The notion that *smc1D588Y* suppresses *scc4Δ* by altering the interaction between Smc1's hinge domain and cohesin's HAWK subunits Scc2 and Pds5 raises a conundrum: how can HAWK proteins, which are known to associate with cohesin's kleisin subunit and its ATPase domains, interact with a hinge domain that is separated from the ATPase domains by a 50-nm-long coiled coil? One possibility is that the HAWK proteins interact with cohesin's hinge and ATPase domains at different points in time. Alternatively, if in fact they interact with hinge and heads simultaneously, then the coiled coil cannot be fully extended. For example, folding at an elbow in the middle of the coiled coil (*Bürmann et al., 2019*) may bring the hinge into proximity of HAWKs associated with the ATPases. Folding has recently been observed at low resolution in a complex between DNA, Scc2, and hydrolysis-impaired EQEQ ATPases engaged in the presence of ATP (*Collier et al., 2020*; *Shi et al., 2020*;

*Higashi et al., 2020*). To investigate this further, we used cryo-EM to determine the structures of the *S. cerevisiae* cohesin trimer (Smc1, Smc3, and Scc1 containing cysteines specifically crosslinking the three intermolecular interfaces; Smc1-Scc1, Smc3-Scc1, and the Smc1-Smc3 hinge [*Collier et al., 2020*] at an efficiency of 20% [data not shown]) bound to either Scc2 (*Figure 5A*) (EMD-12880) or Pds5 (*Figure 6A*) (EMD-12888) in the absence of nucleotide and DNA. The former revealed a coiled

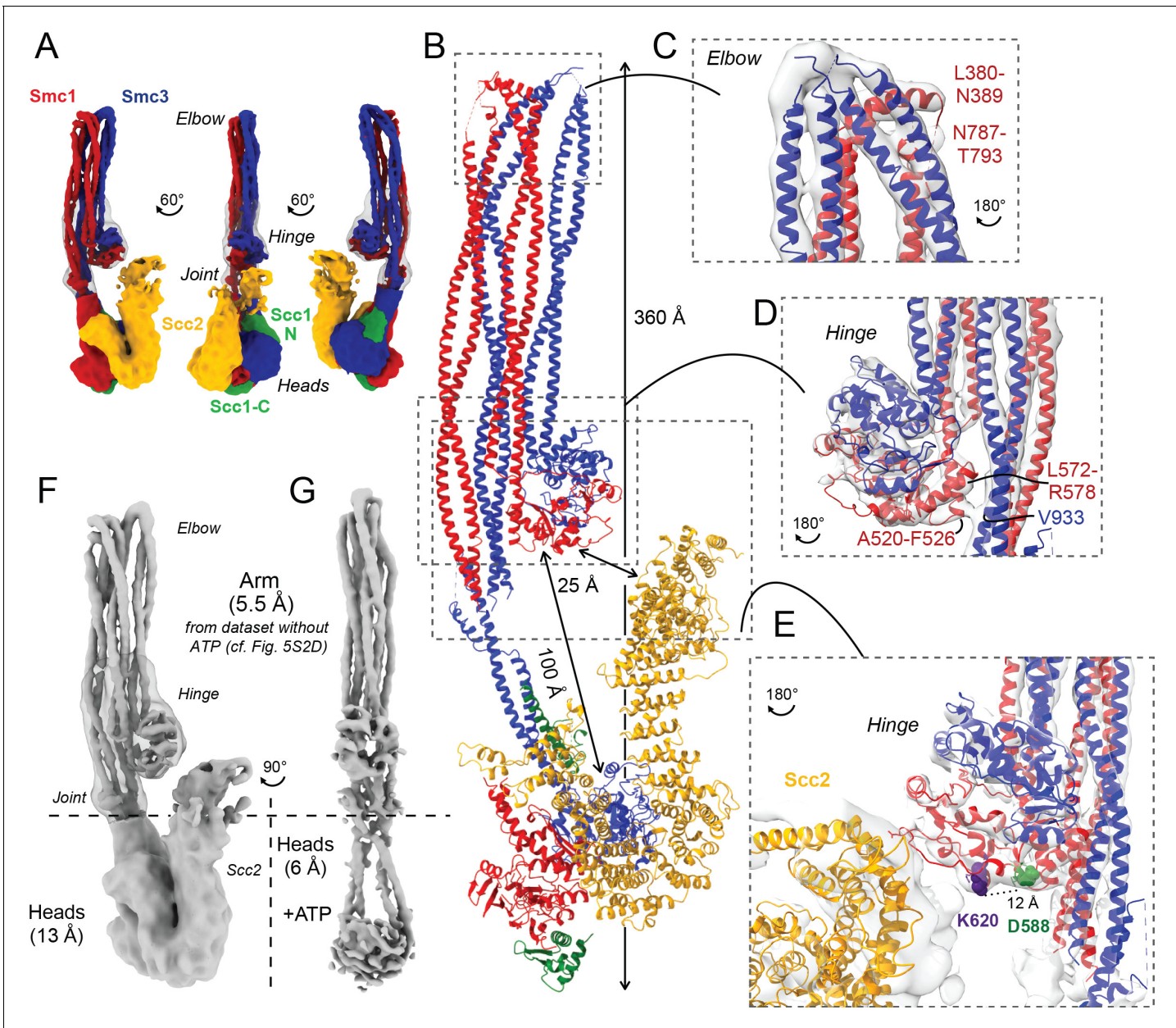

**Figure 5.** Folded cohesin allows interaction of hinge with Scc2 N-terminus. (**A**) Views of cryo-EM reconstruction of Scc2-bound cohesin coloured by subunit. (**B**) Full pseudo-atomic model of folded cohesin trimer bound to Scc2. (**C**) Close-up of breaks in the coiled coils of Smc3 and Smc1 that constitute the elbow region of cohesin (PDB: 7OGT; EMD-12887). (**D**) Close-up of the interaction between the hinge and Smc3 that stabilises the folded state. (**E**) Close-up of Scc2 N-terminus in proximity of hinge residues K620 and D588Y. (**F, G**) Comparison of cryo-EM densities between Scc2-bound and ATP-free cohesin seen in (**F**) (EMD-12880) and ATP-bound cohesin seen in (**G**) (EMD-12889), demonstrating that head engagement is not sufficient for coiled coil unzipping.

The online version of this article includes the following figure supplement(s) for figure 5:

**Figure supplement 1.** Folded cohesin allows interaction of hinge with Scc2 N-terminus.

**Figure supplement 2.** Data processing and reconstruction schematics of all cryo-EM maps.

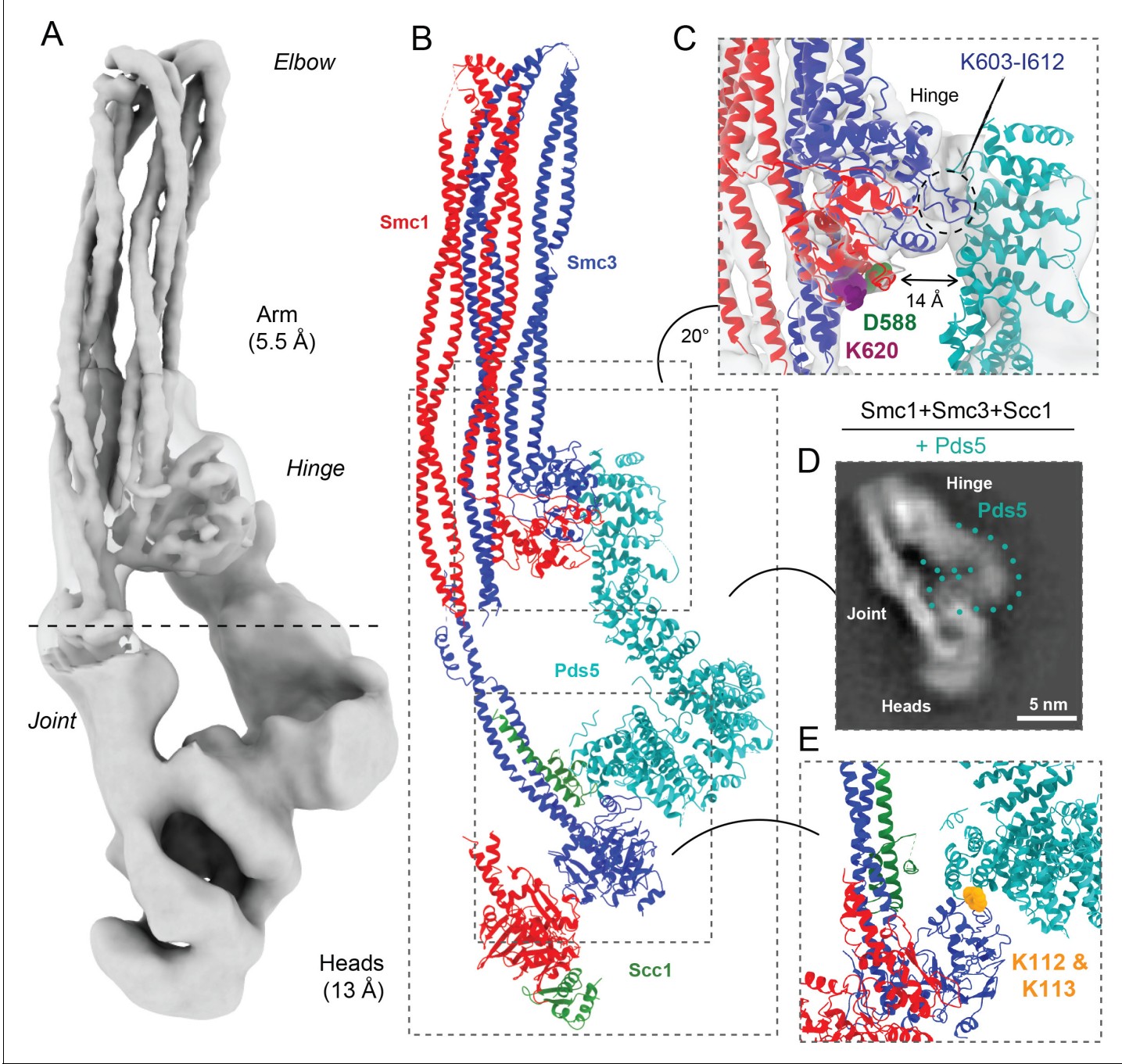

**Figure 6.** Pds5 binds to Smc3 head while contacting the hinge. (**A**) Composite map of cryo-EM reconstructions of Pds5-bound cohesin (EMD-12888). (**B**) Full pseudo-atomic model of folded cohesin trimer bound to Pds5 coloured by subunit. (**C**) Close-up of interaction between hinge and Pds5 showing proximity of N-terminus of the HAWK to hinge residues D588 and K620. (**D**) 2D classes of Pds5-bound ATPase heads. (**E**) Close-up of Pds5 binding to K112- and K113-proximal region of the Smc3 head.

The online version of this article includes the following figure supplement(s) for figure 6:

**Figure supplement 1.** Detailed view of fitted atomic structures in cryo-EM maps.

coil folded at its elbow (*Figure 5B, C*), causing the hinge to interact with sections of the coiled coil that are approximately 10 nm away from the point at which they emerge from the ATPase domains (*Figure 5B*). The cryo-EM reconstruction not only revealed the path of the coiled coils around the hinge-coiled coil interface (where the map is at 5–6 Å resolution; EMD-12887), but also enabled the

production of a pseudo-atomic model of the folded form (PDB: 7OGT). Folding brings a pair of helices within Smc1's hinge, namely the end of the coiled coil around A520-F526 and another short helix around L564-R578, into close proximity of a short stretch of Smc3's coiled coil (*Figure 5D*). Very similar folding was observed when Pds5 was bound instead of Scc2 (*Figure 6A, B*). Though folding permits an association between the hinge and the N-terminal Scc2 sequences, which could in principle stabilise the folded conformation, we also observed similar, if not identical, folding in samples lacking all HAWK proteins (*Figure 5—figure supplement 1E*).

## Coiled coil folding enables interaction of Scc2 or Pds5 with the Smc1 hinge

Initial 2D classes of Scc2-bound cohesin revealed floppiness not only within the HAWK, especially within its C-terminus, but also between the joint and the ATPase heads (*Figure 5—figure supplement 1B*). We therefore split the complex computationally into two regions, with a boundary at the joint, and processed their densities separately (*Figure 5F*). This yielded an overall resolution of 13 Å for the HAWK-bound part, which enabled fitting of a homologous Scc2 crystal structure (PDB: 5ME3; *Chao et al., 2015*) together with both head crystal structures (PDB: 1W1W; *Haering et al., 2004*; PDB: 4UX3; *Gligoris et al., 2014*) to produce a pseudo-atomic model. Analysis revealed that Scc2 binds rigidly to the Smc1 ATPase head in a manner resembling but distinct from its interaction in the clamped state (*Figure 5—figure supplement 1B*; *Collier et al., 2020*). Scc2's C-terminal HEAT repeats 18–24 (residues 1127–1493) dock onto Smc1's F-loop (residues 1095–1118) as well as the emerging coiled coils, a mode of interaction analogous to that between condensin's HAWK Ycs4 and Smc4 (*Lee et al., 2020*). This mode of interaction therefore takes place whether or not heads are engaged. Unlike the engaged and clamped head state, Scc2 makes no contact with Smc3 in the non-engaged, nucleotide-free structure. Contrary to its C-terminal part, Scc2's N-terminal region adopts a range of conformations. Bending around the mid-region of Scc2 enables its N-terminus to contact the joint region of Smc3's coiled coil in the clamped state. However, when heads are disengaged in our nucleotide-free structure, Scc2 is straightened and its N-terminal half adopts the conformation observed in crystals of Scc2 alone (*Chao et al., 2015*). Because cohesin's elbow is further away from its hinge than is the case for condensin, folding of its coiled coils brings the hinge to within 12 nm of the ATPase heads. As a consequence, the N-terminal part of Scc2 molecules bound to Smc1 ATPase heads is in proximity to the hinge, thereby explaining not only its crosslinking to Smc1K620BPA in vivo but also how Smc1D588Y could circumvent the need for Scc4 (*Figure 5E*). We suggest that the addition of a bulky amino acid into Smc1 through D588Y may be sufficient to help bind an otherwise floppy Scc2 N-terminal domain, whose interaction with the hinge is normally stabilised by Scc4.

We processed data collected on Pds5-bound complexes in a similar manner, producing a 13 Å resolution structure, which revealed that Pds5 binds to Smc3 and not, like Scc2, to Smc1's ATPase head domain (placed PDB: 5F0O; *Lee et al., 2016*; *Figure 6A, B*). The contact takes place between the most C-terminal HEAT repeats of Pds5 and the top region of the N-terminal lobe of Smc3's ATPase. Strikingly, this part of Smc3 contains the pair of highly conserved lysine residues K112 and K113 (*Figure 6E*), whose acetylation by Eco1 not only prevents releasing activity (*Unal et al., 2008*; *Rolef Ben-Shahar et al., 2008*) but also stabilises Pds5's interaction with chromosomal cohesin complexes (*Chan et al., 2012*). Unlike Scc2, Pds5 does not rely on negatively charged amino acids for its interaction with the K112/K113 region and may therefore be better suited than Scc2 for binding the acetylated and less positively charged version of Smc3. Furthermore, binding in this manner shields both lysine residues when acetylated, hence explaining how Pds5 hinders de-acetylation during G2/M phases (*Chan et al., 2013*). Like Scc2, Pds5's N-terminal HEAT repeats approach Smc1's hinge domain, which explains the crosslinking to Smc1K620BPA in vivo (*Figure 6C*). The low resolution and flexibility apparent in our map mean that we cannot be sure whether Pds5's C-terminal domain reaches beyond the hinge and contacts the coiled coils. Importantly, the modes of interaction of Scc2 and Pds5 with Smc subunits appear to be incompatible with each other, as has been postulated previously through in vivo and in vitro work (*Petela et al., 2018*).

## Head engagement does not per se drive unzipping of cohesin's coiled coil

A major difference between the apo state bound to Scc2 and the ATP-bound clamped state is the conformation of the Smc coiled coils. Though both folded, they are zipped up in the case of the former but splayed open up to the elbow in the case of the latter. Opening up could be driven by engagement per se. Alternatively, it might additionally require the binding of DNA to engaged heads in the presence of Scc2. In the course of our studies, we identified and solved with a resolution of 6 Å a form of cohesin lacking Scc2, Scc1, DNA, or crosslinker, whose ATPase heads were engaged in the presence of ATP (*Figure 5G*; EMD-12889). Contrary to previous studies with shortened constructs (*Muir et al., 2020*), which suggested that engagement per se might drive coiled coil unzipping, the coiled coils of our engaged heads are fully zipped up, at least from their joints to their hinge domains (*Figure 5G*). Thus, head engagement does not per se cause unzipping. Furthermore, 2D classes of heads-engaged cohesin bound to Scc2 demonstrate that addition of Scc2 to an ATP-bound state is insufficient to promote unzipping (*Figure 5—figure supplement 1A*). We therefore suggest that it is the binding of DNA to the surface on top of engaged heads that causes unzipping to make space for the DNA double helix, as well as the rearrangement of Scc2's NTD necessary for its association with Smc3's coiled coil.

## Folding of cohesin's coiled coils occurs in vivo and is a feature of sister chromatid cohesion

Though the interaction of Scc2 and Pds5 with Smc1's hinge in vivo is fully consistent with coiled coil folding and vice versa, it does not prove that folding actually occurs in vivo. To address this, we identified regions of both Smc3 and Smc1 whose residues when substituted by cysteine should permit crosslinking by BMOE specifically if Smc1's hinge interacted with Smc3's coiled coil in the manner observed in our cryo-EM structure (*Figure 7A*). A pair of residues, Smc1R578C and Smc3V933C, were viable both as single and double mutants, and gave rise to efficient BMOE-dependent crosslinking between Smc1 and Smc3 in vivo only when combined (*Figure 7B*). Because the efficiency of crosslinking was 60% or even more, we conclude that a high fraction of cohesin complexes must be folded at the elbow in vivo.

If, as seems likely, Smc1D588Y bypasses the need for Scc4 by strengthening the interaction between the hinge and Scc2, then folding would appear to be a feature of cohesin complexes engaged in loading in vivo and the observation by cryo-EM that folding is a feature of the clamped state in vitro confirms this. To address whether folding is also a feature of cohesin complexes engaged in holding sister chromatids together, when the cohesin's ATPase is thought to be inactive, we measured whether acetylated Smc3 molecules were also efficiently crosslinked to Smc1 using the cysteine pair that reports folding. Western blots using an antibody specific for acetylated Smc3 revealed that crosslinking between Smc3V933C and Smc1R578C was similar to that between a hinge cysteine pair (*Haering et al., 2008*). Because a large fraction of acetylated Smc3 was crosslinked to Smc1 in both cases (*Figure 7C*), we conclude that folding is a feature of many, if not most, cohesin complexes engaged in holding sister chromatids together. As expected, the Smc3Ac antibody failed to detect any protein in G1-arrested cells (*Figure 7C*), confirming its specificity.

## Discussion

### Folding occurs in vivo and is of functional importance

A major feature of all Smc-kleisin complexes, be they bacterial homodimers or eukaryotic heterodimers, are the 50-nm-long coiled coils connecting their hinge dimerisation domains to their ATPase heads. Recent structural and biochemical studies have revealed that the two coiled coils of Smc dimers have a strong tendency to self-associate or zip up throughout their length both in vitro and in vivo (*Bürmann et al., 2019*; *Chapard et al., 2019*; *Diebold-Durand et al., 2017*; *Soh et al., 2015*). In many cases, for example, MukBEF from *Escherichia coli* as well as the eukaryotic cohesin and condensin complexes, zipping up is accompanied by folding around an elbow, which leads to an association of hinges with sections of the coiled coil closer to the heads. Cryo-EM imaging suggests that complete zipping up may be an invariant property of apo-complexes. In the case of cohesin, clamping of DNA by Scc2 on top of engaged ATPase heads is accompanied by extensive

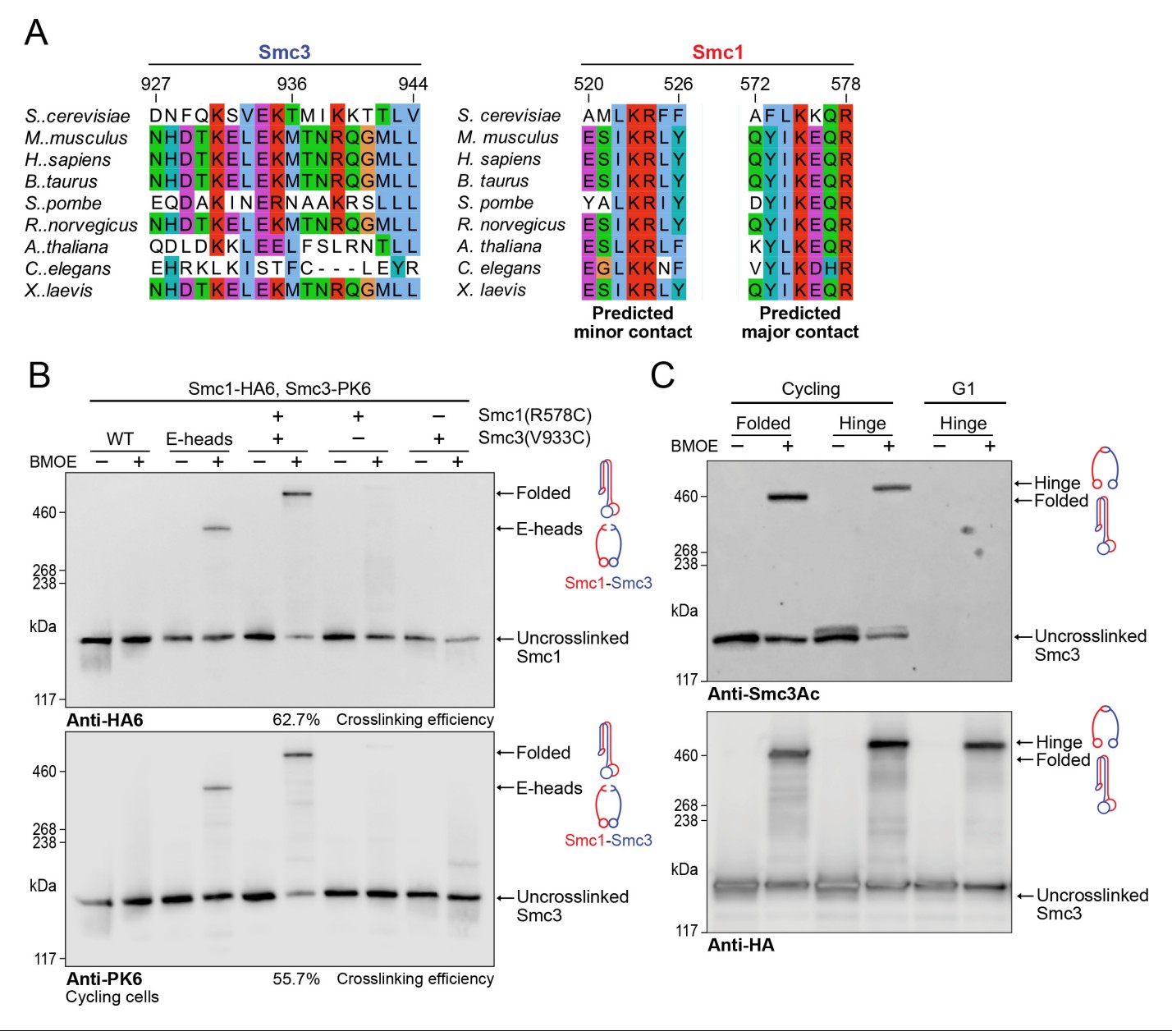

**Figure 7.** Folding of cohesin's coiled coils occurs in vivo and is a feature of sister chromatid cohesion. (**A**) Sequence conservation analysis for the Smc3 coiled coil and Smc1 hinge helices shown in *Figure 4D* shows that the residues are highly conserved. (**B**) Whole-cell extract western blot analysis for the crosslink between Smc1R578C-HA6 and Smc3V933C-PK6 with single cysteine controls probing for hemagglutinin (HA) (top) and PK (bottom). A band shift is observed at the same molecular weight for both blots, confirming the identity of the crosslinked species. Crosslinking of the engaged heads (*Chapard et al., 2019*) was used as a positive control (K28401, K27359, K28585, K28546, K28583). (**C**) Western blot analysis of crosslinking measuring the folded state (Smc1R578C-HA6 and Smc3V933C-PK6) and Smc1-Smc3 hinge dimerisation (*Haering et al., 2008*) probing for acetylated Smc3 (top) and HA (bottom) in logarithmic or pheromone arrested cells (K26081, K28586).

unzipping (*Collier et al., 2020*). The finding reported here (*Figure 5G*), that a cohesin complex whose heads are engaged in the presence of ATP nevertheless possesses coiled coils that are extensively zipped up, suggests that unzipping is not caused by head engagement per se but instead by the binding of DNA to engaged heads. Indeed, crosslinking studies have confirmed that the coiled coils associated with engaged heads are at least sometimes zipped up even in vivo (*Chapard et al., 2019*). Whether DNA clamping causes unfolding as well as unzipping is a matter of considerable interest because it has been speculated that the folding and unfolding of Smc coiled coils might be

a crucial aspect of ATP-driven mechanical cycle responsible for the translocation of Smc-kleisin complexes along DNA during LE (*Hassler et al., 2018*; *Bürmann et al., 2019*).

The observation that a large fraction of clamped complexes possess folded but extensively unzipped coiled coils suggests that unfolding is not a hard-wired response to clamping. This finding, along with the finding that some Smc coiled coils, for example, those in *Bacillus subtilis*, likely do not possess an elbow (*Bürmann et al., 2019*), the discovery that the hinges of cohesin and condensin occupy different positions, and the fact that folding has not hitherto been demonstrated in living cells all raised the possibility that the phenomenon might not in fact have an important physiological role. The work described here provides the first concrete evidence to the contrary. A 5.5 Å structure of cohesin's coiled coils in a folded state enabled us to identify a pair of residues within the Smc1 hinge/Smc3 coiled coil interface, namely Smc1R578 and Smc3V933, whose substitution by cysteine led to efficient crosslinking by BMOE inside cells, demonstrating that folding takes place frequently in vivo. Because Smc1R578C/Smc3V933C crosslinking occurs when Smc3 is acetylated, folding would also appear to be a feature of complexes engaged in holding sister chromatids together. Unlike mammalian cells, yeast lack sororin and acetylation is strictly linked to replication and it is therefore a good marker for complexes associated with cohesion. Future crosslinking studies combining Smc1R578C/Smc3V933C with other cysteine pairs should make it possible to address whether folding is also a feature of other cohesin states in vivo, for example, while DNAs are clamped between Scc2 and engaged heads when cohesin loads onto and translocates along chromatin fibres.

That folding not only occurs but is of physiological importance stemmed from a very different approach; the isolation of extragenic mutations that suppress the loading defect and lethality caused by *scc4* mutations. In addition to *scc2* alleles (e.g. E822K) and mutations that affect the way histones H2A and H2B bind to nucleosomal DNA, substitution by tyrosine (or any other aromatic residue for that matter) of a conserved aspartate residue on the surface of Smc1's hinge domain (Smc1D588Y) restored loading in *scc4* mutants to levels that were in fact 30% higher than WT. Because *smc1D588Y* cannot suppress loss of Scc2 itself, the suppressor acts by enabling Scc2 to function without its auxiliary subunit and not by bypassing Scc2 entirely.

In vivo crosslinking using cohesin complexes in which a surface lysine residue nearby D588 within Smc1's hinge was replaced by the non-canonical amino acid BPA (Smc1K620BPA) demonstrated that the hinge must be in proximity, at least some of the time, to Scc2, Scc3, and Pds5. Crucially, Scc4 facilitated crosslinking between Smc1K620BPA and Scc2 but inhibited that with Pds5, while *smc1D588Y* had a similar effect and compensated for loss of Scc4. To explain how Scc2 and Pds5, which bind to the Smc heads (*Figure 5A* and *Figure 6A*), are close enough to Smc1's hinge domain for BPA-mediated crosslinking, we suggest that cohesin's coiled coil must be folded not merely when cohesin holds sister chromatids together (*Figure 7*) but also during the loading reaction. This also explains how *smc1D588Y* exerts such a powerful effect on cohesin loading. Because Smc1K620BPA crosslinks to N-terminal sequences within Scc2 that are close to where Scc4 normally binds, we suggest that Scc4 helps Scc2 promote loading by facilitating the latter's interaction with the Smc1 hinge, and that Scc4's role can be substituted by insertion of an aromatic residue within the Smc1-Scc2 interface. How mechanistically Scc4 or Smc1D588Y facilitate interaction between the hinge and Scc2 while hindering that with Pds5 is presently unclear. The key point is that our genetic data confirm that the proximity between the N-terminal domain of Scc2 and Smc1 hinges observed in cryo-EM structures whether in the clamped (*Collier et al., 2020*; *Shi et al., 2020*) or ATP-free-state (*Figure 5A*) also occurs in vivo and more importantly has functional significance.

Though folding enables the hinge to interact with cohesin's HAWK proteins, it is likely that the process has functions besides such interactions as folding appears to be more conserved than the HAWKS themselves. It has been suggested that an extension/folding cycle might have a role in cohesin's translocation along DNA during LE (*Bürmann et al., 2019*). Another possibility is that by packing coiled coils on top of each other, folding helps to stabilise the zipping up of Smc1/3 coiled coils, which may have a role in ensuring that unzipping does not occur precociously, in other words, only when DNA is correctly clamped on top of engaged heads by Scc2. The notion that folding acts primarily to reinforce the zipped-up state helps explain why Smc proteins in organisms like *B. subtilis* do not appear to have an elbow around which their coiled coils are folded. Zipping up of *B. subtilis* Smc coiled coils might be strong enough that it does not need to be reinforced by folding. A third possibility is that, by bringing the hinge close to DNA clamped by Scc2 on top of engaged Smc1/3

ATPase heads, folding facilitates passage of DNA through a gate created by hinge opening, thereby mediating entrapment of DNA within S-K rings.

## What is the function of Scc4?

In addition to recruiting the Scc2/4 complex to kinetochores and thereby promoting high rates of cohesin loading at *CEN*s, Scc4 helps Scc2 promote cohesin's efficient association with chromosome arms. It has previously been suggested that Scc4's function involves the nucleosome remodelling complex RSC. Two rather different types of proposal have been made in separate papers by the Uhlmann group. According to the first, Scc2's association with Scc4 enables RSC to create nucleosome-free regions necessary for cohesin loading; in other words, Scc2/4's role is to positively regulate RSC's cohesin loading activity (*Lopez-Serra et al., 2014*). This proposal is difficult to reconcile not only with our finding that RSC inactivation causes only modest, if any, defect in cohesin loading as measured by calibrated ChIP-seq but also with the now incontrovertible evidence that the Scc2/4 complex acts directly on cohesin, enabling it to clamp DNA on top of its Smc ATPase domains. Neither our ChIP-seq data nor recent structural work (*Collier et al., 2020*; *Shi et al., 2020*; *Higashi et al., 2020*) support the notion that that 'Scc2/4 acts in sister chromatid cohesion by maintaining nucleosome-free regions' (*Lopez-Serra et al., 2014*).

The second proposal shares with the first the notion that nucleosome-free regions created by RSC are necessary for cohesin loading but differs from the first in suggesting that by interacting with Scc4, RSC recruits Scc2 to regions that have been cleared of nucleosomes and permits Scc2 to catalyse their association with cohesin (*Muñoz et al., 2019*). Thus, instead of Scc2/4 activating RSC and thereby creating nucleosome-free regions needed for cohesin loading, RSC is envisioned to activate Scc2/4 by bringing it to nucleosome regions created by and associated with RSC. Deletion of Scc2's NTD to which Scc4 binds is normally lethal, but fusion to RSC's Sth1 subunit of a version of Scc2 (Scc2C) lacking its N-terminal Scc4-binding domain restores viability. Though this finding is consistent with the notion that Scc4's function is merely to link Scc2 with RSC, the viability of Sth1-Scc2C fusions could equally well be explained if the artificially efficient recruitment of Scc2 to nucleosome-free regions (associated with RSC) enables Scc2 and cohesin to clamp DNA without the help of Scc4. Crucially, the notion that Scc4's function is to connect Scc2 with RSC fails to explain why RSC inactivation causes only a modest, if any, defect in cohesin loading, and certainly not one comparable to that caused by Scc4 inactivation, nor how a hinge mutation (Smc1D588Y) that alters cohesin's association with its HAWK regulatory subunits Scc2 and Pds5 is sufficient to bypass Scc4. One would have to suppose that Scc2/4's recruitment to RSC involved not merely the latter's association with Scc4 but also with cohesin, and Smc1D588Y acts by facilitating the interaction. It also provides no explanation for why mutations within Scc2 (E822K) that probably alter how the latter forms the clamped state also fully bypass Scc4.

Our finding that the major partners of the part of the hinge containing Smc1D588 are in fact cohesin's HAWKs, principally Pds5 and Scc2, favours an alternative explanation, namely that Scc4 facilitates an interaction between the Scc2/4 complex and cohesin's hinge that either stabilises Scc2's association with cohesin and/or alters its conformation in a manner that enhances its ability to clamp DNA. It is nevertheless striking that the lethality of *scc4* mutants can be bypassed, albeit less effectively than with *smc1D588Y*, by mutations in histones H2A and H2B that presumably reduce the affinity of their interaction with nucleosomal DNA and would therefore favour formation of nucleosome-free DNA. We suggest that nucleosome-free DNA is indeed important for loading because naked DNA is necessary for formation of the clamped state. However, creation of naked DNA is insufficient for efficient clamping in vivo. Association of Scc2 with cohesin's hinge and/or a conformational change that is a consequence of this association is additionally required, a process normally facilitated by Scc4 but whose absence can be compensated by Smc1D588Y. In other words, association of Scc2/4 with cohesin's hinge domain facilitates the binding of naked DNA to its engaged ATPase heads and it is ultimately the latter that promotes a productive association with chromatin. Our data do not exclude the possibility that RSC is in principle able to create the nucleosome-free DNA necessary for clamping, but whether it is normally necessary is at present unclear.

## Why does *smc1D588Y* depress loading at *CEN*s?

It is striking that while *smc1D588Y* facilitates cohesin's association with chromosome arms, in the presence as well as the absence of Scc4, the mutation has the opposite effect on loading at *CEN*s (*Figure 2*). To explain this paradox, we suggest that formation of the clamped state, which we propose is facilitated by *smc1D588Y*, is just the first step in cohesin's productive association with and translocation along chromatin fibres and must be followed by a second step, likely involving ATP hydrolysis and head disengagement. Consistent with the notion that D588Y accelerates the first clamping step, we observed that despite lowering the overall level of pericentric cohesin arising from loading at *CEN*s, *smc1D588Y* actually increased the amount of Scc2 associated with cohesin at *CEN*s. The fact that it also increased the amount of cohesin containing Smc3E1155Q confirms that this population represents the clamped state. Because such complexes do not load productively (*Hu et al., 2011*), hydrolysis of ATP associated with clamped complexes must also be required for loading. In other words, a sequence of clamping, DNA loading, and unclamping while the DNA remains loaded, is necessary. If so, an important question is whether clamping or subsequent unclamping driven by ATP hydrolysis is rate-limiting during the loading process. We suggest that clamping is rate-limiting along chromosome arms but unclamping is rate-limiting at *CEN*s and that this is the reason why *smc1D588Y* enhances arm loading while depressing loading at *CEN*s.

In summary, we provide biochemical evidence that cohesin's coiled coils are indeed folded in vivo and genetic evidence that folding is of physiological importance in loading cohesin onto chromosomes. Whether folding is regulated by cohesin's ATPase cycle and cyclical unfolding has a role in DNA LE are important questions for the future.

# Materials and methods

### Key resources table

| Reagent type (species) or resource | Designation | Source or reference | Identifiers | Additional information |
|---|---|---|---|---|
| Strain, strain background (*Spodoptera frugiperda*) | Sf9 insect cells | Thermo Fisher | Cat# 11496015 | N/A |
| Genetic reagent (*Saccharomyces cerevisiae*) | NCBITaxon:4932 | This paper | Yeast strains | *Supplementary file 4* |
| Biological sample | α-factor peptide | CRUK Peptide Synthesis Service | N/A | N/A |
| Antibody | Mouse monoclonal Anti-V5 | Bio-Rad | Cat# MCA1360 | (1:1000) |
| Antibody | Anti-HA High Affinity (3F10) (Rat) | Roche | Cat# 11867423001 | (1:1000) |
| Antibody | Anti-His (mouse) | GenScript | Cat# A00186 | (1:1000) |
| Antibody | Anti-c-Myc A-14 (9E10) (rabbit) | Santa Cruz Biotech | Cat# sc-789 | (1:1000) |
| Antibody | Anti-Myc 4A6 (mouse) | Millipore | Cat# 05-724 | (1:1000) |
| Antibody | Anti-FLAG (rabbit) | Sigma | Cat# F7425 | (1:1000) |
| Recombinant DNA reagent | pACEbac1 2xStrepII-Scc2151-1493 | *Collier et al., 2020* | N/A | N/A |
| Recombinant DNA reagent | pACEbac1 Smc1-8xHis-Smc3/pIDC Scc1-2xStrepII (trimer) | *Petela et al., 2018* | N/A | N/A |
| Recombinant DNA reagent | pIDS Pds5-Flag | *Petela et al., 2018* | N/A | N/A |
| Commercial assay or kit | Talon Superflow Metal Affinity Resin | Takara Bio. | Cat# 635669 | N/A |
| Commercial assay or kit | NuPAGE 3–8% Tris-Acetate Protein gels | Thermo Fisher | Cat# EA0378BOX | N/A |

*Continued on next page*

*Continued*

| Reagent type (species) or resource | Designation | Source or reference | Identifiers | Additional information |
|---|---|---|---|---|
| Commercial assay or kit | Trans-Blot Turbo Midi 0.2 μm Nitrocellulose Transfer Packs | Bio-Rad | Cat# 1704159 | N/A |
| Commercial assay or kit | Protein G Dynabeads | Thermo Fisher | Cat# 300385 | N/A |
| Commercial assay or kit | ChIP DNA Clean and Concentrator kit | Zymo Research | Cat# D5205 | N/A |
| Commercial assay or kit | NEBNext Fast DNA Library Prep Set for Ion Torrent | NEB | Cat# Z648094 | N/A |
| Commercial assay or kit | Ion Xpress Barcode Adaptors | Thermo Fisher | Cat# 4471250 | N/A |
| Commercial assay or kit | E-Gel SizeSelect II 2% Agarose gels | Thermo Fisher | Cat# G661012 | N/A |
| Commercial assay or kit | KAPA Ion Torrent DNA standards | Roche | Cat# 07960395001 | N/A |
| Commercial assay or kit | EnzChek phosphate assay kit | Thermo Fisher | Cat# E6646 | N/A |
| Commercial assay or kit | StrepTrap HP | Fisher Scientific | Cat# 11540654 | N/A |
| Commercial assay or kit | Superose 6 Increase 10/300 GL | VWR | Cat# 29-0915-96 | N/A |
| Chemical compound | Nocodazole | Sigma | Cat# M1404 | N/A |
| Chemical compound | Bismaleimidoethane (BMOE) | Thermo Fisher | Cat# 22323 | 5 mM |
| Chemical compound | Complete EDTA-free protease inhibitor cocktail | Roche | Cat# 4693132001 | (1:50 mL) |
| Chemical compound | PMSF | Sigma | Cat# 03115836001 | 1 mM |
| Chemical compound | Immobilon Western ECL | Millipore | Cat# WBLKS0500 | N/A |
| Chemical compound | RNase A | Roche | Cat# 10109169001 | N/A |
| Chemical compound | Proteinase K | Roche | Cat# 03115836001 | N/A |
| Chemical compound | BPA | Bachem | Cat# 4017646.0005 | N/A |
| Chemical compound | TCEP | Thermo Fisher | Cat# 20490 | N/A |
| Chemical compound | Desthiobiotin | Fisher Scientific | Cat# 12753064 | N/A |
| Software, algorithm | FastQC | Babraham Bioinformatics | https://www.bioinformatics.babraham.ac.uk/projects/fastqc/ | N/A |
| Software, algorithm | Fastx_trimmer | Hannon Lab | http://hannonlab.cshl.edu/fastx_toolkit/index.html | N/A |
| Software, algorithm | FilterFastq.py | *Petela et al., 2018* | https://github.com/naomipetela/nasmythlab-ngs | N/A |
| Software, algorithm | Bowtie2 | *Langmead and Salzberg, 2012* | http://bowtie-bio.sourceforge.net/bowtie2/index.shtml | N/A |
| Software, algorithm | Samtools | Samtools | http://www.htslib.org | N/A |
| Software, algorithm | IGB browser | *Nicol et al., 2009* | https://www.bioviz.org | N/A |
| Software, algorithm | chr_position.py | *Petela et al., 2018* | https://github.com/naomipetela/nasmythlab-ngs | N/A |
| Software, algorithm | filter.py | *Petela et al., 2018* | https://github.com/naomipetela/nasmythlab-ngs | N/A |
| Software, algorithm | Bcftools call | Samtools | http://www.htslib.org | N/A |

*Continued on next page*

*Continued*

| Reagent type (species) or resource | Designation | Source or reference | Identifiers | Additional information |
|---|---|---|---|---|
| Software, algorithm | MutationFinder.py | *Petela et al., 2018* | https://github.com/naomipetela/nasmythlab-ngs | N/A |
| Software, algorithm | yeastmine.py | *Petela et al., 2018* | https://github.com/naomipetela/nasmythlab-ngs | N/A |
| Software, algorithm | RELION 3.1 | doi:10.1016/j.jsb.2012.09.006 | N/A | N/A |
| Software, algorithm | CtfFind4 | doi:10.1016/j.jsb.2015.08.008 | N/A | N/A |
| Software, algorithm | CrYOLO 1.5 | doi:10.1038/s42003-019-0437 | N/A | N/A |
| Software, algorithm | Chimera | https://www.cgl.ucsf.edu/chimera/ | N/A | N/A |
| Software, algorithm | ChimeraX 1.0 | https://www.cgl.ucsf.edu/chimera/ | N/A | N/A |
| Software, algorithm | COOT | doi:10.1107/S0907444910007493 | N/A | N/A |
| Software, algorithm | MAIN | doi:10.1107/S0907444913008408 | N/A | N/A |
| Software, algorithm | Phenix.real_space_refinement | doi:10.1107/S2059798318006551 | N/A | N/A |
| Software, algorithm | PYMOL 2 | https://pymol.org/2/ | N/A | N/A |
| Software, algorithm | SWISS-MODEL | https://swissmodel.expasy.org | N/A | N/A |
| Other | Quantifoil R 2/2 grid: Cu/Rh 200 cryoEM grids | Quantifoil GmbH | N/A | N/A |

**Table of structures**

| Map description and file name in 'coordinates and maps' | First appearance in figures | Database accession code |
|---|---|---|
| Elbow EM map | *Figure 5F* | EMD-12887 |
| Elbow coordinate map | *Figure 5B* | PDB ID 7OGT |
| Scc2 bound to ATPase heads | *Figure 5F* | EMD-12880 |
| Scc2 bound to ATPase heads masking hinge and N-terminus | *Fig 5—figure supplement 1C* | |
| Pds5 bound to ATPase heads | *Figure 6A* | EMD-12888 |
| Engaged ATPase heads | *Figure 5G* | EMD-12889 |
| Mouse hinge D574Y | *Figure 1D* | PDB ID 7DG5 |

## Yeast strains and growth conditions

All yeast strains were derived from W303 and grown in rich medium (YEP) supplemented with 2% glucose (YPD) at 25°C unless otherwise stated. Cultures were agitated at 200 rpm (Multitron Standard, Infors HT). Strain numbers and relevant genotypes of the strains used are listed in the Key resources table, *Supplementary file 4*. To arrest the cells in G1, α-factor was added to a final concentration of 2 mg/L/h, every 30 min for 2.5 hr. Release was achieved by filtration wherein cells were captured on 1.2 µm filtration paper (Whatman GE Healthcare), washed with 1 L YPD, and resuspended in the appropriate fresh media. To arrest the cells in G2, nocodazole (Sigma) was added to the fresh media to a final concentration of 10 µg/mL and cells were incubated until the synchronisation was achieved (>95% large-budded cells). To inactivate temperature-sensitive alleles, fresh media were pre-warmed prior to filtration (Aquatron, Infors HT). To produce cells deficient in Pds5 using the AID system, cells were arrested with α-factor as described above. 30 min prior to release, auxin was added to 5 mM final concentration. Cells were then filtered as described above and released into YPD medium containing 5 mM auxin.

## Screening for suppressors of scc4-4

Forty independent colonies of the parental strain (YCplac33::*scc4-4::NATMX scc4Δ::HIS3* [K23967]) were picked and grown overnight at 25°C. Each was plated at 5 $OD_{600}$ units per plate over three plates and incubated at 35.5°C until colonies appeared. Up to three colonies were picked from each plate and streaked for single colonies at 25°C before being retested for growth at 35.5°C. Those that grew at 35.5°C were checked by PCR from genomic DNA preparations for revertants of Scc4. Isolated suppressors that did not show revertant mutations were checked for 2:2 segregation and grouped into complementation groups prior to deep sequencing. To check for the ability to rescue the deletion of Scc4, suppressors were streaked onto 1 mg/mL 5-FOA plates and allowed to grow for 2 days.

## Protein purification from *E. coli*

BL21(DE3) strains containing plasmids encoding proteins for purification were grown at 37°C in 2XTY media supplemented with the appropriate antibiotic until an $OD_{600}$ of 0.6 was reached. Expression was induced by addition of IPTG to a concentration of 1 mM for 16 hr at 20°C. Cells were harvested by centrifugation and mixed with five times the cell pellet volume of lysis buffer (250 mM NaCl, 50 mM Tris-HCl pH 7.5, 1 mM EDTA, 2 mM β-mercaptoethanol, 1 tablet/50 mL protease inhibitor cocktail [Roche]). Cells were lysed by passage through a cell disruptor (Constant Systems) at 20 kpsi. PMSF (Sigma) was added to the lysate to a final concentration of 1 mM. Samples were sonicated on ice for 1 min/50 mL in 30 s intervals with a Vibra-cell sonicator (VCX 130FSJ; Sonics and Materials) at 80% amplitude. Lysate was cleared by centrifugation at 50,000 g for 90 min in an Avanti J-26S XP centrifuge (Beckman Coulter).

Affinity purification was performed by incubating cleared lysates with pre-equilibrated Talon Superflow Metal Affinity Resin (500 µL/50 mL lysate; Takare Bio) for 1 hr at 4°C. Beads were sedimented by centrifugation at 700 g for 2 min and the supernatant decanted and discarded. The resin was washed five times with 50 mL lysis buffer containing 10 mM imidazole. Proteins wereeluted with a volume of elution buffer (lysis buffer +250 mM imidazole) equal to twice the volume of resin.

Size exclusion chromatography was performed by injecting up to 2 mL of eluent onto a HiLoad 16/60 Superdex 200 prep grade column (equilibrated with Buffer A: 95 mM NaCl, 20 mM HEPES pH 7.5, 2 mM β-mercaptoethanol) connected to an ÄKTApurifier 100 purification system controlled by UNICORN software (GE Healthcare). Peak fractions were concentrated using Vivaspin columns (Sartorius Stedim Biotech) with a molecular weight cutoff of 10 kDa. Concentration of purified protein was determined based on its absorbance at 280 nm, measured using a NanoDrop-1000 (Thermo Fisher Scientific).

## In vitro hinge binding assay

WT and mutant MBP-Smc1$^{hinge}$-HIS6 and Smc3$^{hinge}$-FLAG3-HIS6 proteins were expressed and purified as described above. Proteins were mixed at an equimolar concentration of 250 mM in Buffer A and incubated at 16°C with shaking at 1000 rpm for 15 min. 400 µL of protein mixture was added to 20 µL of pre-equilibrated ANTI-FLAG M2 affinity gel and incubated at 16°C with shaking at 1000 rpm for 15 min. Resin was sedimented by centrifugation for 1 min at 850 g and washed three times with Buffer A + 1% Triton X-100 (Sigma) before being boiled in 2× SDS sample buffer prior to immunoblotting.

## Protein gel electrophoresis and western blotting

The samples were mixed with 4× LDS sample buffer (NuPAGE Life Technologies), loaded onto 3–8% Tris-acetate gels (NuPAGE, Life Technologies) and the proteins separated using an appropriate current. The proteins were then transferred onto 0.2 µm nitrocellulose using Trans-blot Turbo transfer packs for the Trans-blot Turbo system (Bio-Rad). The following antibodies were used: anti-V5 (Bio-Rad), anti-HA (Roche), His-tag antibody (GenScript), and A-14 (Santa Cruz Biotech). For visualisation, the membrane was incubated with Immobilon Western Chemiluminescent HRP substrate (Millipore) before detection using an ODYSSEY Fc Imaging System (LI-COR).

## Multiple sequence alignment

Multiple sequence alignments were created using Clustal Omega (*Sievers et al., 2011*).

## Calibrated ChIP-seq

Cells were grown exponentially to 0.5 $OD_{600}$ and the required cell cycle stage where necessary. 15 $OD_{600}$ units of *S. cerevisiae* cells were then mixed with 5 $OD_{600}$ units of *Candida glabrata* to a total volume of 45 mL and fixed with 4 mL of fixative (50 mM Tris-HCl, pH 8.0; 100 mM NaCl; 0.5 mM EGTA; 1 mM EDTA; 30% [v/v] formaldehyde) for 30 min at room temperature (RT) with rotation. Fixation was quenched with 2 mL of 2.5 M glycine incubated at RT for 5 min with rotation. The cells were then harvested by centrifugation at 3500 rpm for 3 min and washed with ice-cold 1× PBS. The cells were then resuspended in 300 µL of ChIP lysis buffer (50 mM HEPES-KOH, pH 8.0; 140 mM NaCl; 1 mM EDTA; 1% [v/v] Triton X-100; 0.1% [w/v] sodium deoxycholate; 1 mM PMSF; 1 tablet/25 mL protease inhibitor cocktail [Roche]) and an equal amount of acid-washed glass beads (425–600 µm, Sigma) added before cells were lysed using a FastPrep−24 benchtop homogeniser (M.P. Bio-medicals) at 4˚C (3 × 60 s at 6.5 m/s or until >90% of the cells were lysed as confirmed by microscopy).

The soluble fraction was isolated by centrifugation at 2000 rpm for 3 min, then sonicated using a Bioruptor (Diagenode) for 30 min in bursts of 30 s 'on' and 30 s 'off' at high level in a 4˚C water bath to produce sheared chromatin with a size range of 200–1000 bp. After sonication, the samples were centrifuged at 13,200 rpm at 4˚C for 20 min and the supernatant was transferred into 700 µL of ChIP lysis buffer. 30 µL of protein G Dynabeads (Thermo Fisher) were added, and the samples were pre-cleared for 1 hr at 4˚C. 80 µL of the supernatant was taken as the whole-cell extract (WCE) and 5 µg of antibody (anti-PK; Bio-Rad) was added to the remaining supernatant which was then incubated overnight at 4˚C. 50 µL of protein G Dynabeads were then added and incubated at 4˚C for 2 hr before washing 2× with ChIP lysis buffer, 3× with high salt ChIP lysis buffer (50 mM HEPES-KOH, pH 8.0; 500 mM NaCl; 1 mM EDTA; 1% [v/v] Triton X-100; 0.1% [w/v] sodium deoxycholate; 1 mM PMSF), 2× with ChIP wash buffer (10 mM Tris-HCl, pH 8.0; 0.25 M LiCl; 0.5 % NP-40; 0.5% sodium deoxycholate; 1 mM EDTA; 1 mM PMSF), and 1× with TE pH 7.5. The immunoprecipitated chromatin was then eluted by incubation in 120 µL of TES buffer (50 mM Tris-HCl, pH 8.0; 10 mM EDTA; 1% SDS) for 15 min at 65˚C and the collected supernatant termed the IP sample. The WCE extracts were mixed with 40 µL of TES3 buffer (50 mM Tris-HCl, pH 8.0; 10 mM EDTA; 3% SDS), and all samples were de-crosslinked by incubation at 65˚C overnight. RNA was degraded by incubation with 2 µL RNase A (10 mg/mL; Roche) for 1 hr at 37˚C, and protein was removed by incubation with 10 µL of proteinase K (18 mg/mL; Roche) for 2 hr at 65˚C. DNA was purified using ChIP DNA Clean and Concentrator kit (Zymo Research).

## Extraction of yeast DNA for deep sequencing

Cultures were grown to exponential phase ($OD_{600}$ = 0.5). 12.5 $OD_{600}$ units were then collected and diluted to a final volume of 45 mL before fixation as described in the protocol for ChIP-seq. The samples were treated as specified in the ChIP-seq protocol up to the completion of the sonication step whereby 80 µL of the samples were carried forward and treated as WCE samples.

## Preparation of sequencing libraries

Sequencing libraries were prepared using NEBNext Fast DNA Library Prep Set for Ion Torrent Kit (New England Biolabs) according to the manufacturer's instructions. To summarise, 10–100 ng of fragmented DNA was converted to blunt ends by end repair before ligation of the Ion Xpress Bar-code Adaptors. Fragments of 300 bp were then selected using E-Gel SizeSelect2% Agarose gels (Life Technologies) and amplified with 6–8 PCR cycles. The DNA concentration was determined by qPCR using Ion Torrent DNA standards (Kapa Biosystems) as a reference. 12–16 libraries with different barcodes could then be pooled together to a final concentration of 350 pM and loaded onto the Ion PI V3 Chip (Life Technologies) using the Ion Chef (Life Technologies). Sequencing was performed on the Ion Torrent Proton (Life Technologies), typically producing 6–10 million reads per library with an average read length of 190 bp.

## Data analysis, alignment, and production of BigWigs

Quality of the reads was assessed using FastQC and trimmed as required using fastx_trimmer. Generally, this involved removing the first 10 bases and any bases after the 200th, but trimming more or fewer bases may be required to ensure the removal of kmers and that the per-base sequence

content is equal across the reads. Reads shorter than 50 bp were removed using 'FilterFastq.py' and the remaining reads aligned to the necessary genome(s) using Bowtie2 with the default (–sensitive) parameters (*Langmead and Salzberg, 2012*).

To generate alignments of reads that uniquely align to the *S. cerevisiae* genome, the reads were first aligned to the *C. glabrata* (CBS138, Génolevures; *Dujon et al., 2004*) genome with the unaligned reads saved as a separate file. These reads that could not be aligned to the *C. glabrata* genome were then aligned to the *S. cerevisiae* (sacCer3, SGD) genome and the resulting BAM file converted to BigWigs for visualisation. Similarly, this process was done with the order of genomes reversed to produce alignments of reads that uniquely align to *C. glabrata*.

### Visualisation of ChIP-seq profiles

The resulting BigWigs were visualised using the IGB browser (*Nicol et al., 2009*). To normalise the data to show quantitative ChIP signal, the track was multiplied by the sample's OR and normalised to 1 million reads using the graph multiply function.

In order to calculate the average occupancy at each base pair up to 60 kb around all 16 centromeres, the BAM file that contains reads uniquely aligning to *S. cerevisiae* was separated into files for each chromosome and a pileup of each chromosome was then obtained using samtools mpileup. These files were then amended using our own script 'chr_position.py' to assign all unrepresented genome positions a value of 0. Each pileup was then filtered using another in-house script 'filter.py' to obtain the number of reads at each base pair within up to 60 kb intervals either side of the centromeric CDEIII elements of each chromosome. The number of reads covering each site as one successively moves away from these CDEIII elements could then be averaged across all 16 chromosomes and calibrated by multiplying by the samples OR and normalising to 1 million reads. All scripts written for this analysis method are available on request.

### Identification of mutations from whole genome sequencing

Pileups were created using samtools mpileup (-v –skip-indels –f sacCer3.fa –o *sample name*.vcf *sample name*.bam), then SNPs were called using bcftools call (-v –c –o *sample name*.bcf *sample name*.vcf). To find mutations unique to a suppressor strain, lists of SNPs from the parental strain or backcrossed clones of the suppressor strain were compared to the list of SNPs from the suppressor strain. In the case of parental strains, mutations that were present in both were removed, and in the case of backcrossed clones of the suppressor strain, mutations that were present in both were kept in order to identify the mutation that caused the suppression phenotype. This was done using 'MutationFinder.py' and the resulting lists further narrowed using 'yeastmine.py' which searches the Saccharomyces Genome Database (SGD) for genes that correspond to the position of each mutation so that those that lie outside of genes could be removed. From this it was possible to identify the mutation in each suppressor that gave rise to the suppressor phenotype.

### ATPase assay

ATPase activity was measured by using the EnzChek phosphate assay kit (Invitrogen) by following the provided protocol. Cohesin in various complexes were mixed to a final concentration of 50 nM in under 50 mM NaCl in the presence of 700 nM 40 bp dsDNA in those experiments testing the effect of duplex DNA. The reaction was started with addition of ATP to a final concentration of 1.3 mM, always in a final volume of 150 μL. ATPase activity was measured by recording absorption at 360 nm every 30 s for 1 hr 30 min using a PHERAstar FS. ΔAU at 360 nm was translated to Pi release using an equation derived by a standard curve of $KH_2PO_4$ provided with the EnzChek kit and according to instructions. The reactions were assumed linear for at least the first 10 min of the experiment and rates calculated using this time period. On completion, a fraction of each reaction was analysed by SDS-PAGE and the gel stained with Coomassie Brilliant Blue in order to test that the complexes were intact throughout the experiment and that equal amounts were used when testing various mutants and conditions. At least two independent biological experiments were performed for each experiment.

## Cohesin protein expression and purification

WT or 6C cohesin trimer, Scc2, and Pds5 were expressed and purified as described in *Collier et al., 2020* and *Bürmann et al., 2019*, respectively. In brief, vectors containing *S. cerevisiae* cohesin trimers were generated by combining pACEbac1 *SMC1*-His *SMC3* containing the 6C cysteine mutations (Smc1K639C-Smc3E570C, Smc1G22C-Scc1A547C, and Smc3S1043C-Scc1C56) with pIDC *SCC1*-2xStrepII by a Cre recombinase reaction (New England Biolabs). Sequences of *S. cerevisiae* Scc2 and Pds5 were individually cloned as 2xStrepII-(151-1493)Scc2 and 2xStrepII-Pds5 into Multibac vectors, yielding 2xStrepII-ΔN150-Scc2-pACEbac1 and 2xStrepII-Pds5-pACEbac1 with an HRV 3C protease site (LEVLFQ/GP) in the tag linker. Expression of the 6C trimer, Scc2, and Pds5 was done individually in Sf9 insect cells followed by the same previously described three-step purification protocol: proteins were purified via affinity pulldown of their StrepII and eluted with desthiobiotin, 3C protease was added to the eluents to cleave the affinity tags, the cleavage products were further purified by anion exchange columns, and finally buffer exchanged to Buffer 6C (50 mM Tris-HCl pH 7.5, 150 mM NaCl, 1 mM TCEP, 10% glycerol). The purified trimer, Scc2, and Pds5 proteins were then frozen in liquid nitrogen and stored at −80℃ until further use.

## Cryo-EM grid preparation

For imaging of cohesin with cryo-EM, the purified 6C trimer and Scc2 or Pds5 were mixed at a 1:1.5 molar ratio and injected onto a Superose 6 Increase 3.2/300 column (GE Healthcare) in buffer containing 25 mM HEPES-NaOH pH 7.5, 150 mM NaCl, 1 mM TCEP. The tetramer fraction was incubated with 2 mM BMOE for 3 min at room temperature, and then buffer exchanged into buffer 6C with Zeba spin buffer exchange columns (Sigma Aldrich). For ATP-containing sample 5 mM ATP, 2 mM $MgCl_2$ was added to the buffers.

Grids were prepared by applying 3 μL of sample at a concentration of 0.2–0.3 mg/mL to freshly glow-discharged Cu/Rh 2/2 holey carbon 200 mesh grids (Quantifoil). The grids were blotted for 1.5–2 s at 4℃ with humidity at 100% and were flash frozen using a Vitrobot (Thermo Fisher Scientific).

## Cryo-EM data collection, processing, and modelling

Images were recorded on a Titan Krios electron cryo-microscope (FEI) equipped with a K2 or K3 summit direct electron detector with the use of a Volta phase plate (VPP) and varying pixel sizes between 1.09 and 1.16 Å/pixel. Micrographs were collected with total doses of ~40 electrons per Å 2, dose-fractionated into 40 movie frames, and at defocus ranges of 0.5–0.9 μm. All datasets containing the same sample were merged as described by *Wilkinson et al., 2019*, resulting in a final pixel size of 1.16 Å. Image processing was done in RELION 3.0 (*Zivanov et al., 2018*) and cryo-SPARC (*Punjani et al., 2017*). Movies were aligned using 5 × 5 patches using MotionCor2 with dose-weighting (*Zheng et al., 2017*). CTF parameters were estimated with Gctf (*Zhang, 2016*). All refinements were performed using independent data half-sets (gold-standard refinement) and resolutions were determined based on the Fourier shell correlation (FSC = 0.143) criterion (*Rosenthal and Henderson, 2003*). Due to the elongated shape of cohesin, particle picking was done with the help of the machine learning-based crYOLO software (*Wagner et al., 2019*). Initial 2D classifications and the first initial model made with cryoSPARC revealed intrinsic flexibility between the upper part of the complex, containing the hinge and the coiled coils, and the lower part, containing the HAWK-bound heads. Therefore, after an initial round of 3D refinement, the two parts were extracted and re-centred separately for all downstream processing. Specific EM processing strategies are discussed in detail in *Figure 5—figure supplement 2*. All depictions of these structures within the paper were made with the use of UCSF ChimeraX (*Goddard et al., 2018*).

To produce the coordinate map of the folded elbow, a homology model of the yeast hinge dimer was obtained from SWISS-MODEL (*Waterhouse et al., 2018*) using a crystal structure of the hinge from *Mus musculus* (PDB: 2WD5) as the template (*Kurze et al., 2011*). MAIN (*Turk, 2013*), and COOT (*Emsley et al., 2010*) were used for manual rebuilding, followed by refinement using Phenix. real_space_refinement (*Hu, 2018*). Manual rebuilding and refinement were repeated for several cycles.

## In vivo photo crosslinking

Yeast stains bearing TAG-substituted Smc1-myc9 plasmid and pBH61 were grown in −Trp −Leu SD medium containing 1 mM BPA. Cells were collected and resuspended in 1 mL of ice-cold PBS buffer. The cell suspension was then placed in a Spectrolinker XL-1500a (Spectronics) and irradiated at 360 nm for 2 × 5 min. Extracts were prepared as described previously (*Hu et al., 2011*) and 5 mg of protein were incubated with 5 µL of Anti-PK antibody (Bio-Rad) for 2 hr at 4℃. Next, 50 µL of Protein G Dynabeads (Thermo Fisher) were added and incubated overnight at 4℃ to immunoprecipitate Scc1. After washing five times with lysis buffer, the beads were boiled in 2× SDS-PAGE buffer. Samples were run on a 3–8% Tris-acetate gel (Life Technologies) for 3.5 hr at 150 V. For western blot analysis, anti-Myc (Millipore), anti-FLAG (Sigma), and anti-HA (Roche) antibodies were used to probe the indicated proteins.

## In vivo cysteine crosslinking

15 OD units of cells grown in exponential phase were washed with ice-cold PBS and kept on ice throughout the experiment. Cells were resuspended in 500 µL cold PBS and 300 µL was added to 2 × 2 mL bead beater tubes. 12.5 µL BMOE (125 mM in DMSO to a final concentration of 5 mM) or 12.5 µL DMSO was added before incubating on ice for 6 min. Cells were washed twice with 1 mL cold PBS containing 5 mM DTT.

Crosslinked cells were resuspended in 500 µL lysis buffer (150 mM NaCl; 5 mM EDTA; 0.5% [v/v] NP40; 500 mM Tris-HCl, pH 7.5; 1 mM PMSF; 1 tablet/50 mL protease inhibitor cocktail [Roche]; 1 mM DTT) and an equal volume of acid-washed glass beads (425–600 µm, Sigma) was added. The cells were lysed using a FastPrep−24 benchtop homogeniser (M.P. Biomedicals) at 4℃ for 3 × 60 s at 6.5 m/s with a 5 min rest between cycles, until >90% of the cells were lysed as confirmed microscopically. The insoluble fraction was pelleted by centrifugation at 13,200 rpm for 10 min and the supernatant isolated and analysed by western blot.

## Data and software availability

All scripts written for this analysis method are available to download from https://github.com/naomi-petela/nasmythlab-ngs (copy archived at https://archive.softwareheritage.org/swh:1:rev:d7509c6f3e0a0f34db71b485a9e332223084e7be). The accession number for the next-generation sequencing data (raw and analysed) reported in this paper is GSE167318.

## Acknowledgements

We are grateful to Frank Uhlmann for sharing yeast strains, Katsu Shirahige for the anti-AcSmc3 antibody, and to Maria Demidova, Wentao Chen, and Christophe Chapard for invaluable technical assistance. We would like to thank all the members of the Nasmyth and Löwe groups for valuable discussions. This work was funded by the Wellcome Trust (107935/Z/15/Z to KN; 202754/Z/16/Z to JL; 202062/Z/16/Z to BH), Cancer Research UK (26747 to K N), the European Research Council (294401 to KN), the Medical Research Council (U105184326 to JL), the Biotechnology and Biological Sciences Research Council (BB/S002537/1 to B H), and the National Research Foundation of Korea (B-HO, NRF2020R1A4A3079755).

## Additional information

### Funding

| Funder | Grant reference number | Author |
|---|---|---|
| Wellcome Trust | 107935/Z/15/Z | Kim A Nasmyth |
| Wellcome Trust | 202754/Z/16/Z | Jan Löwe |
| Wellcome Trust | 202062/Z/16/Z | Bin Hu |
| Cancer Research UK | 26747 | Kim A Nasmyth |
| H2020 European Research Council | 294401 | Kim A Nasmyth |

| Medical Research Council | U105184326 | Jan Löwe |
|---|---|---|
| Biotechnology and Biological Sciences Research Council | BB/S002537/1 | Bin Hu |
| National Research Foundation of Korea | NRF2020R1A4A3079755 | Byung-Ha Oh |

The funders had no role in study design, data collection and interpretation, or the decision to submit the work for publication.

## Author contributions

Naomi J Petela, Conceptualization, Data curation, Software, Formal analysis, Validation, Investigation, Writing - original draft, Writing - review and editing; Andres Gonzalez Llamazares, Conceptualization, Formal analysis, Validation, Investigation, Writing - original draft, Writing - review and editing; Sarah Dixon, Byung-Gil Lee, Jean Metson, Heekyo Seo, Antonio Ferrer-Harding, Menelaos Voulgaris, Thomas Gligoris, Investigation; Bin Hu, Funding acquisition, Investigation; James Collier, Resources; Byung-Ha Oh, Supervision, Funding acquisition, Investigation; Jan Löwe, Conceptualization, Formal analysis, Supervision, Funding acquisition, Writing - original draft, Project administration, Writing - review and editing; Kim A Nasmyth, Conceptualization, Formal analysis, Supervision, Funding acquisition, Validation, Writing - original draft, Project administration, Writing - review and editing

## Author ORCIDs

Naomi J Petela https://orcid.org/0000-0001-9607-0422
Andres Gonzalez Llamazares https://orcid.org/0000-0001-5404-6360
Byung-Gil Lee http://orcid.org/0000-0001-9565-6114
James Collier http://orcid.org/0000-0002-9904-9423
Jan Löwe https://orcid.org/0000-0002-5218-6615
Kim A Nasmyth https://orcid.org/0000-0001-7030-4403

## Decision letter and Author response

Decision letter https://doi.org/10.7554/eLife.67268.sa1
Author response https://doi.org/10.7554/eLife.67268.sa2

# Additional files

## Supplementary files

• Supplementary file 1. Table detailing the amino acid substitutions made at position 588 in Smc1, with their respective ability to complement *smc1Δ* and *scc4Δ*.

• Supplementary file 2. Data collection and refinement statistics for the Smc1D574Y-Smc1 mouse hinge structure.

• Supplementary file 3. Data and model building statistics for all cryo-EM structures.

• Supplementary file 4. List of yeast strains and genotypes.

• Transparent reporting form

## Data availability

PDB validation reports of the crystal structures are included in the manuscript. All scripts written for this analysis method are available to download from https://github.com/naomipetela/nasmythlab-ngs (copy archived at https://archive.softwareheritage.org/swh:1:rev:d7509c6f3e0a0f34db71b485a9e332223084e7be).

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
