## [Decision Letter]

**Acceptance summary:**

This paper is of interest for biologists studying SMC proteins and their role in various aspects of genome biology. The paper demonstrates that folding of cohesin's coil-coil, which previously had been demonstrated to occur in vitro, actually occurs in cells. Through a series of biochemical and structural studies, the authors demonstrate that such folding likely enables DNA loading by enabling an association with SCC2, the cohesin loader. The underlying mechanism, and if such folding has a role in DNA loop extrusion, remains unclear.

**Decision letter after peer review:**

Thank you for submitting your article "Folding of cohesin's coiled coil is important for Scc2/4-induced association with chromosomes" for consideration by *eLife*. Your article has been reviewed by 3 peer reviewers, including Adèle L Marston as the Reviewing Editor and Reviewer #1, and the evaluation has been overseen by Cynthia Wolberger as the Senior Editor. The following individual involved in review of your submission has agreed to reveal their identity: Daniel Panne (Reviewer #2).

Essential revisions:

Specific points related to these 4 essential revisions and which should be addressed are given in the individual reviews below.

1) Please provide full details of how the structural analyses were done to allow a full review of the relevant parts of the paper. The authors should provide clarity on which maps and models the authors intend to submit to the databases (EMDB and PDB). A table should be provided indicating what the files are and where they appear in the updated manuscript. They should be aligned (PDB with corresponding maps, where applicable) so that their fitting can be evaluated. More details of specific information that should be included is provided in the comments from reviewers #2 and 3.

2) A key point of the paper is that the SMC1/3 hinge interacts directly with Scc2. The authors provide cross-linking data as evidence that this occurs in vivo. However some of the cross linking data are not always convincing as some controls are missing. FigS4F is lacking the WT control. Data shown in Fig4G is also not fully convincing: Cys crosslinks should only occur if both SMC1 and SCC2 carry the Cys mutation. The BPA crosslink shown in Fig4E is in principle sufficient (but the Figure legend needs to be revised as it it shows SCC2 not PDS5 cross linking). In addition, please provide a full description of how these experiments were done in the methods section. (Specific points from all three reviewers).

3) The paper covers a lot of ground and is therefore difficult to follow in places. In revising the paper, it would be helpful to focus on the main narrative and remove some of the discussion that is not directly relevant to the main conclusions. (Specific points from all three reviewers).

4) The section on the requirement for nucleosome free regions for cohesin loading should be revised. Some of the data presented is inconclusive and while the authors do not find evidence for a role of RSC in cohesin loading, in contrast with an earlier report, this message is a distraction from the more exciting main narrative. It could therefore be given less attention as suggested in specific points from reviewers #1 and 2.

*Reviewer #1:*

In this manuscript, the authors address the mechanism of how cohesin regulators regulate its loading onto chromosomes. Cohesin loading requires the Scc2 protein, which forms a complex with Scc4. Previous work had demonstrated that the role of Scc4 is two-fold. First, it specifically targets cohesin loading to centromeres. Second, it activates Scc2/cohesin in some unknown way to facilitate cohesin loading genome-wide. Here the authors address this second function of Scc4, starting by isolating suppressors of the scc4-4 temperature sensitive mutation. This leads them to a mutation in the SMC hinge (smc1D588Y). This was initially surprising since Scc2 binds cohesin close to the ATPase heads which are distant from the hinge when cohesin is in its extended, rod-like conformation. However, a beautiful cryo-electron microscopy structure of cohesin in complex with Scc2 revealed that cohesin bending at the elbow (as has been observed in other SMC complexes) brings the hinge into close contact with Scc2 and the Smc3 coiled-coil. Using engineered cysteines and a cross-linking approach, the authors also provide convincing evidence that cohesin folding occurs in vivo. Using the same approach, they show that smc1D588Y stabilizes the folded conformation in complex with Scc2, while impairing association with Pds5, a mutually exclusive cohesin maintenance factor. The authors surmise that smc1D588Y allows cohesin loading without Scc4 by favouring the interaction between Scc2 and the Smc1/Smc3 hinge.

In addition to this main narrative, the authors report several additional findings. These include isolation of additional suppressors of the scc4-4 mutation which map to histone residues that interact with DNA. Prompted by this discovery, the authors re-examine previous work which reported a role for the chromosome remodeller, RSC, in cohesin recruitment to chromosomes. While the authors find no strong evidence for a direct role of RSC in cohesin recruitment, the relationship between histones/remodellers and cohesin loading remains unclear and this part of the manuscript is not necessary for the main message of the manuscript.

The strength of this manuscript is the range of complementary approaches used to dissect a central part of the mechanism of a central player in chromosome biology. Experiments are rigorously designed and appropriately interpreted. The manuscript contains a lot of detailed information, some peripheral to the main message, which may challenge readers not immersed in the field. The manuscript may benefit from some streamlining/reorganisation to ensure the key messages are highlighted for the general reader.

Comments for the authors:

1. The authors should considering shortening the section on RSC describing Figure S3 as it detracts from the main message of the manuscript. In addition, while the ChIP-seq in Figure S3D argue against a major role for RSC in cohesin loading, the nucleosome mapping data in Figure S3S are not conclusive, as presented. Why do the upper two and lower two nucleosome profiles look different from each other? Potentially effects on nucleosome positioning will only be visible in metagene analysis: the sth1-3 mutant at least would be expected to alter nucleosome positioning but this is not apparent from the data presented. The authors should either remove this data or perform additional analysis.

2. Figure 7G: The conclusion that cohesin folding at the elbow is a feature of cohesin complexes that are holding sister chromatids together relies on the assumption that all acetylated cohesin is participating in cohesion. However, this may not be the case. In mammals, cohesin acetylation is also observed in G1, indicating that it may have a more general role. This potential caveat should be mentioned.

3. It is surprising that there is high Scc1 ChIP-seq signal at centromeres in scc4-4 cells if Scc4 is required for cohesin loading at centromeres (Figure S4A). How do the authors explain this?

4. Line 273: It looks like smc1D588Y CAN rescue the proliferation defect of scc2-4 in Figure S1E.

5. Figure 3A please show proliferation data for the different histone mutants.

6. Line 402: Smc3 acetylation

7. Supplementary Figure 4F: needs a negative control without cysteine substitutions.

8. Page 19: Line 547: Call to Figure S4D is incorrect.

*Reviewer #2:*

Patela et al. aim to understand how folding of 50nm long coiled-coils contributes to cohesin function. Using a combination of genetics and crosslinking experiments, the authors demonstrate that a mutation in the cohesin hinge enables cohesin loading in the absence of Scc4 and that the hinge directly interacts with Scc2 or Pds5. CryoEM structures confirm that the coiled-coils fold back into position in which the SMC hinges interact with the coiled-coil. Such a conformation is also detected in cross-linking experiments in cells, thus supporting the view that the hinges are frequently positioned in close proximity of Scc2 or PDS5. This study thus presents an important step forward in understanding how DNA loading is achieved by the cohesin complex. The described mechanism may be important for the control of loop extrusion or sister chromatid cohesion.

The conclusions are mostly well supported by the data, but some aspects of the biochemical and structural data require clarification.

1. Biochemical data: Figure 4B: The FLAG blot indicates that x-linked species migrate at different positions on SDS-PAGE while the top panel shows uniform mobility. How do we know that the band labeled SMC1-? corresponds to x-linked SMC1 and not some non-specific anti-myc background (e.g. there is an additional non-labeled band running lower)? Figure 4B: Different amounts of FLAG-HAWKs are recovered in the anti-Myc IP. Can the authors generate an 'input' blot showing expression levels to exclude the possibility that the variable crosslinking efficiency is not due to different expression levels?

2. Figure S4F: Show a WT negative control (no Cys mutation) to confirm that the bands observed do not arise due to non-specific background cross-linking.

3. Figure 4G: Top: Why is there a crosslinked band in the lane containing Smc1K620C alone? Would it not be expected to see cross-linking only when both Scc2N200C and Smc1K620C are present? Why is the non-specific crosslinking (labeled as *) present everywhere? Which cross-linked band they are referring to? While there is a small amount of a band visible above the non-specific band, this weak band is also present in Smc1K620C alone!

4. Structural data: Figure 5A: The authors need to clarify if in their cryoEM structure, the SMC ATPase heads are in the 'apo' state, as indicated in the text or in the engaged state as indicated in the Discussion (L. 641).

5. The authors need to explicitly state in their description of their structural data (Line 512 et seq) that they used a crosslinked version of the cohesin trimer containing 6 Cysteines, positioned at the SMC-SCC1 and SMC1-3 ATPase head heterodimerization interfaces.

6. Crosslinking would be expected to stabilize the SMC ATP heads in close proximity, potentially an engaged state. Would full SMC head disengagement be prevented by such cross-linking even in the absence of nucleotide?

7. From Figure 5 the state of the heads is not immediately apparent. Please include a Figure comparing the state of the 'disengaged' SMC heads obtained using their cross-linking method with that of the ATP engaged state published previously.

8. If indeed, crosslinking stabilizes an engaged but nucleotide-free form, the authors need explicitly discuss the potential implications. For example, crosslinking of the heads could prevent the heads from properly disengaging. This has potential implications for the conformation/interactions of Scc2 or PDS5.

9. Figure 5G: The authors need to clarify how they obtain the ATP-bound state. Did they also use a crosslinked cohesin trimer, hydrolysis-impaired mutant ATPase heads or non-hydrolyzable ATP variants?

10. What is the distance between Smc1D588 and SCC2? Figure 5B indicates ~25Å which would be too far away for a direct interaction. In contrast, Figure 5E and their text description suggest that 'folding of the coiled coil brings the hinge to within 12nm (they probably mean Å?) of the heads'. Figure 5E indicates a distance of 12Å but it is between Smc1K620BPA and Smc1D588Y and not SCC2.

11. If indeed no direct interaction is apparent, or if the low resolution and flexibility do not allow firm conclusions, they need to indicate this during their discussion of these results.

12. Figure 6: While biochemical data indicate that PDS5 inhibits ATP hydrolysis, the presented structural data does not reveal how this is accomplished. One caveat of using a cross-linked version is that the procedure may prevent access of PDS5 to fully disengaged SMC heads and thus prevent SMC head engagement (in analogy to the role of the condensin YCS4 subunit). The authors need to indicate potential pitfalls of their cross-linking procedure on PDS5 positioning and discuss if the conformation observed is potentially off pathway.

Comments for the authors:

1. Figure 4C-F: Please indicate if the experiment has been done once or repeated several times with consistency.

2. Can they indicate if their preparation used for CryoEM is fully or partially BMOE cross-linked (e.g. by SDS-PAGE analysis).

3. Figure 6: While at lower resolution, it would still be important to show how well the PDB models fit into the cryoEM density map shown in Figure 6A. This would give the reader a sense for how reliable positioning of the PDB models (shown in Figure 6B) are.

4. Can they indicate where the previously published (Rowland et al. 2009, Sutani et al. 2009) eco1-1 suppressor mutations of PDS5 are located? Are suppressor mutations located in observed interfaces? Would they be predicted to interfere with PDS5 interaction?

5. L.593 et seq: The authors mention that their data are in contrast to 'previous studies with shortened constructs'. Previous work needs to be cited.

6. L280 : This should read: 'relative to the position of D574'?

7. Figure S4E: Cartoon shows SMC3! Do they mean SMC1?

8. Details on CryoEM data processing and reconstruction information need to be included.

9. PDB validation reports of the modeled cryoEM structures are missing.

10. It is not clear how the PDB model for the cohesin elbow is derived. No details of CryoEM/Xray data collection are given and PDB validation reports are again missing.

11. L. 1029: Spelling error: Katsu Shirahiga

*Reviewer #3:*

Chemical crosslinking experiments have established that sister chromatid cohesion requires "topological embrace" of sister chromatids within a cohesin ring. Three contacts define the ring: Scc1-N:Smc3-head, Scc1-C:Smc1-head, and Smc1-hinge:Smc3-hinge. The strongest experimental evidence for a specific DNA entry point is that artificial dimerization of the Smc1-hinge:Smc3-hinge interface prevents cohesion establishment (Gruber et al. 2006; Srinivasan et al. 2018). In the current manuscript, Patela et al. present genetic, biochemical, and structural data that indicate that the cohesin hinge interacts with the cohesin loading complex (Scc2/4), that this interaction depends on Scc4, and that mutations in the hinge that appear to enhance the interaction positively affect cohesin's association with chromosomes. Considered together, these data provide new evidence that the cohesin hinge is connected to the loading reaction. The manuscript is really interesting, includes many very clever experiments, and extends previous models for how cohesin works.

There are major issues with the paper that need to be addressed before publication. First, the paper is confusing. It should be simplified so that the main points are clearer, and errors in the text and figures should be corrected. Second, the section about nucleosome remodeling must be rewritten. There is great value in pointing out differences between studies, but these points can be made more succinctly. Third, the descriptions of the structures are insufficient and do not permit comparison with published structures. The reader cannot currently judge the validity of the interpretations presented.

With the exception of some minor updates to the biochemical experiments, the majority of these changes can be made without adding new data. The rewriting and corrections I have requested will nevertheless require significant time and attention. In addition, I have also requested the EM maps and corresponding docked models be made available to reviewers before a preliminary decision is reached. After this is done, I would support publication of a substantially updated version of the current manuscript in *eLife*.

1. The manuscript contains a collection of interesting observations, all of which address the overall hypothesis that Scc2/4 stimulates cohesion establishment by interacting with the cohesin hinge.

– See below for comments on the structures. The Pds5 structure should be moved to the supplemental material or removed altogether. The new structure is good enough to discern the overall position of Pds5 relative to cohesin, but it does not provide much new reliable information beyond this. It is pertinent in that it confirms competition between Scc2 and Pds5. Finer details like the status of the head domains, the identity of specific contacts, and the relationship to cohesin acetylation/Eco1 binding will have to wait for higher-resolution structures that will ideally be trapped in defined identifiable biochemical states.

– Like the Smc1/3 E-to-Q mutants, smc1D588Y appears to slow or stall cohesin at an early/intermediate step in the loading process (Figure 2C-D). The cells are viable, so the stall is not terminal. How do the authors reconcile this observation with their finding that cohesin loading on chromosome arms is actually elevated relative to wild type?

– Related to the above question: Have the authors explored how the mutation affects engagement with chromosomal DNA analogous to the recent Chappard paper? Are S-K complexes depleted in favor of others? Can they observe so-called E or J heads, and is either enriched in the mutant background? One line 353, the authors stop short of suggesting the smc1D588Y mutant favors the clamped state but imply it may stop the loading reaction at a step after the one stalled by the clamped-favoring EQ mutations. The current data do not conclusively show that the step slowed by smc1D588Y is before or after the step stalled by the EQ mutations. The authors should remove speculation on this point from the Results section.

– The description of results that begins on line 336 is very confusing. Specifically, "This implies that the reduced loading arises…" but the authors say directly above that Smc1D588Y is at 110% (CENs) of WT. Also, the first two sentences of this paragraph use "it," but the referent is not clear. Last, a simple explanation of the phenotype shown in Figure 2 seems to be that smc1D588Y preserves CEN targeting but disrupts translocation to pericentromeres. Arm loading is not disrupted (slightly enhanced). A simpler description of the phenotype would make the paragraph much easier to read and focus the reader on the important question: what accounts for the subtle defects seen in the smc1D588Y mutant?

– Line 456 – "Nevertheless, the level of Smc1-Scc2 crosslinked protein was comparable to that of Scc3…" This is an inappropriate comparison. Crosslinking efficiency could be far more efficient if there is a fortuitous lysine on the target (Scc3 or Scc2 here).

– Line 562 – This is a concrete biochemical prediction that is totally testable. Since it is central to the proposed mechanism of suppression, it should be tested using pulldown assays with purified proteins (cohesin tetramer, Scc2/4 or Scc2-C).

2. The section describing the RSC-Scc4 interaction needs to be revised. The authors focus on and oversimplify a secondary conclusion from a previous paper (Lopez-Serra et al. 2014), which holds that Scc4 stimulates RSC. The majority of the work and discussion in this and a related paper from the same group (Lopez-Serra et al. 2014 and Munoz et al. 2020) indicate that Scc2/4 localization depends on RSC. The Munoz paper offers a biochemical explanation for this observation but does not follow up on the secondary observation from the first paper that Scc2/4 influences nucleosome positioning.

– The current findings regarding the genome-wide distribution of cohesin in sth1-3, smc1D588Y, or related mutants seem mostly consistent with the results presented in Lopez-Serra and in Munoz. The authors should revise their text to reflect this concordance. In fact, all three studies emphasize the importance of nucleosome-free DNA. This could be pointed out.

– Any differences in the cohesin distribution between the two studies (Patela and Lopez-Serra), especially in the sth1-3 mutant must be more carefully annotated. Direct comparisons of the original studied loci (Lopez-Serra et al. Figure 3B, for example) must be presented if the authors wish to question the previous findings.

– That artificial chromosome tethering of Scc2-C rescues scc4D (Munoz Figure 6B) suggests the partially Scc4-dependent hinge interaction described here is not an essential function of Scc4. The authors suggest that this Scc4 function is rendered irrelevant when the loader is brought near DNA. At this point, there is not enough information to discriminate between these very similar models, and it would be better not to set the current manuscript up in opposition to previous work when an experimental distinction has not been made.

– Was the MNase-seq calibrated? Also, it looks like there are differences in nucleosome occupancy around AST2, which was also highlighted in the Lopez-Serra work (Figure S3G here). How are the two WT panels different? They show very different results.

– The authors say that RSC is not uniquely required for Scc2/4 recruitment, but Figure 3A from Lopez-Serra et al. shows that sth1-3 produces a pronounced cohesion defect, and other chromatin modifying enzyme mutants do not.

– Line 413 -The double mutant experiment (smc1D588Y sth1-3) is not very informative. It's unclear to me that suppression (or not) of the RSC phenotype by any of the mutants described in the current work would prove the previous sth1-3 findings to be unreliable. In addition, it would be important to compare cohesion, which is not done here. The double mutant experiments can be included, but the conclusions drawn are very limited and need to be rewritten.

– Line 416 – "We were also unable to reproduce the reported effects of scc4-4 and sth1-3 on nucleosome positioning…" If the authors really wish to make this statement/comparison, they need to be much more deliberate about evaluating the reported effects and then comparing their own data. There are clearly differences in Figure S3G, but the authors do not look at them with sufficient granularity. In particular, the Lopez-Serra paper looks at Scc2-bound genes/promoters. The authors must also compare their data with a more recent report (Kubik…Shore, Mol Cell, 2018). This study shows effects specifically at the +1 nucleosome of transcribed genes and is much more in line with the Lopez-Serra observations.

3. The new structures presented in the paper are compelling. There are several issues that must be cleared up before the manuscript can be published.

– It is unclear how the structures were determined. The authors should present data processing flowcharts to show how the most important structures in the paper were generated. It would be very helpful to know whether the structures described are major or minor components of the samples that were analyzed (relative to other determine structures). Therefore, particle numbers should be included for 3D classification steps, and if any other medium-high resolution structures came from the same datasets, those should be included in the chart for the benefit of the reader. If the authors wish not to disclose unpublished structures, they may elect to exclude some of these, but the overall breakdown of the data needs to be clear. If other structures produced by the dataset are already published, this is fine.

– Data collection tables need to be included, as do tables describing any "pseudo-atomic models." This is especially true for structures that will be deposited in the PDB. It looks like 5F, 5G, and 6A show structures that should be treated this way. For parts the authors do not wish to model explicitly (low-resolution parts corresponding to Scc2/Pds5), they can exclude these from the PDB and carry out model-map correlation calculations using maps truncated to the modeled residues.

– 2D classification results can be unreliable, especially when there are flexible modules. The analysis shown in Figure S4A-C should be removed, and the text should be modified so that any conclusions made from these images are removed. In particular, the consequences of ATP binding, discussed in relation to Figure S4C is inappropriate given that (1) the authors cannot resolve nucleotide with these images (is it really bound? hydrolyzed? Were there other classes in which the opposite was true?); (2) designation of Scc2 density is reasonable but not probative; and (3) it is not clear what the relationship is between the cohesin head domains. Subtle changes in head engagement could make huge differences in the interpretation. In the displayed images, it is not entirely clear that the heads are even engaged (versus "juxtaposed").

– How confident are the authors in their positioning of the Pds5 module and cohesin head domains in Figure 6A? The density below the "joint" as drawn is very difficult to interpret and looks like it might be artifactual.

– The authors should compare the cohesin head domain conformations they observe, especially for the Scc2 complex, to those observed in the recent structures showing cohesin-nucleotide-Scc3-Scc2-DNA interactions. Are the current structures well enough resolved to make these comparisons? Can the authors see Scc1-N:Smc3-head domain contact?

– Do the authors truly know the cohesin structure displayed in 5G is ATP-bound? To make this statement, they would need to resolve nucleotide and a connected γ phosphate.

– Why did the authors leave out Scc3 from the cryo-EM samples? How do they relate their observations to the published structures, which show the hinge attached to Scc3/Psc3 and dislocated from the Smc1/3 head domains? Both conformations probably inform on mechanism, but the reader is not presented with a helpful way to think about this. Does Scc3 "grab" the hinge after head-Scc2 engagement, as proposed by Uhlmann's group recently (Higashi et al. 2020)? To be sure, Srinivasan et al. proposed an Scc3-dependent late step in loading already, so both of these could easily be referenced in a paragraph contextualizing the current structures with respect to Scc3.

4. There are disagreements between data and text and confusing inconsistencies in the figures:

o S1C: why is Y40A not an scc4-ts allele?

o S1E – It looks like smc1D588Y is an scc2-4 suppressor. The text says the opposite.

o Figure S3 is discussed below.

o Genotypes should be given in color for all tetrad dissections (as in Figure 1).

o Size markers are missing from the blots. These are important for the crosslinking experiments so that positions can be compared for different antibodies (Figure 4B, for example).

o In at least once case, the authors should show a myc blot for Smc1-WT (no Cys mutation) so that it is clear the higher band is indeed an Smc1 crosslinking product.

o For these assays, the text and legend disagree on which subunit was immobilized for the pulldown (Scc1 or Smc1). If it was different in different experiments, this should be stated.

o Figure S4F is not helpful. There is no control shown (either Scc2 or Smc1, and ideally both, lacking Cys mutation). The crosslinking efficiencies do not seem to correlate for the two subunits, and it is not clear why some Scc2 mutants show an even higher band.

In the end, the experiment would have only been valuable if a really great crosslink position were found. The experiment can be removed without seriously damaging the manuscript.

– The scatter plot in figure S3E does not contradict the Lopez-Serra finding and is not a good way to look at this data. The Lopez-Serra paper addressed a small subset of chromosomal locations. It is not clear how far away from the diagonal the distribution of dots would need to spread to constitute a truly different localization pattern (no control for this).

– If the authors are sure of their placements for Scc2 and Pds5, then why not directly compare the two structures by overlaying them anchored on the coiled-coiled/hinge domains? This would be a great way of showing that (i) Pds5 is indeed more closely connected to the hinge and (ii) both modules engage the same overall cohesin conformation/fold (or not, as the case may be).

– Figure 5F-G make it look like the two structures are different views of the same. Only G is required for the point that head engagement does not unzip the coiled coils, so there is no need to make it look like a comparison.

– Figure S5D does not give the reader enough information to evaluate what is being shown. What dataset does this map come from? Is the corresponding main figure 5A? Was Smc3 included in the fit? DNA? Was nucleotide included in the sample? Why is there a large part of Scc2 poking out (presumably related to mentioned "floppiness")?

– Neither the Murayama papers nor the Huis in 't Veld papers are discussed. The Huis in 't Veld paper reports contact between SA1 and the cohesin hinge. Murayama and Uhlmann (2015) reconstitute this interaction (Psm1/3-Psc3) for the fission yeast components and state that it means the ring must bend and that the loader is probably involved. While neither study proved hinge-Scc3 contact is essential for the loading reaction, they were important contributions and should be cited.

– Presentation issues for structures:

o At least once in each figure (but not necessarily each panel), colored text should be shown for each included protein as is done in Figure 1. This is missing in Figure 5 (for example).

o The overlay in Figure 1F is hard to understand. The mutated structure should be shaded differently so the reader can see the difference clearly (pale green vs green and same for red).

o The nucleosome structure in Figure 3B is impossible to understand. The entire particle should be shown so the reader can see the DNA wrap, which is relevant here (do the mutations enhance unwrapping?). Only the mutated histones and mutated residues (highlight color) need to be colored.

[Editors' note: further revisions were suggested prior to acceptance, as described below.]

Thank you for resubmitting your work entitled "Folding of cohesin's coiled coil is important for Scc2/4-induced association with chromosomes" for further consideration by *eLife*. Your revised article has been evaluated by Cynthia Wolberger as the Senior Editor, and a Reviewing Editor.

The manuscript has been improved but there are some remaining issues that need to be addressed, as outlined in the points made by reviewers below. It is particularly important to clarify where the crosslinks reside in the cohesin variant used for structural analysis, as raised in Points 2 and 3 from Reviewer 2. If the crosslinks are in the heads, as indicated in the methods section, then this has important implications for the interpretation.

*Reviewer #2:*

The authors have addressed most comments. There are however a couple of issues that require further clarification:

1. Figure 4G: The author's response is only partially convincing: If Smc1K620C also crosslinks to Scc2C224, it is not fully clear that Scc2N220C specifically allows readout of contacts between the Smc1 hinge and Scc2. As they have tested a variety of Scc2 cysteine substitutions, it would have been important to test a Scc2 version that does not contain C224, in order to avoid such confounding non-specific crosslinking.

In any case, they need to further clarify how the different blots are developed. Presumably, the top panel shows an anti-FLAG western blot and the bottom panel an anti-myc blot (as indicated in the author's response)? The top panel is not labeled at all and the bottom panel is mislabeled as Anti-HA!

2. The authors state line 523 that they use a cohesin variant 'containing cysteines specifically crosslinking the three intermolecular interfaces'. They need to explicitly state which interfaces are crosslinked in the variant they use for structure determination. The author response indicates that 'the 6-cysteine version used interrogates the hinge dimerisation and not the head dimerisation'. Their methods section indicates however that the variant they use for structure determination does not contain Cys mutations in the hinges. Instead, they use Smc1G22C N1192C, Smc3S1043C R1222C, Scc1C56A547C.

These mutations are located (1) in the Smc3-NScc1 interface (Scc1C56 and Smc3S1043C), (2) Smc1-cScc1 (Scc1A547C and Smc1G22C) interface and (3), in the interface between the Smc1 and Smc3 ATPase heads (Smc1 N1192C and Smc3 R1222C). That is, the third pair of cysteines is clearly placed in the heads and not the hinges.

3. It is therefore not clear why the authors maintain (Reviewer #2 points 6, 7, 8, 12) that 'we did not cross-link the ATP head dimerisation interface'. Clearly, something is wrong! Maybe it is the methods section? If not, the authors need to clearly indicate that the version of cohesin they use is crosslinked at the SMC1/3 ATPase head domains. The implications of such head crosslinking need to be taken into account in the interpretation of their data and discussion of the results.

*Reviewer #3:*

The manuscript is greatly improved. The updated figures make it much clearer. The rewritten section describing Sth1, nucleosome positioning, and histone mutants is good. This is important work, and the breadth and strength of the experiments described are impressive.

Most of my comments have been adequately addressed. The requests below relate to the presentation and description of the data and should be addressed before publication but do not require extensive writing or any new experiments.

Line 593-597:

"In the presence of ATP" implies that ATP occupies both ATPase active sites. Neither γ phosphate can be resolved at this resolution. The structure could be a reaction intermediate resulting from slow π dissociation from one or both sites, for example. This is not to detract from the observations or statements made. Instead, it would be wise to state more clearly that ATP was included in the sample but avoid any suggestion that the structure provides detailed information about the SMC ATPase cycle, which will doubtless be a topic for future studies. This is rightly stated in the final paragraph of the Discussion.

Figure 5G needs to be modified. Figure 5S2 shows that 5G is a composite structure derived from two different samples. One contained ATP and the other did not. Overlaying them is inappropriate. Yes, the angle of the head-proximal coiled-coils is similar (Figure 5F vs. 5G), and it is likely that the remainder of the CC density beyond (hinge-proximal) the joint is similar/identical, but this is not proven by the data in the manuscript. In fact, it appears this section of the complex was excluded from analysis in the case of the sample containing nucleotide (Figure 5G and 5S2D). Without recalculating structures or even remaking complicated figures, it must be possible to present this data more clearly. One possibility: simply removing the "Arm" density from figure 5G. Another possibility is to compare the head domain density (including head-proximal cc) from Figure 5F, which is currently obscured by Scc2 density, with the head domain reconstruction from 5G.

Figure 5 – supplement 1A does not conclusively show complexes lacking Scc2. Compare with 5S1A, for example. One needs to consider variability in Scc2 occupancy, Scc2 position, and also the fraction and quality of the particle images contributing to these classes. The description "in samples lacking all HAWK proteins" (line 537) therefore overstates the certainty of the claim given the evidence. The statement and figure panel could be removed without damaging the overall message. Removing panel A has the added benefit that it eliminates the need to discuss a dataset not referenced in figure 5S2 (cohesin, Scc2, ATP).

**Author response:**

*Reviewer #1:*

*Comments for the authors:*

*1. The authors should considering shortening the section on RSC describing Figure S3 as it detracts from the main message of the manuscript. In addition, while the ChIP-seq in Figure S3D argue against a major role for RSC in cohesin loading, the nucleosome mapping data in Figure S3S are not conclusive, as presented. Why do the upper two and lower two nucleosome profiles look different from each other? Potentially effects on nucleosome positioning will only be visible in metagene analysis: the sth1-3 mutant at least would be expected to alter nucleosome positioning but this is not apparent from the data presented. The authors should either remove this data or perform additional analysis.*

We have removed the nucleosome positioning data in Figure S3G and amended the text accordingly.

*2. Figure 7G: The conclusion that cohesin folding at the elbow is a feature of cohesin complexes that are holding sister chromatids together relies on the assumption that all acetylated cohesin is participating in cohesion. However, this may not be the case. In mammals, cohesin acetylation is also observed in G1, indicating that it may have a more general role. This potential caveat should be mentioned.*

Our work is exclusively about what happens in yeast, where acetylation is linked to replication, where sororin is absent, and where acetylation (not sororin association) is the best (known) marker for cohesion. The fact that acetylation is clearly not a good marker for cohesion in mammalian cells does not therefore merit a “caveat”. As far as we know, acetylation is as good a marker in yeast as is sororin association in mammalian cells. The reviewer is correct that this should be pointed out and we have altered the text accordingly.

*3. It is surprising that there is high Scc1 ChIP-seq signal at centromeres in scc4-4 cells if Scc4 is required for cohesin loading at centromeres (Figure S4A). How do the authors explain this?*

It has been previously documented that there is an Scc4-independent mechanism that promotes centromeric loading in the presence of nocodazole (Petela et al. 2018). As this experiment was done in the presence of nocodazole, this mechanism is likely responsible for the high Scc1 ChIP-seq signal observed here.

*4. Line 273: It looks like smc1D588Y CAN rescue the proliferation defect of scc2-4 in Figure S1E.*

This is quite right, and the text has been amended accordingly.

*5. Figure 3A please show proliferation data for the different histone mutants.*

We show proliferation data for the mutant that we use for all subsequent experiments in Figure 3A. This mutant was made de novo and is a good representative for the other mutants. We think showing additional proliferation data for all the mutants is therefore unnecessary.

*6. Line 402: Smc3 acetylation*

Corrected, many thanks.

*7. Supplementary Figure 4F: needs a negative control without cysteine substitutions.*

In line with comments from the other reviewers, we agree this panel is not helpful and not needed and therefore have removed it and amended the text accordingly.

*8. Page 19: Line 547: Call to Figure S4D is incorrect.*

Corrected, many thanks again.

*Reviewer #2:*

*The conclusions are mostly well supported by the data, but some aspects of the biochemical and structural data require clarification.*

*1. Biochemical data: Figure 4B: The FLAG blot indicates that x-linked species migrate at different positions on SDS-PAGE while the top panel shows uniform mobility. How do we know that the band labeled SMC1-? corresponds to x-linked SMC1 and not some non-specific anti-myc background (e.g. there is an additional non-labeled band running lower)?*

The Myc blot shows multiple crosslinked species corresponding to Smc1 crosslinking to multiple proteins. As can be seen from the FLAG blot these species migrate at slightly different positions and have different efficiencies. The most efficient crosslink by far is to Pds5, corresponding to the intense band labelled “Smc1-?” in the Myc blot, which is present in all cases and does mask the other species. It should be noted that this band runs slightly higher in the Pds5-FLAG lane, due to the tag increasing the molecular weight. This crosslinking species has been documented previously, with additional controls, in Bürmann et al., 2019.

We do understand however that our labelling of the blot may have been misleading and so have tried to improve the clarity by amending the “Smc1-?” label to reflect the multiple species present.

*Figure 4B: Different amounts of FLAG-HAWKs are recovered in the anti-Myc IP. Can the authors generate an 'input' blot showing expression levels to exclude the possibility that the variable crosslinking efficiency is not due to different expression levels?*

We agree this would be useful and have modified the figure panel to include the expression levels of Scc2, Scc3 and Pds5 in whole cell extracts relative to a loading control. Indeed, the proteins are expressed to different levels as expected but this does not correlate to the crosslinking efficiency. Pds5 for example is not as abundant as Scc3 but crosslinks much more efficiently.

*2. Figure S4F: Show a WT negative control (no Cys mutation) to confirm that the bands observed do not arise due to non-specific background cross-linking.*

As detailed above, this figure panel has been removed and the text amended accordingly.

*3. Figure 4G: Top: Why is there a crosslinked band in the lane containing Smc1K620C alone? Would it not be expected to see cross-linking only when both Scc2N200C and Smc1K620C are present? Why is the non-specific crosslinking (labeled as *) present everywhere? Which cross-linked band they are referring to? While there is a small amount of a band visible above the non-specific band, this weak band is also present in Smc1K620C alone!*

The crosslinking band observed in the lane containing only Smc1K620C is most likely due to Smc1K620C crosslinking to a natural cysteine in Scc2, probably C224 which sits on a small helix just under that containing N200. This would explain the very similar molecular weight, the fact the band also appears in the Myc blot and the increased intensity of the band on addition of Scc2N200C. The text has been modified to reflect this. We do not know the identity of the non-specific crosslinking band present in the Myc blot but it is clearly independent of either cysteine mutations as it is observed in lane 1. To improve the clarity, we have amended the labelling.

*4. Structural data: Figure 5A: The authors need to clarify if in their cryoEM structure, the SMC ATPase heads are in the 'apo' state, as indicated in the text or in the engaged state as indicated in the Discussion (L. 641).*

Changed. Thank you, this was a typo that has now been corrected.

*5. The authors need to explicitly state in their description of their structural data (Line 512 et seq) that they used a crosslinked version of the cohesin trimer containing 6 Cysteines, positioned at the SMC-SCC1 and SMC1-3 ATPase head heterodimerization interfaces.*

Done, but please bear in mind, as below, that the 6-cysteine version used interrogates the hinge dimerisation and not the head dimerisation.

*6. Crosslinking would be expected to stabilize the SMC ATP heads in close proximity, potentially an engaged state. Would full SMC head disengagement be prevented by such cross-linking even in the absence of nucleotide?*

See last point: we did not cross-link the ATP head dimerisation interface and therefore would not expect head (dis)engagement to be affected in any way.

*7. From Figure 5 the state of the heads is not immediately apparent. Please include a Figure comparing the state of the 'disengaged' SMC heads obtained using their cross-linking method with that of the ATP engaged state published previously.*

Again, the cross-linking does not affect the head engagement of the molecule and at this resolution it would seem somewhat over-interpretative to analyse the precise placement of the head subunits.

*8. If indeed, crosslinking stabilizes an engaged but nucleotide-free form, the authors need explicitly discuss the potential implications. For example, crosslinking of the heads could prevent the heads from properly disengaging. This has potential implications for the conformation/interactions of Scc2 or PDS5.*

Again, see above. The head dimerisation domains were not cross-linked.

*9. Figure 5G: The authors need to clarify how they obtain the ATP-bound state. Did they also use a crosslinked cohesin trimer, hydrolysis-impaired mutant ATPase heads or non-hydrolyzable ATP variants?*

As described, the ATP-bound state was simply achieved by incubating the trimer with ATP. For clarification we have added that no cross-linker was used.

*10. What is the distance between Smc1D588 and SCC2? Figure 5B indicates ~25Å which would be too far away for a direct interaction. In contrast, Figure 5E and their text description suggest that 'folding of the coiled coil brings the hinge to within 12nm (they probably mean Å?) of the heads'. Figure 5E indicates a distance of 12Å but it is between Smc1K620BPA and Smc1D588Y and not SCC2.*

As described in Figure 5B, the distance between Scc2 and Smc1D588 is indeed 25 Å. This refers to the distance of the best cryo-EM map reconstruction (which by definition is an average) and therefore only represents a single snapshot of the entire range of Scc2’s movements above the neck as mentioned in the manuscript. The distance is supposed to highlight this flexibility when comparing it to the hinge-bound structures published previously (e.g. Shi et al. 2020).

We apologise if this was unclear and have reworded the following to aid with clarity:

“Initial 2D classes of Scc2-bound cohesin revealed (Figure S5B) floppiness not only within the HAWK, especially in its C-terminus, but also between the joint and the ATPase heads.”

We recognise the phrase 'folding of the coiled coil brings the hinge to within 12 nm of the heads’ is misleading and should have referred to the ATPase heads to be precise and so as to avoid confusion with the HAWK’s head. We have now changed this accordingly

*11. If indeed no direct interaction is apparent, or if the low resolution and flexibility do not allow firm conclusions, they need to indicate this during their discussion of these results.*

Our manuscript never claims to observe a rigid interaction between Scc2 and the hinge – and the 25 Å distance in Figure 5 reaffirms this – but rather that the folding of the elbow is fundamental in allowing for it and that the clear interaction observed in previously published structures must stem from an intrinsic floppiness in the head region of Scc2 that we also observe in our processing.

12. Figure 6: While biochemical data indicate that PDS5 inhibits ATP hydrolysis, the presented structural data does not reveal how this is accomplished. One caveat of using a cross-linked version is that the procedure may prevent access of PDS5 to fully disengaged SMC heads and thus prevent SMC head engagement (in analogy to the role of the condensin YCS4 subunit). The authors need to indicate potential pitfalls of their cross-linking procedure on PDS5 positioning and discuss if the conformation observed is potentially off pathway.

See above. Head engagement should not be affected in any way by the crosslinking as the third pair of cysteines are placed in the hinge and not the heads.

*Comments for the authors:*

1. Figure 4C-F: Please indicate if the experiment has been done once or repeated several times with consistency.

The experiments were performed twice with the same results. This is now stated in the figure legend.

2. Can they indicate if their preparation used for CryoEM is fully or partially BMOE cross-linked (e.g. by SDS-PAGE analysis).

The estimated efficiency of cross-linking is around 20% as previously mentioned in Collier et al., 2020 (also see SDS-PAGE gel in Author response image 1; the total intensity of the SMCs [including the smear] and Scc1 from lane 1 [trimer crosslinking no cysteines] compared to the 6C crosslinking from the top band from lane 8 [trimer crosslinking 6C]). We have now added that to the manuscript as suggested:

“…To investigate this further, we used cryo-EM to determine the structures of the *S. cerevisiae* cohesin trimer (Smc1, Smc3, Scc1, in their 6C version as reported in Collier et al., 2020; specifically cross-linked at the three intermolecular interfaces at an efficiency of 20% (data not shown)) bound to either Scc2 (Figure 5A)”

3. Figure 6: While at lower resolution, it would still be important to show how well the PDB models fit into the cryoEM density map shown in Figure 6A. This would give the reader a sense for how reliable positioning of the PDB models (shown in Figure 6B) are.

Agreed and done.

4. Can they indicate where the previously published (Rowland et al. 2009, Sutani et al. 2009) eco1-1 suppressor mutations of PDS5 are located? Are suppressor mutations located in observed interfaces? Would they be predicted to interfere with PDS5 interaction?

The suppressor mutation cluster around residues 81-89 in PDS5 reported by Rowland et al. would sit at the hinge:Pds5 interaction interface. The rest of the mutations do not reside in any of the other interfaces predicted by our model. However, as we have explained in the manuscript, the density around the N-terminus of Pds5 is not ideal and we would prefer to refrain from making any too detailed conclusions.

5. L.593 et seq: The authors mention that their data are in contrast to 'previous studies with shortened constructs'. Previous work needs to be cited.

We have added a reference to Muir et al., 2020.

6. L280: This should read: 'relative to the position of D574'?

Thank you, this has been corrected.

7. Figure S4E: Cartoon shows SMC3! Do they mean SMC1?

Yes it should read Smc1, this has also been corrected.

8. Details on CryoEM data processing and reconstruction information need to be included.

We have added a detailed overview of cryo-EM data processing and reconstruction workflows for all maps in Supplementary Figure 7.

9. PDB validation reports of the modeled cryoEM structures are missing.

PDB and EMDB validation reports for all the maps have been attached. The accession codes have been added both to the figure legends and the text. The Table of Structures” in Materials and methods details this for each map:

10. It is not clear how the PDB model for the cohesin elbow is derived. No details of CryoEM/Xray data collection are given and PDB validation reports are again missing.

Please be referred to point 8. We have also added more detail in the Methods section about the model building of the cohesin elbow PDB. In addition, we have added Supplementary Table 3 with information about data collection.

11. L. 1029: Spelling error: Katsu Shirahiga

Thank you, this has been corrected.

*Reviewer #3:*

Chemical crosslinking experiments have established that sister chromatid cohesion requires "topological embrace" of sister chromatids within a cohesin ring. Three contacts define the ring: Scc1-N:Smc3-head, Scc1-C:Smc1-head, and Smc1-hinge:Smc3-hinge. The strongest experimental evidence for a specific DNA entry point is that artificial dimerization of the Smc1-hinge:Smc3-hinge interface prevents cohesion establishment (Gruber et al. 2006; Srinivasan et al. 2018). In the current manuscript, Patela et al. present genetic, biochemical, and structural data that indicate that the cohesin hinge interacts with the cohesin loading complex (Scc2/4), that this interaction depends on Scc4, and that mutations in the hinge that appear to enhance the interaction positively affect cohesin's association with chromosomes. Considered together, these data provide new evidence that the cohesin hinge is connected to the loading reaction. The manuscript is really interesting, includes many very clever experiments, and extends previous models for how cohesin works.

There are major issues with the paper that need to be addressed before publication. First, the paper is confusing. It should be simplified so that the main points are clearer, and errors in the text and figures should be corrected. Second, the section about nucleosome remodeling must be rewritten. There is great value in pointing out differences between studies, but these points can be made more succinctly. Third, the descriptions of the structures are insufficient and do not permit comparison with published structures. The reader cannot currently judge the validity of the interpretations presented.

With the exception of some minor updates to the biochemical experiments, the majority of these changes can be made without adding new data. The rewriting and corrections I have requested will nevertheless require significant time and attention. In addition, I have also requested the EM maps and corresponding docked models be made available to reviewers before a preliminary decision is reached. After this is done, I would support publication of a substantially updated version of the current manuscript in eLife.

1. The manuscript contains a collection of interesting observations, all of which address the overall hypothesis that Scc2/4 stimulates cohesion establishment by interacting with the cohesin hinge.

– See below for comments on the structures. The Pds5 structure should be moved to the supplemental material or removed altogether. The new structure is good enough to discern the overall position of Pds5 relative to cohesin, but it does not provide much new reliable information beyond this. It is pertinent in that it confirms competition between Scc2 and Pds5. Finer details like the status of the head domains, the identity of specific contacts, and the relationship to cohesin acetylation/Eco1 binding will have to wait for higher-resolution structures that will ideally be trapped in defined identifiable biochemical states.

We respectfully disagree with the reviewer. The overall position of Pds5 relative to cohesin is of a clear interest to the wider cohesin community as it not only provides the first visualisation of the interaction but starts to hint at the answers to fundamental questions of this interaction. At the same time, it finally allows much needed further biochemical work to interrogate said interaction that would not be possible without it. It also serves as structural confirmation of the biochemical/genetic data reported in this paper and therefore plays an important role in the manuscript.

– Like the Smc1/3 E-to-Q mutants, smc1D588Y appears to slow or stall cohesin at an early/intermediate step in the loading process (Figure 2C-D). The cells are viable, so the stall is not terminal. How do the authors reconcile this observation with their finding that cohesin loading on chromosome arms is actually elevated relative to wild type?

We believe we addressed this point in the discussion “Why does smc1D588Y depress loading at CENs?”

In particular:

“We suggest that clamping is rate limiting along chromosome arms but unclamping is rate limiting at *CEN*s and that this is the reason why *smc1D588Y* enhances arm loading while depressing loading at *CEN*s.”

– Related to the above question: Have the authors explored how the mutation affects engagement with chromosomal DNA analogous to the recent Chappard paper? Are S-K complexes depleted in favor of others? Can they observe so-called E or J heads, and is either enriched in the mutant background? One line 353, the authors stop short of suggesting the smc1D588Y mutant favors the clamped state but imply it may stop the loading reaction at a step after the one stalled by the clamped-favoring EQ mutations. The current data do not conclusively show that the step slowed by smc1D588Y is before or after the step stalled by the EQ mutations. The authors should remove speculation on this point from the Results section.

Although these are valid questions that would be interesting to address, the experiments required to address them are not trivial and we do not think the answers are necessary for this manuscript, which is already quite long. Concerning the second point, we merely point out that the enhanced association of both Scc2 or EQ complexes at CENs caused by D588Y suggests that the mutation does not affect formation of the clamped state at CENs in vivo but some later step in the loading reaction. We do not agree that this is an unjustified speculation.

– The description of results that begins on line 336 is very confusing. Specifically, "This implies that the reduced loading arises…" but the authors say directly above that Smc1D588Y is at 110% (CENs) of WT. Also, the first two sentences of this paragraph use "it," but the referent is not clear. Last, a simple explanation of the phenotype shown in Figure 2 seems to be that smc1D588Y preserves CEN targeting but disrupts translocation to pericentromeres. Arm loading is not disrupted (slightly enhanced). A simpler description of the phenotype would make the paragraph much easier to read and focus the reader on the important question: what accounts for the subtle defects seen in the smc1D588Y mutant?

The sentence now reads: “This implies that the reduced loading *around centromeres* arises not from defective formation of the clamped state *at CENs* by Scc2/4 complexes associated with Ctf19 but from a defect in a subsequent step in the loading/translocation reaction that requires ATP hydrolysis.”

– Line 456 – "Nevertheless, the level of Smc1-Scc2 crosslinked protein was comparable to that of Scc3…" This is an inappropriate comparison. Crosslinking efficiency could be far more efficient if there is a fortuitous lysine on the target (Scc3 or Scc2 here).

The reviewer is of course correct. What we merely implied is that irrespective of such an effect, may or may not be the case, the amount of Smc1-Scc2 crosslinking was comparable to that of Smc1-Scc3 despite Scc2 being less abundant in the cell. This is surely a valid point to make.

– Line 562 – This is a concrete biochemical prediction that is totally testable. Since it is central to the proposed mechanism of suppression, it should be tested using pulldown assays with purified proteins (cohesin tetramer, Scc2/4 or Scc2-C).

This refers to the statement “We suggest that the addition of a bulky amino acid into Smc1 through D588Y may be sufficient to help bind an otherwise floppy Scc2 N-terminal domain, whose interaction with the hinge is normally stabilised by Scc4”. We disagree that this is a testable hypothesis using pulldown assays as we are referring to the stabilisation of a particular conformation of binding not binding at all, which itself was difficult to measure in this way due to the multiple contacts Scc2 makes with the tetramer (Petela et al. 2018). Although this would be good to address, doing so is not trivial.

2. The section describing the RSC-Scc4 interaction needs to be revised. The authors focus on and oversimplify a secondary conclusion from a previous paper (Lopez-Serra et al. 2014), which holds that Scc4 stimulates RSC. The majority of the work and discussion in this and a related paper from the same group (Lopez-Serra et al. 2014 and Munoz et al. 2020) indicate that Scc2/4 localization depends on RSC. The Munoz paper offers a biochemical explanation for this observation but does not follow up on the secondary observation from the first paper that Scc2/4 influences nucleosome positioning.

– The current findings regarding the genome-wide distribution of cohesin in sth1-3, smc1D588Y, or related mutants seem mostly consistent with the results presented in Lopez-Serra and in Munoz. The authors should revise their text to reflect this concordance. In fact, all three studies emphasize the importance of nucleosome-free DNA. This could be pointed out.

We are grateful to the reviewer for pointing out this deficiency in our discussion of the papers from the Uhlmann lab. We now point out in our revised manuscript that they have in fact proposed two, possibly contradictory, hypotheses for how Scc2/4 functions together with RSC. Crucially, both are inconsistent with our finding that sth1-3 causes only a modest if any defect in cohesin loading in vivo. We would also like to point out that the conclusion that Scc4 acts by stimulating RSC cannot be considered merely a minor secondary conclusion of the Lopez-Serra paper. The very title of their paper was “Scc2/4 acts in sister chromatid cohesion by …maintaining nucleosome regions”. We also respectfully disagree with the reviewer’s statement that the genome-wide distribution of cohesin in sth1-3, smc1D588Y, or related mutants seem mostly consistent with the results presented in Lopez-Serra and in Munoz. We do not believe in papering over the cracks in this manner. There is a clear inconsistency between our calibrated ChIP-seq data and their anecdotal qPCR measurements. We have nevertheless greatly revised our discussion of this issue and hope it that it more accurately describes the claims made by the Uhlmann lab while at the same time being easier to read.

– Any differences in the cohesin distribution between the two studies (Patela and Lopez-Serra), especially in the sth1-3 mutant must be more carefully annotated. Direct comparisons of the original studied loci (Lopez-Serra et al. Figure 3B, for example) must be presented if the authors wish to question the previous findings.

We have now included comparisons of the original studied loci.

– That artificial chromosome tethering of Scc2-C rescues scc4D (Munoz Figure 6B) suggests the partially Scc4-dependent hinge interaction described here is not an essential function of Scc4. The authors suggest that this Scc4 function is rendered irrelevant when the loader is brought near DNA. At this point, there is not enough information to discriminate between these very similar models, and it would be better not to set the current manuscript up in opposition to previous work when an experimental distinction has not been made.

We have now revised this aspect of the discussion.

– Was the MNase-seq calibrated? Also, it looks like there are differences in nucleosome occupancy around AST2, which was also highlighted in the Lopez-Serra work (Figure S3G here). How are the two WT panels different? They show very different results.

In line with comments from the other reviewers we have removed this data and amended the text accordingly.

– The authors say that RSC is not uniquely required for Scc2/4 recruitment, but Figure 3A from Lopez-Serra et al. shows that sth1-3 produces a pronounced cohesion defect, and other chromatin modifying enzyme mutants do not.

We do not discuss the involvement of nucleosome remodellers in sister chromatid cohesion. The observations we discuss relate only to the occupancy of cohesin on DNA and cannot distinguish cohesive and non-cohesive complexes. We note that there may be a greater defect in Smc3 acetylation in sth1-3 mutants than there is a defect in cohesin loading (Figure S3C), which might conceivably be accompanied by cohesion defects. We did not follow this up as we were concerned with cohesin loading and not with the process of cohesion establishment and besides which it was not our goal to re-investigate the role of RSC in this process. That said, this section has been rewritten.

– Line 413 -The double mutant experiment (smc1D588Y sth1-3) is not very informative. It's unclear to me that suppression (or not) of the RSC phenotype by any of the mutants described in the current work would prove the previous sth1-3 findings to be unreliable. In addition, it would be important to compare cohesion, which is not done here. The double mutant experiments can be included, but the conclusions drawn are very limited and need to be rewritten.

This has been re-written to make it clear that even if there is a modest defect in loading in sth1 mutants, it is not altered by smc1D588Y, which bypasses the requirement for Scc4. We hope our logic is now clearer.

– Line 416 – "We were also unable to reproduce the reported effects of scc4-4 and sth1-3 on nucleosome positioning…" If the authors really wish to make this statement/comparison, they need to be much more deliberate about evaluating the reported effects and then comparing their own data. There are clearly differences in Figure S3G, but the authors do not look at them with sufficient granularity. In particular, the Lopez-Serra paper looks at Scc2-bound genes/promoters. The authors must also compare their data with a more recent report (Kubik…Shore, Mol Cell, 2018). This study shows effects specifically at the +1 nucleosome of transcribed genes and is much more in line with the Lopez-Serra observations.

In line with the comments from the other reviewers, we have removed this data and amended the text accordingly.

3. The new structures presented in the paper are compelling. There are several issues that must be cleared up before the manuscript can be published.

– It is unclear how the structures were determined. The authors should present data processing flowcharts to show how the most important structures in the paper were generated. It would be very helpful to know whether the structures described are major or minor components of the samples that were analyzed (relative to other determine structures). Therefore, particle numbers should be included for 3D classification steps, and if any other medium-high resolution structures came from the same datasets, those should be included in the chart for the benefit of the reader. If the authors wish not to disclose unpublished structures, they may elect to exclude some of these, but the overall breakdown of the data needs to be clear. If other structures produced by the dataset are already published, this is fine.

We have addressed all of these points. Please see responses to points 8, 9, and 10 of reviewer #2.

– Data collection tables need to be included, as do tables describing any "pseudo-atomic models." This is especially true for structures that will be deposited in the PDB. It looks like 5F, 5G, and 6A show structures that should be treated this way. For parts the authors do not wish to model explicitly (low-resolution parts corresponding to Scc2/Pds5), they can exclude these from the PDB and carry out model-map correlation calculations using maps truncated to the modeled residues.

See above. We have also added a fit of the three fitted maps used in Figures 5 and 6 as requested in the form of Supplementary Figure 6.

– 2D classification results can be unreliable, especially when there are flexible modules. The analysis shown in Figure S4A-C should be removed, and the text should be modified so that any conclusions made from these images are removed. In particular, the consequences of ATP binding, discussed in relation to Figure S4C is inappropriate given that (1) the authors cannot resolve nucleotide with these images (is it really bound? hydrolyzed? Were there other classes in which the opposite was true?); (2) designation of Scc2 density is reasonable but not probative; and (3) it is not clear what the relationship is between the cohesin head domains. Subtle changes in head engagement could make huge differences in the interpretation. In the displayed images, it is not entirely clear that the heads are even engaged (versus "juxtaposed").

We assume the reviewer refers to Figure S5 here and not S4. In that case, we believe that the conclusions made from the Figures S5A-C are all within reason, i.e. the 2D classes clearly show that (1) cohesin folds without HAWKS, (2) that ATP heads are floppy relative to the folded coiled-coil, and (3) that head engagement and Scc2 binding are not enough for coiled-coil unzipping.

However, we have taken the reviewer’s points into consideration and agree that

comparison of (3) with a sample without ATP would make these points even clearer, which is why we have now added this to help clarify these points. They also demonstrate the consequences of ATP binding while lacking the atomic resolution to observe the ATP molecule. We also agree that S5A does not add anything to the manuscript that has not been shown in Figure 5 previously and so have removed it.

We have no reason to believe the heads in S5C are juxtaposed as the changes are a clear consequence of the presence of ATP in the sample, and that the removal of Scc2 allows us to produce maps that show clear engagement like that of Figure 5G.

*– How confident are the authors in their positioning of the Pds5 module and cohesin head domains in Figure 6A? The density below the "joint" as drawn is very difficult to interpret and looks like it might be artifactual.*

We are very confident in the positioning of the Pds5 model and refer the reviewer to the provided maps to have a look for themselves. The resolution, although not high enough for any detailed modelling, really does only allow a single pose of Pds5 and the heads.

There is no reason to believe this density is artefactual as none of the processing had included neither the atomic structure of Pds5 nor the heads as an initial model.

*– The authors should compare the cohesin head domain conformations they observe, especially for the Scc2 complex, to those observed in the recent structures showing cohesin-nucleotide-Scc3-Scc2-DNA interactions. Are the current structures well enough resolved to make these comparisons? Can the authors see Scc1-N:Smc3-head domain contact?*

Please see below. Figure S5D does this by comparing the binding pose of Scc2 of the ES/EK state with that of our manuscript.

We can indeed see both the Scc1-N:Smc1 and Scc1-C:Smc3 contacts at lower thresholds.

*– Do the authors truly know the cohesin structure displayed in 5G is ATP-bound? To make this statement, they would need to resolve nucleotide and a connected γ phosphate.*

We are confident that Figure 5G shows an ATP-bound map. The structure could only be found in the presence of ATP – a molecule that binds the ATPase heads and causes engagement – and it perfectly recapitulates the crystal structures of engaged heads bound to ATP. In light of this it would seem unreasonable to us to assume that the structure is in an *apo* state.

*– Why did the authors leave out Scc3 from the cryo-EM samples? How do they relate their observations to the published structures, which show the hinge attached to Scc3/Psc3 and dislocated from the Smc1/3 head domains? Both conformations probably inform on mechanism, but the reader is not presented with a helpful way to think about this. Does Scc3 "grab" the hinge after head-Scc2 engagement, as proposed by Uhlmann's group recently (Higashi et al. 2020)? To be sure, Srinivasan et al. proposed an Scc3-dependent late step in loading already, so both of these could easily be referenced in a paragraph contextualizing the current structures with respect to Scc3.*

*We were looking at the putative simultaneous interaction of single HAWK proteins with both hinge and heads and therefore wanted to avoid the addition of another confounding variable in the form of Scc3. We do not present any structural data regarding Scc3 in our manuscript and therefore have abstained from making any conclusions about its potential “grabbing mechanism” as we cannot support it with evidence.*

*4. There are disagreements between data and text and confusing inconsistencies in the figures:*

*o S1C: why is Y40A not an scc4-ts allele?*

Because it grows at the restrictive temperature of 37°C (Figure S1A). We don’t know why Y40A, like Y40N, disrupts co-immunoprecipitation of Scc4 and Scc2 but does not exhibit temperature sensitivity. Presumably, the temperature sensitivity of Y40N cannot solely be due to disruption of Scc2 binding.

*o S1E – It looks like smc1D588Y is an scc2-4 suppressor. The text says the opposite.*

As above, this has been amended.

*o Figure S3 is discussed below.*

See Below.

*o Genotypes should be given in color for all tetrad dissections (as in Figure 1).*

Done.

*o Size markers are missing from the blots. These are important for the crosslinking experiments so that positions can be compared for different antibodies (Figure 4B, for example).*

Size markers are now shown on crosslinking blots.

*o In at least once case, the authors should show a myc blot for Smc1-WT (no Cys mutation) so that it is clear the higher band is indeed an Smc1 crosslinking product.*

This is shown in Lane 1 of Figure 4G.

*o For these assays, the text and legend disagree on which subunit was immobilized for the pulldown (Scc1 or Smc1). If it was different in different experiments, this should be stated.*

This has been corrected, many thanks.

*o Figure S4F is not helpful. There is no control shown (either Scc2 or Smc1, and ideally both, lacking Cys mutation). The crosslinking efficiencies do not seem to correlate for the two subunits, and it is not clear why some Scc2 mutants show an even higher band.*

*In the end, the experiment would have only been valuable if a really great crosslink position were found. The experiment can be removed without seriously damaging the manuscript.*

As described previously, we agree and have removed the figure.

*– The scatter plot in figure S3E does not contradict the Lopez-Serra finding and is not a good way to look at this data. The Lopez-Serra paper addressed a small subset of chromosomal locations. It is not clear how far away from the diagonal the distribution of dots would need to spread to constitute a truly different localization pattern (no control for this).*

The scatterplots have been removed.

– If the authors are sure of their placements for Scc2 and Pds5, then why not directly compare the two structures by overlaying them anchored on the coiled-coiled/hinge domains? This would be a great way of showing that (i) Pds5 is indeed more closely connected to the hinge and (ii) both modules engage the same overall cohesin conformation/fold (or not, as the case may be).

We thank the reviewer for the suggestion and have added a comparison to the Supplemental Figure 5D to show the region of clashes.

– Figure 5F-G make it look like the two structures are different views of the same. Only G is required for the point that head engagement does not unzip the coiled coils, so there is no need to make it look like a comparison.

We believe it to be helpful to show the respective resolutions of each map discussed in the figure, but we agree it may cause some confusion, so we have removed the arrow signalling head engagement.

– Figure S5D does not give the reader enough information to evaluate what is being shown. What dataset does this map come from? Is the corresponding main figure 5A? Was Smc3 included in the fit? DNA? Was nucleotide included in the sample? Why is there a large part of Scc2 poking out (presumably related to mentioned "floppiness")?

Figure S5D intends to show a comparison between our map and the binding pose of Scc2 described for the ES/EK state in Collier et al., 2020. The map originates from the same data set as that of Figure 5A and has been processed to remove the floppy part to produce a map with a higher resolution of the binding of Scc2 to Smc1 to allow for a better comparison. We appreciate that this information was missing from the figure legend and have now included it.

It now reads: “(D) Fitting of atomic map from Collier et al. (6ZZ6) 2020 in cryo-EM map made by focused classification. The map originates from the same data as that of Figure 5A and has been processed to remove the floppy C-terminal head domain of Scc2”.

– Neither the Murayama papers nor the Huis in 't Veld papers are discussed. The Huis in 't Veld paper reports contact between SA1 and the cohesin hinge. Murayama and Uhlmann (2015) reconstitute this interaction (Psm1/3-Psc3) for the fission yeast components and state that it means the ring must bend and that the loader is probably involved. While neither study proved hinge-Scc3 contact is essential for the loading reaction, they were important contributions and should be cited.

Thank you. The papers have now been cited and the following added to the introduction:

“Further, it has been noted that a potential simultaneous interaction of a HAWK with the hinge and kleisin would require some sort of folding (Murayama and Uhlmann, 2015, Huis in 't Veld et al., 2014, Bürmann et al., 2019).”

*– Presentation issues for structures:*

o At least once in each figure (but not necessarily each panel), colored text should be shown for each included protein as is done in Figure 1. This is missing in Figure 5 (for example).

Done.

o The overlay in Figure 1F is hard to understand. The mutated structure should be shaded differently so the reader can see the difference clearly (pale green vs green and same for red).

Done.

*o The nucleosome structure in Figure 3B is impossible to understand. The entire particle should be shown so the reader can see the DNA wrap, which is relevant here (do the mutations enhance unwrapping?). Only the mutated histones and mutated residues (highlight color) need to be colored.*

Done.

[Editors' note: further revisions were suggested prior to acceptance, as described below.]

The manuscript has been improved but there are some remaining issues that need to be addressed, as outlined in the points made by reviewers below. It is particularly important to clarify where the crosslinks reside in the cohesin variant used for structural analysis, as raised in Points 2 and 3 from Reviewer 2. If the crosslinks are in the heads, as indicated in the methods section, then this has important implications for the interpretation.

We thank the reviewers for their keen eye, particularly with reference to the points on head crosslinking which was indeed a mistake on our part in the methods section. Please see our responses to each comment below for more details.

Reviewer #2:

The authors have addressed most comments. There are however a couple of issues that require further clarification:

1. Figure 4G: The author's response is only partially convincing: If Smc1K620C also crosslinks to Scc2C224, it is not fully clear that Scc2N220C specifically allows readout of contacts between the Smc1 hinge and Scc2. As they have tested a variety of Scc2 cysteine substitutions, it would have been important to test a Scc2 version that does not contain C224, in order to avoid such confounding non-specific crosslinking.

In any case, they need to further clarify how the different blots are developed. Presumably, the top panel shows an anti-FLAG western blot and the bottom panel an anti-myc blot (as indicated in the author's response)? The top panel is not labeled at all and the bottom panel is mislabeled as Anti-HA!

The reviewer is quite right, the top panel is anti-FLAG and the bottom is anti-myc. The anti-HA label corresponds the small panel in Figure 4B. We see how this is confusing and have modified the figure to make it clearer.

2. The authors state line 523 that they use a cohesin variant 'containing cysteines specifically crosslinking the three intermolecular interfaces'. They need to explicitly state which interfaces are crosslinked in the variant they use for structure determination. The author response indicates that 'the 6-cysteine version used interrogates the hinge dimerisation and not the head dimerisation'. Their methods section indicates however that the variant they use for structure determination does not contain Cys mutations in the hinges. Instead, they use Smc1G22C N1192C, Smc3S1043C R1222C, Scc1C56A547C.

These mutations are located (1) in the Smc3-NScc1 interface (Scc1C56 and Smc3S1043C), (2) Smc1-cScc1 (Scc1A547C and Smc1G22C) interface and (3), in the interface between the Smc1 and Smc3 ATPase heads (Smc1 N1192C and Smc3 R1222C). That is, the third pair of cysteines is clearly placed in the heads and not the hinges.

We profoundly apologise for the confusion caused by our mistake and thank the reviewer for pointing this out again. The correct cysteines are Smc1K639C-Smc3E570C, Smc1G22C-Scc1A547C, and Smc3S1043C-Scc1C56. We have amended the manuscript accordingly in the respective places. Therefore, our original interpretation stands, i.e. the ATPase heads have not been crosslinked to each other.

3. It is therefore not clear why the authors maintain (Reviewer #2 points 6, 7, 8, 12) that 'we did not cross-link the ATP head dimerisation interface'. Clearly, something is wrong! Maybe it is the methods section? If not, the authors need to clearly indicate that the version of cohesin they use is crosslinked at the SMC1/3 ATPase head domains. The implications of such head crosslinking need to be taken into account in the interpretation of their data and discussion of the results.

See above.

*Reviewer #3:*

*The manuscript is greatly improved. The updated figures make it much clearer. The rewritten section describing Sth1, nucleosome positioning, and histone mutants is good. This is important work, and the breadth and strength of the experiments described are impressive.*

*Most of my comments have been adequately addressed. The requests below relate to the presentation and description of the data and should be addressed before publication but do not require extensive writing or any new experiments.*

*Line 593-597:*

*"In the presence of ATP" implies that ATP occupies both ATPase active sites. Neither γ phosphate can be resolved at this resolution. The structure could be a reaction intermediate resulting from slow π dissociation from one or both sites, for example. This is not to detract from the observations or statements made. Instead, it would be wise to state more clearly that ATP was included in the sample but avoid any suggestion that the structure provides detailed information about the SMC ATPase cycle, which will doubtless be a topic for future studies. This is rightly stated in the final paragraph of the Discussion.*

We assume that the reviewer refers to one or both of these instances: “We identified and solved … a form of cohesin at 6Å … whose ATPase heads were engaged in the presence of ATP” and/or “A cohesin complex whose heads are engaged in the presence of ATP…”

If so, we respectfully disagree with the reviewer. What is precisely stated in our manuscript is that our engaged cohesin heads, which we explicitly mention are at no more than 6 Å resolution, were in the presence of ATP. We never allude to the state of the molecule or even hint at the possibility of seeing the gamma phosphate at this resolution. Adding that the heads could also be bound to ADP and Pi or even be in a transition would just serve to confuse the reader in an, in our opinion, very straightforward and factual statement of the sample conditions.

*Figure 5G needs to be modified. Figure 5S2 shows that 5G is a composite structure derived from two different samples. One contained ATP and the other did not. Overlaying them is inappropriate. Yes, the angle of the head-proximal coiled-coils is similar (Figure 5F vs. 5G), and it is likely that the remainder of the CC density beyond (hinge-proximal) the joint is similar/identical, but this is not proven by the data in the manuscript. In fact, it appears this section of the complex was excluded from analysis in the case of the sample containing nucleotide (Figure 5G and 5S2D). Without recalculating structures or even remaking complicated figures, it must be possible to present this data more clearly. One possibility: simply removing the "Arm" density from figure 5G. Another possibility is to compare the head domain density (including head-proximal cc) from Figure 5F, which is currently obscured by Scc2 density, with the head domain reconstruction from 5G.*

We thank the reviewer for the suggestion and we agree that the figure should be more transparent. We have removed the separating discontinuous line that suggested that these were two structures from different datasets and have added that the folded coiled-coil was derived from a dataset without ATP to the figure. In addition, we have referred the reader explicitly to figure 5S2D to see how the map was obtained. However, we believe that the composite structure does add valuable information that aids the reader’s understanding by putting the engaged ATPase heads into the structural context of a folded cohesion and have therefore decided to leave it with the mentioned remarks. Especially given the 2D classes in Figure 5 Supplement 1E.

*Figure 5 – supplement 1A does not conclusively show complexes lacking Scc2. Compare with 5S1A, for example. One needs to consider variability in Scc2 occupancy, Scc2 position, and also the fraction and quality of the particle images contributing to these classes. The description "in samples lacking all HAWK proteins" (line 537) therefore overstates the certainty of the claim given the evidence. The statement and figure panel could be removed without damaging the overall message. Removing panel A has the added benefit that it eliminates the need to discuss a dataset not referenced in figure 5S2 (cohesin, Scc2, ATP).*

The reviewer is absolutely right – this was due to a mistake while reorganising the supplementary figures 5 during the previous revision. We have added 2D classes showing exactly that, i.e, a folded cohesion complex in the absence of HAWK proteins to the figure (cf. Figure 5S1E) and have amended the text that referred to the wrong figure.

---

## [Author Response]

*Reviewer #1:*

Comments for the authors:1. The authors should considering shortening the section on RSC describing Figure S3 as it detracts from the main message of the manuscript. In addition, while the ChIP-seq in Figure S3D argue against a major role for RSC in cohesin loading, the nucleosome mapping data in Figure S3S are not conclusive, as presented. Why do the upper two and lower two nucleosome profiles look different from each other? Potentially effects on nucleosome positioning will only be visible in metagene analysis: the sth1-3 mutant at least would be expected to alter nucleosome positioning but this is not apparent from the data presented. The authors should either remove this data or perform additional analysis.

We have removed the nucleosome positioning data in Figure S3G and amended the text accordingly.

2. Figure 7G: The conclusion that cohesin folding at the elbow is a feature of cohesin complexes that are holding sister chromatids together relies on the assumption that all acetylated cohesin is participating in cohesion. However, this may not be the case. In mammals, cohesin acetylation is also observed in G1, indicating that it may have a more general role. This potential caveat should be mentioned.

Our work is exclusively about what happens in yeast, where acetylation is linked to replication, where sororin is absent, and where acetylation (not sororin association) is the best (known) marker for cohesion. The fact that acetylation is clearly not a good marker for cohesion in mammalian cells does not therefore merit a “caveat”. As far as we know, acetylation is as good a marker in yeast as is sororin association in mammalian cells. The reviewer is correct that this should be pointed out and we have altered the text accordingly.

3. It is surprising that there is high Scc1 ChIP-seq signal at centromeres in scc4-4 cells if Scc4 is required for cohesin loading at centromeres (Figure S4A). How do the authors explain this?

It has been previously documented that there is an Scc4-independent mechanism that promotes centromeric loading in the presence of nocodazole (Petela et al. 2018). As this experiment was done in the presence of nocodazole, this mechanism is likely responsible for the high Scc1 ChIP-seq signal observed here.

4. Line 273: It looks like smc1D588Y CAN rescue the proliferation defect of scc2-4 in Figure S1E.

This is quite right, and the text has been amended accordingly.

5. Figure 3A please show proliferation data for the different histone mutants.

We show proliferation data for the mutant that we use for all subsequent experiments in Figure 3A. This mutant was made de novo and is a good representative for the other mutants. We think showing additional proliferation data for all the mutants is therefore unnecessary.

6. Line 402: Smc3 acetylation

Corrected, many thanks.

7. Supplementary Figure 4F: needs a negative control without cysteine substitutions.

In line with comments from the other reviewers, we agree this panel is not helpful and not needed and therefore have removed it and amended the text accordingly.

8. Page 19: Line 547: Call to Figure S4D is incorrect.

Corrected, many thanks again.

*Reviewer #2:*

The conclusions are mostly well supported by the data, but some aspects of the biochemical and structural data require clarification.1. Biochemical data: Figure 4B: The FLAG blot indicates that x-linked species migrate at different positions on SDS-PAGE while the top panel shows uniform mobility. How do we know that the band labeled SMC1-? corresponds to x-linked SMC1 and not some non-specific anti-myc background (e.g. there is an additional non-labeled band running lower)?

The Myc blot shows multiple crosslinked species corresponding to Smc1 crosslinking to multiple proteins. As can be seen from the FLAG blot these species migrate at slightly different positions and have different efficiencies. The most efficient crosslink by far is to Pds5, corresponding to the intense band labelled “Smc1-?” in the Myc blot, which is present in all cases and does mask the other species. It should be noted that this band runs slightly higher in the Pds5-FLAG lane, due to the tag increasing the molecular weight. This crosslinking species has been documented previously, with additional controls, in Bürmann et al., 2019.

We do understand however that our labelling of the blot may have been misleading and so have tried to improve the clarity by amending the “Smc1-?” label to reflect the multiple species present.

Figure 4B: Different amounts of FLAG-HAWKs are recovered in the anti-Myc IP. Can the authors generate an 'input' blot showing expression levels to exclude the possibility that the variable crosslinking efficiency is not due to different expression levels?

We agree this would be useful and have modified the figure panel to include the expression levels of Scc2, Scc3 and Pds5 in whole cell extracts relative to a loading control. Indeed, the proteins are expressed to different levels as expected but this does not correlate to the crosslinking efficiency. Pds5 for example is not as abundant as Scc3 but crosslinks much more efficiently.

2. Figure S4F: Show a WT negative control (no Cys mutation) to confirm that the bands observed do not arise due to non-specific background cross-linking.

As detailed above, this figure panel has been removed and the text amended accordingly.

3. Figure 4G: Top: Why is there a crosslinked band in the lane containing Smc1K620C alone? Would it not be expected to see cross-linking only when both Scc2N200C and Smc1K620C are present? Why is the non-specific crosslinking (labeled as *) present everywhere? Which cross-linked band they are referring to? While there is a small amount of a band visible above the non-specific band, this weak band is also present in Smc1K620C alone!

The crosslinking band observed in the lane containing only Smc1K620C is most likely due to Smc1K620C crosslinking to a natural cysteine in Scc2, probably C224 which sits on a small helix just under that containing N200. This would explain the very similar molecular weight, the fact the band also appears in the Myc blot and the increased intensity of the band on addition of Scc2N200C. The text has been modified to reflect this. We do not know the identity of the non-specific crosslinking band present in the Myc blot but it is clearly independent of either cysteine mutations as it is observed in lane 1. To improve the clarity, we have amended the labelling.

4. Structural data: Figure 5A: The authors need to clarify if in their cryoEM structure, the SMC ATPase heads are in the 'apo' state, as indicated in the text or in the engaged state as indicated in the Discussion (L. 641).

Changed. Thank you, this was a typo that has now been corrected.

5. The authors need to explicitly state in their description of their structural data (Line 512 et seq) that they used a crosslinked version of the cohesin trimer containing 6 Cysteines, positioned at the SMC-SCC1 and SMC1-3 ATPase head heterodimerization interfaces.

Done, but please bear in mind, as below, that the 6-cysteine version used interrogates the hinge dimerisation and not the head dimerisation.

6. Crosslinking would be expected to stabilize the SMC ATP heads in close proximity, potentially an engaged state. Would full SMC head disengagement be prevented by such cross-linking even in the absence of nucleotide?

See last point: we did not cross-link the ATP head dimerisation interface and therefore would not expect head (dis)engagement to be affected in any way.

7. From Figure 5 the state of the heads is not immediately apparent. Please include a Figure comparing the state of the 'disengaged' SMC heads obtained using their cross-linking method with that of the ATP engaged state published previously.

Again, the cross-linking does not affect the head engagement of the molecule and at this resolution it would seem somewhat over-interpretative to analyse the precise placement of the head subunits.

8. If indeed, crosslinking stabilizes an engaged but nucleotide-free form, the authors need explicitly discuss the potential implications. For example, crosslinking of the heads could prevent the heads from properly disengaging. This has potential implications for the conformation/interactions of Scc2 or PDS5.

Again, see above. The head dimerisation domains were not cross-linked.

9. Figure 5G: The authors need to clarify how they obtain the ATP-bound state. Did they also use a crosslinked cohesin trimer, hydrolysis-impaired mutant ATPase heads or non-hydrolyzable ATP variants?

As described, the ATP-bound state was simply achieved by incubating the trimer with ATP. For clarification we have added that no cross-linker was used.

10. What is the distance between Smc1D588 and SCC2? Figure 5B indicates ~25Å which would be too far away for a direct interaction. In contrast, Figure 5E and their text description suggest that 'folding of the coiled coil brings the hinge to within 12nm (they probably mean Å?) of the heads'. Figure 5E indicates a distance of 12Å but it is between Smc1K620BPA and Smc1D588Y and not SCC2.

As described in Figure 5B, the distance between Scc2 and Smc1D588 is indeed 25 Å. This refers to the distance of the best cryo-EM map reconstruction (which by definition is an average) and therefore only represents a single snapshot of the entire range of Scc2’s movements above the neck as mentioned in the manuscript. The distance is supposed to highlight this flexibility when comparing it to the hinge-bound structures published previously (e.g. Shi et al. 2020).

We apologise if this was unclear and have reworded the following to aid with clarity:

“Initial 2D classes of Scc2-bound cohesin revealed (Figure S5B) floppiness not only within the HAWK, especially in its C-terminus, but also between the joint and the ATPase heads.”

We recognise the phrase 'folding of the coiled coil brings the hinge to within 12 nm of the heads’ is misleading and should have referred to the ATPase heads to be precise and so as to avoid confusion with the HAWK’s head. We have now changed this accordingly

11. If indeed no direct interaction is apparent, or if the low resolution and flexibility do not allow firm conclusions, they need to indicate this during their discussion of these results.

Our manuscript never claims to observe a rigid interaction between Scc2 and the hinge – and the 25 Å distance in Figure 5 reaffirms this – but rather that the folding of the elbow is fundamental in allowing for it and that the clear interaction observed in previously published structures must stem from an intrinsic floppiness in the head region of Scc2 that we also observe in our processing.

12. Figure 6: While biochemical data indicate that PDS5 inhibits ATP hydrolysis, the presented structural data does not reveal how this is accomplished. One caveat of using a cross-linked version is that the procedure may prevent access of PDS5 to fully disengaged SMC heads and thus prevent SMC head engagement (in analogy to the role of the condensin YCS4 subunit). The authors need to indicate potential pitfalls of their cross-linking procedure on PDS5 positioning and discuss if the conformation observed is potentially off pathway.

See above. Head engagement should not be affected in any way by the crosslinking as the third pair of cysteines are placed in the hinge and not the heads.

Comments for the authors:1. Figure 4C-F: Please indicate if the experiment has been done once or repeated several times with consistency.

The experiments were performed twice with the same results. This is now stated in the figure legend.

2. Can they indicate if their preparation used for CryoEM is fully or partially BMOE cross-linked (e.g. by SDS-PAGE analysis).

The estimated efficiency of cross-linking is around 20% as previously mentioned in Collier et al., 2020 (also see SDS-PAGE gel in Author response image 1; the total intensity of the SMCs [including the smear] and Scc1 from lane 1 [trimer crosslinking no cysteines] compared to the 6C crosslinking from the top band from lane 8 [trimer crosslinking 6C]). We have now added that to the manuscript as suggested:

“…To investigate this further, we used cryo-EM to determine the structures of the *S. cerevisiae* cohesin trimer (Smc1, Smc3, Scc1, in their 6C version as reported in Collier et al., 2020; specifically cross-linked at the three intermolecular interfaces at an efficiency of 20% (data not shown)) bound to either Scc2 (Figure 5A)”

**Author response image 1. sa1fig1:** 

3. Figure 6: While at lower resolution, it would still be important to show how well the PDB models fit into the cryoEM density map shown in Figure 6A. This would give the reader a sense for how reliable positioning of the PDB models (shown in Figure 6B) are.

Agreed and done.

4. Can they indicate where the previously published (Rowland et al. 2009, Sutani et al. 2009) eco1-1 suppressor mutations of PDS5 are located? Are suppressor mutations located in observed interfaces? Would they be predicted to interfere with PDS5 interaction?

The suppressor mutation cluster around residues 81-89 in PDS5 reported by Rowland et al. would sit at the hinge:Pds5 interaction interface. The rest of the mutations do not reside in any of the other interfaces predicted by our model. However, as we have explained in the manuscript, the density around the N-terminus of Pds5 is not ideal and we would prefer to refrain from making any too detailed conclusions.

5. L.593 et seq: The authors mention that their data are in contrast to 'previous studies with shortened constructs'. Previous work needs to be cited.

We have added a reference to Muir et al., 2020.

6. L280: This should read: 'relative to the position of D574'?

Thank you, this has been corrected.

7. Figure S4E: Cartoon shows SMC3! Do they mean SMC1?

Yes it should read Smc1, this has also been corrected.

8. Details on CryoEM data processing and reconstruction information need to be included.

We have added a detailed overview of cryo-EM data processing and reconstruction workflows for all maps in Supplementary Figure 7.

9. PDB validation reports of the modeled cryoEM structures are missing.

PDB and EMDB validation reports for all the maps have been attached. The accession codes have been added both to the figure legends and the text. The Table of Structures” in Materials and methods details this for each map:

10. It is not clear how the PDB model for the cohesin elbow is derived. No details of CryoEM/Xray data collection are given and PDB validation reports are again missing.

Please be referred to point 8. We have also added more detail in the Methods section about the model building of the cohesin elbow PDB. In addition, we have added Supplementary Table 3 with information about data collection.

11. L. 1029: Spelling error: Katsu Shirahiga

Thank you, this has been corrected.

*Reviewer #3:*

Chemical crosslinking experiments have established that sister chromatid cohesion requires "topological embrace" of sister chromatids within a cohesin ring. Three contacts define the ring: Scc1-N:Smc3-head, Scc1-C:Smc1-head, and Smc1-hinge:Smc3-hinge. The strongest experimental evidence for a specific DNA entry point is that artificial dimerization of the Smc1-hinge:Smc3-hinge interface prevents cohesion establishment (Gruber et al. 2006; Srinivasan et al. 2018). In the current manuscript, Patela et al. present genetic, biochemical, and structural data that indicate that the cohesin hinge interacts with the cohesin loading complex (Scc2/4), that this interaction depends on Scc4, and that mutations in the hinge that appear to enhance the interaction positively affect cohesin's association with chromosomes. Considered together, these data provide new evidence that the cohesin hinge is connected to the loading reaction. The manuscript is really interesting, includes many very clever experiments, and extends previous models for how cohesin works.There are major issues with the paper that need to be addressed before publication. First, the paper is confusing. It should be simplified so that the main points are clearer, and errors in the text and figures should be corrected. Second, the section about nucleosome remodeling must be rewritten. There is great value in pointing out differences between studies, but these points can be made more succinctly. Third, the descriptions of the structures are insufficient and do not permit comparison with published structures. The reader cannot currently judge the validity of the interpretations presented.With the exception of some minor updates to the biochemical experiments, the majority of these changes can be made without adding new data. The rewriting and corrections I have requested will nevertheless require significant time and attention. In addition, I have also requested the EM maps and corresponding docked models be made available to reviewers before a preliminary decision is reached. After this is done, I would support publication of a substantially updated version of the current manuscript in eLife.

1. The manuscript contains a collection of interesting observations, all of which address the overall hypothesis that Scc2/4 stimulates cohesion establishment by interacting with the cohesin hinge.

– See below for comments on the structures. The Pds5 structure should be moved to the supplemental material or removed altogether. The new structure is good enough to discern the overall position of Pds5 relative to cohesin, but it does not provide much new reliable information beyond this. It is pertinent in that it confirms competition between Scc2 and Pds5. Finer details like the status of the head domains, the identity of specific contacts, and the relationship to cohesin acetylation/Eco1 binding will have to wait for higher-resolution structures that will ideally be trapped in defined identifiable biochemical states.

We respectfully disagree with the reviewer. The overall position of Pds5 relative to cohesin is of a clear interest to the wider cohesin community as it not only provides the first visualisation of the interaction but starts to hint at the answers to fundamental questions of this interaction. At the same time, it finally allows much needed further biochemical work to interrogate said interaction that would not be possible without it. It also serves as structural confirmation of the biochemical/genetic data reported in this paper and therefore plays an important role in the manuscript.

– Like the Smc1/3 E-to-Q mutants, smc1D588Y appears to slow or stall cohesin at an early/intermediate step in the loading process (Figure 2C-D). The cells are viable, so the stall is not terminal. How do the authors reconcile this observation with their finding that cohesin loading on chromosome arms is actually elevated relative to wild type?

We believe we addressed this point in the discussion “Why does smc1D588Y depress loading at CENs?”

In particular:

“We suggest that clamping is rate limiting along chromosome arms but unclamping is rate limiting at *CEN*s and that this is the reason why *smc1D588Y* enhances arm loading while depressing loading at *CEN*s.”

– Related to the above question: Have the authors explored how the mutation affects engagement with chromosomal DNA analogous to the recent Chappard paper? Are S-K complexes depleted in favor of others? Can they observe so-called E or J heads, and is either enriched in the mutant background? One line 353, the authors stop short of suggesting the smc1D588Y mutant favors the clamped state but imply it may stop the loading reaction at a step after the one stalled by the clamped-favoring EQ mutations. The current data do not conclusively show that the step slowed by smc1D588Y is before or after the step stalled by the EQ mutations. The authors should remove speculation on this point from the Results section.

Although these are valid questions that would be interesting to address, the experiments required to address them are not trivial and we do not think the answers are necessary for this manuscript, which is already quite long. Concerning the second point, we merely point out that the enhanced association of both Scc2 or EQ complexes at CENs caused by D588Y suggests that the mutation does not affect formation of the clamped state at CENs in vivo but some later step in the loading reaction. We do not agree that this is an unjustified speculation.

– The description of results that begins on line 336 is very confusing. Specifically, "This implies that the reduced loading arises…" but the authors say directly above that Smc1D588Y is at 110% (CENs) of WT. Also, the first two sentences of this paragraph use "it," but the referent is not clear. Last, a simple explanation of the phenotype shown in Figure 2 seems to be that smc1D588Y preserves CEN targeting but disrupts translocation to pericentromeres. Arm loading is not disrupted (slightly enhanced). A simpler description of the phenotype would make the paragraph much easier to read and focus the reader on the important question: what accounts for the subtle defects seen in the smc1D588Y mutant?

The sentence now reads: “This implies that the reduced loading *around centromeres* arises not from defective formation of the clamped state *at CENs* by Scc2/4 complexes associated with Ctf19 but from a defect in a subsequent step in the loading/translocation reaction that requires ATP hydrolysis.”

– Line 456 – "Nevertheless, the level of Smc1-Scc2 crosslinked protein was comparable to that of Scc3…" This is an inappropriate comparison. Crosslinking efficiency could be far more efficient if there is a fortuitous lysine on the target (Scc3 or Scc2 here).

The reviewer is of course correct. What we merely implied is that irrespective of such an effect, may or may not be the case, the amount of Smc1-Scc2 crosslinking was comparable to that of Smc1-Scc3 despite Scc2 being less abundant in the cell. This is surely a valid point to make.

– Line 562 – This is a concrete biochemical prediction that is totally testable. Since it is central to the proposed mechanism of suppression, it should be tested using pulldown assays with purified proteins (cohesin tetramer, Scc2/4 or Scc2-C).

This refers to the statement “We suggest that the addition of a bulky amino acid into Smc1 through D588Y may be sufficient to help bind an otherwise floppy Scc2 N-terminal domain, whose interaction with the hinge is normally stabilised by Scc4”. We disagree that this is a testable hypothesis using pulldown assays as we are referring to the stabilisation of a particular conformation of binding not binding at all, which itself was difficult to measure in this way due to the multiple contacts Scc2 makes with the tetramer (Petela et al. 2018). Although this would be good to address, doing so is not trivial.

2. The section describing the RSC-Scc4 interaction needs to be revised. The authors focus on and oversimplify a secondary conclusion from a previous paper (Lopez-Serra et al. 2014), which holds that Scc4 stimulates RSC. The majority of the work and discussion in this and a related paper from the same group (Lopez-Serra et al. 2014 and Munoz et al. 2020) indicate that Scc2/4 localization depends on RSC. The Munoz paper offers a biochemical explanation for this observation but does not follow up on the secondary observation from the first paper that Scc2/4 influences nucleosome positioning.– The current findings regarding the genome-wide distribution of cohesin in sth1-3, smc1D588Y, or related mutants seem mostly consistent with the results presented in Lopez-Serra and in Munoz. The authors should revise their text to reflect this concordance. In fact, all three studies emphasize the importance of nucleosome-free DNA. This could be pointed out.

We are grateful to the reviewer for pointing out this deficiency in our discussion of the papers from the Uhlmann lab. We now point out in our revised manuscript that they have in fact proposed two, possibly contradictory, hypotheses for how Scc2/4 functions together with RSC. Crucially, both are inconsistent with our finding that sth1-3 causes only a modest if any defect in cohesin loading in vivo. We would also like to point out that the conclusion that Scc4 acts by stimulating RSC cannot be considered merely a minor secondary conclusion of the Lopez-Serra paper. The very title of their paper was “Scc2/4 acts in sister chromatid cohesion by …maintaining nucleosome regions”. We also respectfully disagree with the reviewer’s statement that the genome-wide distribution of cohesin in sth1-3, smc1D588Y, or related mutants seem mostly consistent with the results presented in Lopez-Serra and in Munoz. We do not believe in papering over the cracks in this manner. There is a clear inconsistency between our calibrated ChIP-seq data and their anecdotal qPCR measurements. We have nevertheless greatly revised our discussion of this issue and hope it that it more accurately describes the claims made by the Uhlmann lab while at the same time being easier to read.

– Any differences in the cohesin distribution between the two studies (Patela and Lopez-Serra), especially in the sth1-3 mutant must be more carefully annotated. Direct comparisons of the original studied loci (Lopez-Serra et al. Figure 3B, for example) must be presented if the authors wish to question the previous findings.

We have now included comparisons of the original studied loci.

– That artificial chromosome tethering of Scc2-C rescues scc4D (Munoz Figure 6B) suggests the partially Scc4-dependent hinge interaction described here is not an essential function of Scc4. The authors suggest that this Scc4 function is rendered irrelevant when the loader is brought near DNA. At this point, there is not enough information to discriminate between these very similar models, and it would be better not to set the current manuscript up in opposition to previous work when an experimental distinction has not been made.

We have now revised this aspect of the discussion.

– Was the MNase-seq calibrated? Also, it looks like there are differences in nucleosome occupancy around AST2, which was also highlighted in the Lopez-Serra work (Figure S3G here). How are the two WT panels different? They show very different results.

In line with comments from the other reviewers we have removed this data and amended the text accordingly.

– The authors say that RSC is not uniquely required for Scc2/4 recruitment, but Figure 3A from Lopez-Serra et al. shows that sth1-3 produces a pronounced cohesion defect, and other chromatin modifying enzyme mutants do not.

We do not discuss the involvement of nucleosome remodellers in sister chromatid cohesion. The observations we discuss relate only to the occupancy of cohesin on DNA and cannot distinguish cohesive and non-cohesive complexes. We note that there may be a greater defect in Smc3 acetylation in sth1-3 mutants than there is a defect in cohesin loading (Figure S3C), which might conceivably be accompanied by cohesion defects. We did not follow this up as we were concerned with cohesin loading and not with the process of cohesion establishment and besides which it was not our goal to re-investigate the role of RSC in this process. That said, this section has been rewritten.

– Line 413 -The double mutant experiment (smc1D588Y sth1-3) is not very informative. It's unclear to me that suppression (or not) of the RSC phenotype by any of the mutants described in the current work would prove the previous sth1-3 findings to be unreliable. In addition, it would be important to compare cohesion, which is not done here. The double mutant experiments can be included, but the conclusions drawn are very limited and need to be rewritten.

This has been re-written to make it clear that even if there is a modest defect in loading in sth1 mutants, it is not altered by smc1D588Y, which bypasses the requirement for Scc4. We hope our logic is now clearer.

– Line 416 – "We were also unable to reproduce the reported effects of scc4-4 and sth1-3 on nucleosome positioning…" If the authors really wish to make this statement/comparison, they need to be much more deliberate about evaluating the reported effects and then comparing their own data. There are clearly differences in Figure S3G, but the authors do not look at them with sufficient granularity. In particular, the Lopez-Serra paper looks at Scc2-bound genes/promoters. The authors must also compare their data with a more recent report (Kubik…Shore, Mol Cell, 2018). This study shows effects specifically at the +1 nucleosome of transcribed genes and is much more in line with the Lopez-Serra observations.

In line with the comments from the other reviewers, we have removed this data and amended the text accordingly.

3. The new structures presented in the paper are compelling. There are several issues that must be cleared up before the manuscript can be published.– It is unclear how the structures were determined. The authors should present data processing flowcharts to show how the most important structures in the paper were generated. It would be very helpful to know whether the structures described are major or minor components of the samples that were analyzed (relative to other determine structures). Therefore, particle numbers should be included for 3D classification steps, and if any other medium-high resolution structures came from the same datasets, those should be included in the chart for the benefit of the reader. If the authors wish not to disclose unpublished structures, they may elect to exclude some of these, but the overall breakdown of the data needs to be clear. If other structures produced by the dataset are already published, this is fine.

We have addressed all of these points. Please see responses to points 8, 9, and 10 of reviewer #2.

– Data collection tables need to be included, as do tables describing any "pseudo-atomic models." This is especially true for structures that will be deposited in the PDB. It looks like 5F, 5G, and 6A show structures that should be treated this way. For parts the authors do not wish to model explicitly (low-resolution parts corresponding to Scc2/Pds5), they can exclude these from the PDB and carry out model-map correlation calculations using maps truncated to the modeled residues.

See above. We have also added a fit of the three fitted maps used in Figures 5 and 6 as requested in the form of Supplementary Figure 6.

– 2D classification results can be unreliable, especially when there are flexible modules. The analysis shown in Figure S4A-C should be removed, and the text should be modified so that any conclusions made from these images are removed. In particular, the consequences of ATP binding, discussed in relation to Figure S4C is inappropriate given that (1) the authors cannot resolve nucleotide with these images (is it really bound? hydrolyzed? Were there other classes in which the opposite was true?); (2) designation of Scc2 density is reasonable but not probative; and (3) it is not clear what the relationship is between the cohesin head domains. Subtle changes in head engagement could make huge differences in the interpretation. In the displayed images, it is not entirely clear that the heads are even engaged (versus "juxtaposed").

We assume the reviewer refers to Figure S5 here and not S4. In that case, we believe that the conclusions made from the Figures S5A-C are all within reason, i.e. the 2D classes clearly show that (1) cohesin folds without HAWKS, (2) that ATP heads are floppy relative to the folded coiled-coil, and (3) that head engagement and Scc2 binding are not enough for coiled-coil unzipping.

However, we have taken the reviewer’s points into consideration and agree that

comparison of (3) with a sample without ATP would make these points even clearer, which is why we have now added this to help clarify these points. They also demonstrate the consequences of ATP binding while lacking the atomic resolution to observe the ATP molecule. We also agree that S5A does not add anything to the manuscript that has not been shown in Figure 5 previously and so have removed it.

We have no reason to believe the heads in S5C are juxtaposed as the changes are a clear consequence of the presence of ATP in the sample, and that the removal of Scc2 allows us to produce maps that show clear engagement like that of Figure 5G.

– How confident are the authors in their positioning of the Pds5 module and cohesin head domains in Figure 6A? The density below the "joint" as drawn is very difficult to interpret and looks like it might be artifactual.

We are very confident in the positioning of the Pds5 model and refer the reviewer to the provided maps to have a look for themselves. The resolution, although not high enough for any detailed modelling, really does only allow a single pose of Pds5 and the heads.

There is no reason to believe this density is artefactual as none of the processing had included neither the atomic structure of Pds5 nor the heads as an initial model.

– The authors should compare the cohesin head domain conformations they observe, especially for the Scc2 complex, to those observed in the recent structures showing cohesin-nucleotide-Scc3-Scc2-DNA interactions. Are the current structures well enough resolved to make these comparisons? Can the authors see Scc1-N:Smc3-head domain contact?

Please see below. Figure S5D does this by comparing the binding pose of Scc2 of the ES/EK state with that of our manuscript.

We can indeed see both the Scc1-N:Smc1 and Scc1-C:Smc3 contacts at lower thresholds.

– Do the authors truly know the cohesin structure displayed in 5G is ATP-bound? To make this statement, they would need to resolve nucleotide and a connected γ phosphate.

We are confident that Figure 5G shows an ATP-bound map. The structure could only be found in the presence of ATP – a molecule that binds the ATPase heads and causes engagement – and it perfectly recapitulates the crystal structures of engaged heads bound to ATP. In light of this it would seem unreasonable to us to assume that the structure is in an *apo* state.

– Why did the authors leave out Scc3 from the cryo-EM samples? How do they relate their observations to the published structures, which show the hinge attached to Scc3/Psc3 and dislocated from the Smc1/3 head domains? Both conformations probably inform on mechanism, but the reader is not presented with a helpful way to think about this. Does Scc3 "grab" the hinge after head-Scc2 engagement, as proposed by Uhlmann's group recently (Higashi et al. 2020)? To be sure, Srinivasan et al. proposed an Scc3-dependent late step in loading already, so both of these could easily be referenced in a paragraph contextualizing the current structures with respect to Scc3.We were looking at the putative simultaneous interaction of single HAWK proteins with both hinge and heads and therefore wanted to avoid the addition of another confounding variable in the form of Scc3. We do not present any structural data regarding Scc3 in our manuscript and therefore have abstained from making any conclusions about its potential “grabbing mechanism” as we cannot support it with evidence.4. There are disagreements between data and text and confusing inconsistencies in the figures:o S1C: why is Y40A not an scc4-ts allele?

Because it grows at the restrictive temperature of 37°C (Figure S1A). We don’t know why Y40A, like Y40N, disrupts co-immunoprecipitation of Scc4 and Scc2 but does not exhibit temperature sensitivity. Presumably, the temperature sensitivity of Y40N cannot solely be due to disruption of Scc2 binding.

o S1E – It looks like smc1D588Y is an scc2-4 suppressor. The text says the opposite.

As above, this has been amended.

o Figure S3 is discussed below.

See Below.

o Genotypes should be given in color for all tetrad dissections (as in Figure 1).

Done.

o Size markers are missing from the blots. These are important for the crosslinking experiments so that positions can be compared for different antibodies (Figure 4B, for example).

Size markers are now shown on crosslinking blots.

o In at least once case, the authors should show a myc blot for Smc1-WT (no Cys mutation) so that it is clear the higher band is indeed an Smc1 crosslinking product.

This is shown in Lane 1 of Figure 4G.

o For these assays, the text and legend disagree on which subunit was immobilized for the pulldown (Scc1 or Smc1). If it was different in different experiments, this should be stated.

This has been corrected, many thanks.

o Figure S4F is not helpful. There is no control shown (either Scc2 or Smc1, and ideally both, lacking Cys mutation). The crosslinking efficiencies do not seem to correlate for the two subunits, and it is not clear why some Scc2 mutants show an even higher band.

In the end, the experiment would have only been valuable if a really great crosslink position were found. The experiment can be removed without seriously damaging the manuscript.

As described previously, we agree and have removed the figure.

– The scatter plot in figure S3E does not contradict the Lopez-Serra finding and is not a good way to look at this data. The Lopez-Serra paper addressed a small subset of chromosomal locations. It is not clear how far away from the diagonal the distribution of dots would need to spread to constitute a truly different localization pattern (no control for this).

The scatterplots have been removed.

– If the authors are sure of their placements for Scc2 and Pds5, then why not directly compare the two structures by overlaying them anchored on the coiled-coiled/hinge domains? This would be a great way of showing that i) Pds5 is indeed more closely connected to the hinge and ii) both modules engage the same overall cohesin conformation/fold (or not, as the case may be).

We thank the reviewer for the suggestion and have added a comparison to the Supplemental Figure 5D to show the region of clashes.

– Figure 5F-G make it look like the two structures are different views of the same. Only G is required for the point that head engagement does not unzip the coiled coils, so there is no need to make it look like a comparison.

We believe it to be helpful to show the respective resolutions of each map discussed in the figure, but we agree it may cause some confusion, so we have removed the arrow signalling head engagement.

– Figure S5D does not give the reader enough information to evaluate what is being shown. What dataset does this map come from? Is the corresponding main figure 5A? Was Smc3 included in the fit? DNA? Was nucleotide included in the sample? Why is there a large part of Scc2 poking out (presumably related to mentioned "floppiness")?

Figure S5D intends to show a comparison between our map and the binding pose of Scc2 described for the ES/EK state in Collier et al., 2020. The map originates from the same data set as that of Figure 5A and has been processed to remove the floppy part to produce a map with a higher resolution of the binding of Scc2 to Smc1 to allow for a better comparison. We appreciate that this information was missing from the figure legend and have now included it.

It now reads: “(D) Fitting of atomic map from Collier et al. (6ZZ6) 2020 in cryo-EM map made by focused classification. The map originates from the same data as that of Figure 5A and has been processed to remove the floppy C-terminal head domain of Scc2”.

– Neither the Murayama papers nor the Huis in 't Veld papers are discussed. The Huis in 't Veld paper reports contact between SA1 and the cohesin hinge. Murayama and Uhlmann (2015) reconstitute this interaction (Psm1/3-Psc3) for the fission yeast components and state that it means the ring must bend and that the loader is probably involved. While neither study proved hinge-Scc3 contact is essential for the loading reaction, they were important contributions and should be cited.

Thank you. The papers have now been cited and the following added to the introduction:

“Further, it has been noted that a potential simultaneous interaction of a HAWK with the hinge and kleisin would require some sort of folding (Murayama and Uhlmann, 2015, Huis in 't Veld et al., 2014, Bürmann et al., 2019).”

– Presentation issues for structures:o At least once in each figure (but not necessarily each panel), colored text should be shown for each included protein as is done in Figure 1. This is missing in Figure 5 (for example).

Done.

o The overlay in Figure 1F is hard to understand. The mutated structure should be shaded differently so the reader can see the difference clearly (pale green vs green and same for red).

Done.

o The nucleosome structure in Figure 3B is impossible to understand. The entire particle should be shown so the reader can see the DNA wrap, which is relevant here (do the mutations enhance unwrapping?). Only the mutated histones and mutated residues (highlight color) need to be colored.

Done.

[Editors' note: further revisions were suggested prior to acceptance, as described below.]

The manuscript has been improved but there are some remaining issues that need to be addressed, as outlined in the points made by reviewers below. It is particularly important to clarify where the crosslinks reside in the cohesin variant used for structural analysis, as raised in Points 2 and 3 from Reviewer 2. If the crosslinks are in the heads, as indicated in the methods section, then this has important implications for the interpretation.

We thank the reviewers for their keen eye, particularly with reference to the points on head crosslinking which was indeed a mistake on our part in the methods section. Please see our responses to each comment below for more details.

Reviewer #2:

The authors have addressed most comments. There are however a couple of issues that require further clarification:

1. Figure 4G: The author's response is only partially convincing: If Smc1K620C also crosslinks to Scc2C224, it is not fully clear that Scc2N220C specifically allows readout of contacts between the Smc1 hinge and Scc2. As they have tested a variety of Scc2 cysteine substitutions, it would have been important to test a Scc2 version that does not contain C224, in order to avoid such confounding non-specific crosslinking.

In any case, they need to further clarify how the different blots are developed. Presumably, the top panel shows an anti-FLAG western blot and the bottom panel an anti-myc blot (as indicated in the author's response)? The top panel is not labeled at all and the bottom panel is mislabeled as Anti-HA!

The reviewer is quite right, the top panel is anti-FLAG and the bottom is anti-myc. The anti-HA label corresponds the small panel in Figure 4B. We see how this is confusing and have modified the figure to make it clearer.

2. The authors state line 523 that they use a cohesin variant 'containing cysteines specifically crosslinking the three intermolecular interfaces'. They need to explicitly state which interfaces are crosslinked in the variant they use for structure determination. The author response indicates that 'the 6-cysteine version used interrogates the hinge dimerisation and not the head dimerisation'. Their methods section indicates however that the variant they use for structure determination does not contain Cys mutations in the hinges. Instead, they use Smc1G22C N1192C, Smc3S1043C R1222C, Scc1C56A547C.

These mutations are located (1) in the Smc3-NScc1 interface (Scc1C56 and Smc3S1043C), (2) Smc1-cScc1 (Scc1A547C and Smc1G22C) interface and (3), in the interface between the Smc1 and Smc3 ATPase heads (Smc1 N1192C and Smc3 R1222C). That is, the third pair of cysteines is clearly placed in the heads and not the hinges.

We profoundly apologise for the confusion caused by our mistake and thank the reviewer for pointing this out again. The correct cysteines are Smc1K639C-Smc3E570C, Smc1G22C-Scc1A547C, and Smc3S1043C-Scc1C56. We have amended the manuscript accordingly in the respective places. Therefore, our original interpretation stands, i.e. the ATPase heads have not been crosslinked to each other.

3. It is therefore not clear why the authors maintain (Reviewer #2 points 6, 7, 8, 12) that 'we did not cross-link the ATP head dimerisation interface'. Clearly, something is wrong! Maybe it is the methods section? If not, the authors need to clearly indicate that the version of cohesin they use is crosslinked at the SMC1/3 ATPase head domains. The implications of such head crosslinking need to be taken into account in the interpretation of their data and discussion of the results.

See above.

Reviewer #3:

The manuscript is greatly improved. The updated figures make it much clearer. The rewritten section describing Sth1, nucleosome positioning, and histone mutants is good. This is important work, and the breadth and strength of the experiments described are impressive.

Most of my comments have been adequately addressed. The requests below relate to the presentation and description of the data and should be addressed before publication but do not require extensive writing or any new experiments.

Line 593-597:

"In the presence of ATP" implies that ATP occupies both ATPase active sites. Neither γ phosphate can be resolved at this resolution. The structure could be a reaction intermediate resulting from slow π dissociation from one or both sites, for example. This is not to detract from the observations or statements made. Instead, it would be wise to state more clearly that ATP was included in the sample but avoid any suggestion that the structure provides detailed information about the SMC ATPase cycle, which will doubtless be a topic for future studies. This is rightly stated in the final paragraph of the Discussion.

We assume that the reviewer refers to one or both of these instances: “We identified and solved … a form of cohesin at 6Å … whose ATPase heads were engaged in the presence of ATP” and/or “A cohesin complex whose heads are engaged in the presence of ATP…”

If so, we respectfully disagree with the reviewer. What is precisely stated in our manuscript is that our engaged cohesin heads, which we explicitly mention are at no more than 6 Å resolution, were in the presence of ATP. We never allude to the state of the molecule or even hint at the possibility of seeing the gamma phosphate at this resolution. Adding that the heads could also be bound to ADP and Pi or even be in a transition would just serve to confuse the reader in an, in our opinion, very straightforward and factual statement of the sample conditions.

Figure 5G needs to be modified. Figure 5S2 shows that 5G is a composite structure derived from two different samples. One contained ATP and the other did not. Overlaying them is inappropriate. Yes, the angle of the head-proximal coiled-coils is similar (Figure 5F vs. 5G), and it is likely that the remainder of the CC density beyond (hinge-proximal) the joint is similar/identical, but this is not proven by the data in the manuscript. In fact, it appears this section of the complex was excluded from analysis in the case of the sample containing nucleotide (Figure 5G and 5S2D). Without recalculating structures or even remaking complicated figures, it must be possible to present this data more clearly. One possibility: simply removing the "Arm" density from figure 5G. Another possibility is to compare the head domain density (including head-proximal cc) from Figure 5F, which is currently obscured by Scc2 density, with the head domain reconstruction from 5G.

We thank the reviewer for the suggestion and we agree that the figure should be more transparent. We have removed the separating discontinuous line that suggested that these were two structures from different datasets and have added that the folded coiled-coil was derived from a dataset without ATP to the figure. In addition, we have referred the reader explicitly to figure 5S2D to see how the map was obtained. However, we believe that the composite structure does add valuable information that aids the reader’s understanding by putting the engaged ATPase heads into the structural context of a folded cohesion and have therefore decided to leave it with the mentioned remarks. Especially given the 2D classes in Figure 5 Supplement 1E.

Figure 5 – supplement 1A does not conclusively show complexes lacking Scc2. Compare with 5S1A, for example. One needs to consider variability in Scc2 occupancy, Scc2 position, and also the fraction and quality of the particle images contributing to these classes. The description "in samples lacking all HAWK proteins" (line 537) therefore overstates the certainty of the claim given the evidence. The statement and figure panel could be removed without damaging the overall message. Removing panel A has the added benefit that it eliminates the need to discuss a dataset not referenced in figure 5S2 (cohesin, Scc2, ATP).

The reviewer is absolutely right – this was due to a mistake while reorganising the supplementary figures 5 during the previous revision. We have added 2D classes showing exactly that, i.e, a folded cohesion complex in the absence of HAWK proteins to the figure (cf. Figure 5S1E) and have amended the text that referred to the wrong figure.